# Refined Last-Iterate Convergence Analysis with Optimism in Solving Extensive-Form Games

## Abstract

Recent studies have shown that, establishing last-iterate convergence of Counterfactual Regret Minimization (CFR) algorithms to the Nash Equilibrium (NE) of extensive-form games (EFGs), can be reformulated as establishing the last-iterate convergence in solving a sequence of (perturbed) regularized EFGs. Then, the asymptotic last-iterate convergence of $CFR^+$ algorithm, a classical CFR variant, to the NE of regularized EFGs has been proved under the reward transformation (RT) framework. However, little is known about the last-iterate convergence of $PCFR^+$, the predictive version of $CFR^+$ that incorporates optimism technique from optimization and always leads to superior theoretical/practical performance. In this work, we provide for $PCFR^+$ the last-iterate convergence to the NE of perturbed regularized EFGs. Our result involves the last-iterate convergence of $CFR^+$ as a special case, and holds for any initialization and step sizes in $PCFR^+$ algorithm. Based on this analysis, we develop a novel average-regret bound for $PCFR^+$ in the original EFGs. Such regret bound is sharper than previous one in general cases, and guarantees average-iterate convergence in solving EFGs, revealing the benefit of our last-iterate convergence analysis. Furthermore, we extend our analysis to the $PDCFR^+$ algorithm that discounts regrets from earlier iterations in $PCFR^+$. By simultaneously decreasing the perturbation and regularization, $PDCFR^+$ is guaranteed to converge to the set of NEs of original EFG with any non-increasing discounting factors. Experiments show that our developed algorithms converge faster than the latest RTCFR+ in most EFGs, and yield comparable performance to discounted CFR in large-scale Poker games.

[1] Anonymous Institution, Anonymous City, Anonymous Region, Anonymous Country. Correspondence to: Anonymous Author <anon.email@domain.com>.

Preliminary work. Under review by the International Conference on Machine Learning (ICML). Do not distribute.

## 1. Introduction

Extensive-form games (EFGs) are a foundational framework in game theory used to model sequential decision making of multiple agents with imperfect information, and have been widely applied to solve real-world multi-agent learning problems, such as auctions (Shubik, 1971), cyber security (Kakkad et al., 2019), medical treatment (Sandholm, 2015) and Poker games (Bowling et al., 2015). The goal in solving EFGs is to find a Nash Equilibrium (NE), where no agent can increase her utility by unilaterally changing her strategy.

Efficient approaches for solving EFGs include first-order methods (FOM) (Hoda et al., 2010; Kroer et al., 2020), and counterfactual regret minimization (CFR) (Zinkevich et al., 2007; Tammelin et al., 2015). The latter CFR algorithms are preferable, because of their flexibility in combination with acceleration techniques (like sampling (Lanctot et al., 2009) and function approximation (Brown et al., 2019)) and the superhuman practical performance (Moravčík et al., 2017; Brown & Sandholm, 2018). By minimizing the individual regret of each player via CFR algorithm, the average strategy over all iteration forms an approximate NE, resulting in the so-called *average-iterate* convergence in solving EFGs (Farina et al., 2019a; Brown & Sandholm, 2019).

However, averaging strategies over all iterations can lead to complications. It not only incurs more computation and memory overhead (to store historical strategies), but also induces additional optimization error when employing function approximation. For example, the Deep CFR model need to allocate a separate neural network module to approximate the average strategy (Brown et al., 2019). Such undesirable properties of CFR algorithm forces researchers to investigate its convergence of last-iterate output strategy to the NEs of EFGs, which is referred to as *last-iterate* convergence (Wei et al., 2021). But unfortunately, (Cai et al., 2025, Fig 1) and (Lee et al., 2021, Thm 1) demonstrate empirically or theoretically that, the vanilla CFR algorithm (Zinkevich et al., 2007) and its classical variant $CFR^+$ (Tammelin et al., 2015) fail to converge last iterate even in a normal-form game (NFG), which is a strict subclass of EFG. Therefore, without additional conditions or new techniques, it seems impossible to directly establish the last-iterate convergence results for the commonly used CFR/$CFR^+$ algorithms.

Reward transformation (RT) framework is such a technique proposed in recent years for proving the last-iterate convergence of CFR-type algorithms to the NEs of EFGs (Pérolat et al., 2021; 2022; Liu et al., 2023; Bernasconi et al., 2024). It modifies the utility function of the original game by adding *regularization* and *perturbation* terms, and transforms the task of finding the NEs of original EFGs into learning the NEs of a sequence of perturbed regularized EFGs. Further, RT method guarantees that the sequence of NEs of the perturbed regularized EFGs converges to the set of NEs of original EFGs, through decreasing the regularization and perturbation (if it is used) components simultaneously. Therefore, to achieve last-iterate convergence in solving original EFGs, it suffices to establish last-iterate convergence to the NE of the perturbed regularized EFGs.

Within the RT framework, Liu et al. (2023) for the first time prove that regularized counterfactual regret minimization, with a variant of online mirror descent (OMD) algorithm (Nemirovskij & Yudin, 1983) as regret minimizer at local decision points, can achieve asymptotic last-iterate convergence for finding the NE of perturbed regularized EFGs. Further, by employing the OMD-based formulation of regret matching$^+$ algorithm (i.e. the local regret minimizer in CFR$^+$ (Hart & Mas-Colell, 2000)), Meng et al. (2025a) demonstrate that the reward transformation-based CFR$^+$ yields asymptotic last-iterate convergence in solving perturbed regularized EFGs, and their developed RTCFR$^+$ algorithm shows faster empirical convergence than existing counterparts that enjoy theoretical last-iterate convergence. The empirically superior performance of RTCFR$^+$ motivates a natural question: *Can other CFR variants, especially those that outperform CFR$^+$ in original EFGs, attain last-iterate convergence in solving perturbed regularized EFGs?*

We answer this question in the affirmative, and show that PCFR$^+$ (Farina et al., 2021b), the predictive version of CFR$^+$ that incorporates *optimism* technique from optimization (Rakhlin & Sridharan, 2013a;b; Syrgkanis et al., 2015), not only enjoys last-iterate convergence in finding NE of perturbed regularized EFGs, but also retains its leading edge in experimental performance. In summary, **our contributions** are five-fold: (1) We develop the asymptotic last-iterate convergence of PCFR$^+$ in solving perturbed regularized EFGs. The refined convergence analysis includes the last-iterate convergence result for RTCFR$^+$ as a special case, and holds for any initialization of cumulative counterfactual regret and any step size (which we called *invariant* convergence). (2) Based on the invariant last-iterate convergence analysis, we establish the average regret bound for the proposed RTPCFR$^+$ algorithm in solving perturbed regularized EFGs. Such bound also holds for original PCFR$^+$, and guarantees a sharper average-iterate convergence to the NEs of original EFGs. (3) We further integrate the discounting technique (Brown & Sandholm, 2019) into the optimism framework

to discount regrets from earlier iterations in algorithm, and prove the last-/average-iterate convergence of the developed RTPDCFR$^+$ algorithm in solving perturbed EFGs. (4) Experiments on nine small-to-medium-sized EFG benchmarks show that: ($i$) the proposed RTPCFR$^+$/RTPDCFR$^+$ (without fine-tuning hyperparameters) outperform the latest RTCFR$^+$ algorithm in eight EFGs; ($ii$) The last-iterate convergence of RTPCFR$^+$ is at least five orders of magnitude faster than the average-iterate convergence of the most effective method PCFR$^+$ in five EFGs, and is about two orders of magnitude faster than PCFR$^+$ in another two EFGs (see Fig. 4). (5) Experiments on two large-scale EFGs show that, the last-iterate convergence of RTPDCFR$^+$ and its non-predictive version RTDCFR$^+$ is comparable or even faster than the convergence of state-of-the-art DCFR algorithm.

## 2. Related Work

We only describe the works for perturbed regularized EFGs here, and more details for the last-iterate convergence in EFGs, counterfactual regret minimization (CFR) algorithms, and algorithms over treeplex can be found in Appendix A.

**Last-/Average-Iterate Convergence in Perturbed Regularized EFGs.** To the best of our knowledge, (Pérolat et al., 2021) is the first one to use reward transformation technique to solve imperfect information games (i.e. EFGs) with the continuous-time feedback. (Pérolat et al., 2021, Thms 5.2 & 6.1) prove a linear last-iterate convergence in regularized EFGs, and the asymptotic last-iterate result in original EFGs for their regularized follow-the-regularized-leader algorithm. Then, under the discrete-time feedback, (Liu et al., 2023, Thms 4.1-4.2) develop the first linear last-iterate convergence for their dilated online mirror descent (OMD) algorithm in regularized EFGs, and ensure the sublinear convergence to the set of NEs in original EFGs. (Bernasconi et al., 2024, Thm 4.3) demonstrates rigorously that only letting the regularization vanish faster than the perturbation, can the sequence of NEs of perturbed regularized EFGs converge pointwise to the set of NEs in original EFGs. Under the CFR framework, (Liu et al., 2023, Eq (5.2)) utilizes dual-stabilized OMD algorithm as the local regret minimizer, and demonstrates its asymptotic last-iterate convergence and optimal $O(1/T)$ average-iterate convergence to the unique NE of perturbed regularized EFGs. (Meng et al., 2025a, Eq (3)) runs regret matching$^+$ algorithm (Tammelin et al., 2015) at local decision point to output an action strategy (i.e. probability vector), and then applies an affine transformation to this action strategy to ensure it lies within the feasible set of perturbed regularized EFGs. (Meng et al., 2025a, Thms 4.1 & F.1) further give for their RTCFR$^+$ the asymptotic last-iterate convergence and sublinear $O(1/\sqrt{T})$ average-iterate convergence to the NE of perturbed regularized EFGs. Our work follows the line of (Meng et al., 2025a), and provides refined last-iterate convergence analy-

sis and average-iterate results for the predictive/discounted versions of CFR$^+$ algorithm under RT framework.

## 3. Preliminaries

For any integer $d$, denote the set $[d] = \{0, 1, 2, \cdots, d\}$ for conciseness, $\Delta^d$ as the $(d-1)$-dimension simplex, $\|\cdot\|$ as a certain norm in Euclidean space $\mathbb{R}^d$. We use upper boldface letter $\boldsymbol{\mathcal{X}}$, lower boldface letter $\boldsymbol{x}$ and non-bold letter $x$ to denote certain space, vector and scalar respectively.

### 3.1. Imperfect-Information Extensive-Form Games

**Extensive-form games (EFGs).** An imperfect-information EFG can be characterized by a tree-based formalism $G = \{\mathcal{N}, \mathcal{H}, P, A, \mathcal{I}, \{u_i\}_{i\in\mathcal{N}}\}$ (Osborne & Rubinstein, 1994, p. 200). Concretely, $\mathcal{N}$ is the set of players, $\mathcal{H}$ the set of all possible histories/nodes. For each node $h \in \mathcal{H}$, $P(h) \in \mathcal{N}$ represents the player taking action at node $h$, and $A(h)$ denotes the actions available at node $h$. For any player $i \in \mathcal{N}$, the set of nodes $h$ that player $i$ can not distinguish their correct position in the tree $\mathcal{H}$ (according to the disclosed information) is called an **information set** (infoset), denoted as $I$. All infosets belonging to player $i$ constitute a partition $\mathcal{I}_i$ of $\mathcal{H}$, and $\mathcal{I} = \cup_{i\in\mathcal{N}}\mathcal{I}_i$. For any infoset $I \in \mathcal{I}_i$, since any two histories $h, h' \in I$ are indistinguishable to player $i$, we have $A(h) = A(h')$ and denote $A(I) \triangleq A(h)$ for simplicity. We use $C_i(I, a) \subseteq \mathcal{I}_i$ to denote the set of infosets that is encountered immediately after player $i$ choosing action $a \in A(I)$. Denote $A_{max} \triangleq \max_{I\in\mathcal{I}} |A(I)|$ and $C_{max} \triangleq \max_{i\in\mathcal{N}, I\in\mathcal{I}_i, a\in A(I)} |C_i(I, a)|$. When player $i$ reaches the terminal/leaf node $z \in \mathcal{Z} \subseteq \mathcal{H}$, it receives the utility $u_i(z) \in [-1, 1]$. In two-player zero-sum EFGs, $u_0(z) = -u_1(z), \forall z \in \mathcal{Z}$. Let $H$ be the maximum number of actions taken along any path from the root to a leaf node.

**Sequence-Form strategy and Behavioral strategy.** The sequence-form strategy space is originally proposed as a *treeplex* by (Hoda et al., 2010). For any player $i \in \mathcal{N}$, the sequence is an infoset-action pair $(I, a) \in \Sigma_i \triangleq \{(I, a) : I \in \mathcal{I}_i, a \in A(I)\}$. For any sequence-form strategy $\boldsymbol{x}_i \in \boldsymbol{\mathcal{X}}_i$, it is indexed by all sequences, and $\boldsymbol{x}_i(I, a)$ represents the probability that player $i$ reaches the sequence $(I, a)$ from the root node $r$ of tree. We assume that $\forall i \in \mathcal{N}$ and $\boldsymbol{x}_i \in \boldsymbol{\mathcal{X}}_i$, $\|\boldsymbol{x}_i\|_1 \leq D$. The sequence-form strategy $\boldsymbol{x}_i$ can be considered as the **global strategy** over the whole tree $\mathcal{H}$, and the following behavioral strategy $\sigma_i$ can be regarded as the **local strategy** defined on each inforset $I \in \mathcal{I}_i$. Concretely, the behavioral strategy $\sigma_i(I) = [\sigma_i(I, a)|a \in A(I)] \in \Delta^{|A(I)|}$ at inforset $I$ is a probability vector, where $\sigma_i(I, a)$ is the probability of player $i$ taking action $a \in A(I)$. If all players follow the strategy profile $\sigma = \{\sigma_i\}_{i\in\mathcal{N}}$, then the reaching probability to inforset $I$ is denoted by $\pi^\sigma(I)$. The contribution of $i$ to this probability is $\pi_i^\sigma(I)$ and $\pi_{-i}^\sigma(I)$ is the contribution of other players. The overall value of a strategy profile $\sigma$ to player $i$ is the expected payoff over all terminal nodes, $u_i(\sigma) = \sum_{h\in\mathcal{Z}} u_i(h)\pi^\sigma(h)$. It is not difficult to see that there exists a one-one mapping between the sequence-form strategy $\boldsymbol{x}_i \in \boldsymbol{\mathcal{X}}_i$ and the behavioral strategy $\sigma_i \in \times_{I\in\mathcal{I}_i} \Delta^{|A(I)|}$. More precisely, for any $i \in \mathcal{N}, I \in \mathcal{I}_i, a \in A(I)$, we have $\boldsymbol{x}_i(I, a) = \pi_i^\sigma(I)\sigma_i(I, a)$.

**Nash equilibrium (NE).** A NE is a behavioral strategy profile $\sigma^*$ where no player can increase her utility $u_i(\sigma^*)$ by unilaterally changing her strategy $\sigma_i^*$. Under the sequence-form strategy framework (Von Stengel, 1996; Hoda et al., 2010), learning a NE of two-player zero-sum EFGs can be reformulated as **a bilinear saddle point problem** (BSPP):

$$\min_{\boldsymbol{x}_0\in\boldsymbol{\mathcal{X}}_0} \max_{\boldsymbol{x}_1\in\boldsymbol{\mathcal{X}}_1} \boldsymbol{x}_0^\top \boldsymbol{A}\boldsymbol{x}_1, \tag{1}$$

where $\boldsymbol{A}$ is the payoff matrix, $\boldsymbol{\mathcal{X}}_0$ and $\boldsymbol{\mathcal{X}}_1$ are the sequence-form strategy spaces for player 0 and player 1 respectively. We denote $\boldsymbol{\mathcal{X}} = \times_{i\in\mathcal{N}}\boldsymbol{\mathcal{X}}_i$, and $\boldsymbol{\mathcal{X}}^*$ as the set of NEs. Then, we can define the loss $\boldsymbol{\ell}_0^\sigma = \boldsymbol{A}\boldsymbol{x}_1$ and $\boldsymbol{\ell}_1^\sigma = -\boldsymbol{A}^\mathrm{T}\boldsymbol{x}_0$ for players 0/1 respectively. Following previous works (Farina et al., 2023; Meng et al., 2025a), we assume without loss of generality that, there exist game-dependent constants $L > 0$ and $P > 0$, such that $\forall \boldsymbol{x}, \boldsymbol{x}' \in \boldsymbol{\mathcal{X}}$, $\|\boldsymbol{\ell}^\sigma - \boldsymbol{\ell}^{\sigma'}\|_1 \leq L\|\boldsymbol{x} - \boldsymbol{x}'\|_1$ and $\|\boldsymbol{\ell}^\sigma\|_1 \leq P$, where $\boldsymbol{\ell}^\sigma = [\boldsymbol{\ell}_i^\sigma | i \in \mathcal{N}]$.

### 3.2. Finding Nash Equilibriums via Online Learning

**Online Learning over Sequence-Form Strategies.** To find the NE $\boldsymbol{x}_i^* \in \boldsymbol{\mathcal{X}}^*$, player $i$ chooses to output a series of sequence-form strategies $\{\boldsymbol{x}_i^t\}_{t\geq 1}$ by running an online learning algorithm (Shalev-Shwartz et al., 2012). Under such framework, Player $i$'s cumulative regret is defined as $R_i^T \triangleq \max_{\boldsymbol{x}_i\in\boldsymbol{\mathcal{X}}_i} \sum_{t=1}^T \langle \boldsymbol{\ell}_i^t, \boldsymbol{x}_i^t - \boldsymbol{x}_i \rangle$, where $\boldsymbol{\ell}_i^t = \boldsymbol{\ell}_i^{\sigma^t}$ is the loss suffered at iteration $t$ after choosing strategy $\boldsymbol{x}_i^t$. It is well known that if all players use online algorithms to minimize the cumulative regret with an upper bound $R_i^T \leq \epsilon_i$, then the average strategy profile $\sum_{t=1}^T \boldsymbol{x}^t/T$ converges with a rate of $(\epsilon_1 + \epsilon_2)/T$ to the set $\boldsymbol{\mathcal{X}}^*$ of NEs. Such typical algorithms include excess gap technique (Hoda et al., 2010) and online mirror descent (Farina et al., 2019c), all of which need to be ran with a dilated regularization function over the sequence-form strategy space $\boldsymbol{\mathcal{X}}_i$.

**Counterfactual Regret Minimization (CFR) over Behavioral Strategies.** Instead of directly minimizing the cumulative regret $R_i^T$ of player $i$, CFR algorithm (Zinkevich et al., 2007; Farina et al., 2019b) chooses to decompose the global regret $R_i^T$ into the sum of local counterfactual regret $R_i^T(I)$ at each infoset $I$. Minimizing the counterfactual regrets $R_i^T(I)$ has been demonstrated more effectively in practice, and has led to several superhuman agents in imperfect-information EFGs (Bowling et al., 2015; Moravčík et al., 2017; Brown & Sandholm, 2018). Formally, given any strategy $\sigma$ and loss $\boldsymbol{\ell}_i^\sigma$ for player $i$, the CFR algorithm computes the **counterfactual value** $\boldsymbol{v}_i^\sigma(I, a)$ of action $a \in A(I)$ as

$$\boldsymbol{v}_i^\sigma(I, a) = -\boldsymbol{\ell}_i^\sigma(I, a) + \sum_{I'\in C_i(I,a)} \langle \boldsymbol{v}_i^\sigma(I'), \sigma_i(I') \rangle, \tag{2}$$

where $\boldsymbol{v}_i^\sigma(I) = (\boldsymbol{v}_i^\sigma(I, a))_{a \in A(I)}$, and $\sigma_i$ represents the behavioral strategy associated with $\boldsymbol{x}_i$. Then, define the **instantaneous counterfactual regret** at infoset $I$ as

$$\boldsymbol{m}_i^\sigma(I) = \boldsymbol{v}_i^\sigma(I) - \langle \boldsymbol{v}_i^\sigma(I), \sigma_i(I) \rangle \mathbf{1}. \quad (3)$$

Denote $R_i^T(I) = \sum_{t=1}^T \boldsymbol{m}_i^{\sigma^t}(I)$ as the **cumulative counterfactual regret** at $I$, (Liu et al., 2022, Lemma 2.2) obtains the following equivalence relation between $R_i^T(I)$ and cumulative regret $R_i^T$ of player $i$ (see details in Lemma D.6):

$$R_i^T = \max_{\sigma'} \sum_{I \in \mathcal{I}_i} \pi_i^{\sigma'}(I) \langle R_i^T(I), \sigma_i'(I) \rangle \quad (4)$$

Thus, it is sufficient for CFR method to select an online algorithm to output strategies $\{\sigma_i^t\}_{t \geq 1}$ and minimize the local regret $\langle R_i^T(I), \sigma_i'(I) \rangle = \sum_{t=1}^T \langle \boldsymbol{v}_i^{\sigma^t}(I), \sigma_i' - \sigma_i^t \rangle$ in the right-hand side of Eq. (4) for any reference strategy $\sigma_i'$.

**Regret Matching$^+$ as Local Regret Minimizer in CFR.** When setting the local regret minimizer as regret matching$^+$ (RM$^+$) algorithm, we get CFR$^+$ (Tammelin, 2014), the classical CFR variant that always converges faster in practice than the original CFR algorithm (Zinkevich et al., 2007). By applying Blackwell approachability technique (Blackwell, 1956; Abernethy et al., 2011), (Farina et al., 2021b) further show that RM$^+$ is equivalent to running online mirror descent (OMD) algorithm over the convex cone of the simplex $\Delta^{|A(I)|}$, i.e. $\mathbb{R}_{\geq 0}^{|A(I)|}$, and normalizing the output vector to get the behavioral strategy. Concretely, at each iteration $t$ and infoset $I \in \mathcal{I}_i$, the update rule of OMD algorithm is

$$\boldsymbol{\theta}_I^{t+1} \in \underset{\boldsymbol{\theta}_I \in \mathbb{R}_{\geq 0}^{|A(I)|}}{\arg\min} \langle -\boldsymbol{m}_i^t(I), \boldsymbol{\theta}_I \rangle + \frac{1}{\eta} D_\psi(\boldsymbol{\theta}_I \| \boldsymbol{\theta}_I^t), \quad (5)$$

where $\eta > 0$ is the constant step size, $\boldsymbol{m}_i^t(I) = \boldsymbol{m}_i^{\sigma^t}(I)$ is the instantaneous counterfactual regret associated with strategy profile $\sigma^t$, and $D_\psi(\boldsymbol{u} \| \boldsymbol{v}) = \psi(\boldsymbol{u}) - \psi(\boldsymbol{v}) - \langle \nabla\psi(\boldsymbol{v}), \boldsymbol{u} - \boldsymbol{v} \rangle$ is the Bregman divergence generated by function $\psi(\cdot) = \frac{1}{2} \| \cdot \|_2^2$. Then the output behavioral strategy is normalized as $\sigma_i^{t+1}(I) = \boldsymbol{\theta}_I^{t+1} / \langle \boldsymbol{\theta}_I^{t+1}, \mathbf{1} \rangle$. (Farina et al., 2021b) further show that $\sigma_i^{t+1}(I)$ remains unchanged for any step size $\eta > 0$, which is referred to as *infoset step size invariance* and is proved as a key property for the practical success of CFR$^+$ algorithm (Chakrabarti et al., 2024).

## 4. Optimization Objective

**Perturbed Extensive-Form Games (Perturbed EFGs).** Such game is a variant of the original (non-perturbed) EFG, to ensure that each player reaches every infoset with at least a positive probability $\gamma \in (0, 1)$. Concretely, for any infoset $I \in \mathcal{I}_i$, the behavioral strategy space over the action set $A(I)$ in a $\gamma$-perturbed EFG is a $\gamma$-perturbed simplex $\Delta_\gamma^{|A(I)|} \subset \Delta^{|A(I)|}$, where for any $\hat{\sigma}_i(I) \in \Delta_\gamma^{|A(I)|}$, we have $\gamma \leq \hat{\sigma}_i(I, a) \leq 1$ for any action $a \in A(I)$. Correspondingly, we denote the sequence-form strategy space for

player $i$ as $\mathcal{X}_i^\gamma$, where for any $\hat{x}_i \in \mathcal{X}_i^\gamma$, $I \in \mathcal{I}_i$, we have $\hat{x}_i(I) = \pi_i^{\hat{\sigma}}(I) \hat{\sigma}_i(I) \geq \pi_i^{\hat{\sigma}}(I) \gamma \mathbf{1}$. We denote $\mathcal{X}^\gamma$ and $\mathcal{X}^{*,\gamma}$ as the joint strategy space $\times_{i \in \mathcal{N}} \mathcal{X}_i^\gamma$ and the set of NEs of $\gamma$-perturbed EFGs, respectively. Notice that $\mathcal{X}^\gamma \subset \mathcal{X}$. When $\gamma \to 0$, the limit $\mathcal{X}^{*,\gamma}$ is the set of extensive-form perfect equilibria (EFPE) (Selten, 1975), a refinement of NE.

**Introduction of Reward Transformation Framework.** Recent works Pérolat et al. (2021; 2022); Liu et al. (2023); Abe et al. (2024); Meng et al. (2025a) adopt the Reward Transformation (RT) framework to demonstrate the last-iterate convergence of algorithms in solving EFGs. Concretely, these works transform the original EFG into a sequence of perturbed regularized EFGs by adding perturbation and regularization terms, thus the reward function (i.e. the loss $\boldsymbol{\ell}^\sigma$) will be changed. $\forall \hat{x}, \hat{x}' \in \mathcal{X}^\gamma$, the introduction of perturbation ensures the smoothness of counterfactual value $\|\boldsymbol{v}_i^{\hat{\sigma}}(I) - \boldsymbol{v}_i^{\hat{\sigma}'}(I)\|_1 \leq O(\|\hat{x} - \hat{x}'\|_1)$, and the regularization term ensures the strongly monotonicity of loss function $O(\langle \boldsymbol{\ell}^{\hat{\sigma}} - \boldsymbol{\ell}^{\hat{\sigma}'}, \hat{x} - \hat{x}' \rangle) \geq \|\hat{x} - \hat{x}'\|_2^2$, which also guarantees the uniqueness of Nash equilibrium (Sokota et al., 2023). Therefore, it suffices to establish the last-iterate convergence to the NEs of perturbed regularized EFGs, and ensure the sequence of NEs of these perturbed regularized EFGs converges to the set of NEs of original EFG.

**Perturbed Regularized Extensive-Form Games.** Formally, the perturbed regularized EFG is defined as follow:

$$\min_{\hat{x}_0 \in \mathcal{X}_0^\gamma} \max_{\hat{x}_1 \in \mathcal{X}_1^\gamma} \hat{x}_0^\top A \hat{x}_1 + \mu D_\psi(\hat{x}_0, r_0) - \mu D_\psi(\hat{x}_1, r_1), \quad (6)$$

where $\gamma > 0$ and $\mu > 0$ are constants, and $\boldsymbol{r} = [\boldsymbol{r}_0; \boldsymbol{r}_1] \in \mathcal{X}$ is the reference sequence-form strategy profile. The unique NE of this perturbed regularized EFG is denoted by $\hat{x}^{*,\gamma,\mu,\boldsymbol{r}}$ or $\hat{\sigma}^{*,\gamma,\mu,\boldsymbol{r}}$, which is abbreviated as $\hat{x}^{*,\gamma}$ or $\hat{\sigma}^{*,\gamma}$ for conciseness throughout the rest of this paper. The transformed reward/loss is then defined as follow:

$$\begin{aligned} \boldsymbol{\ell}_0^{\hat{\sigma}} &= A \hat{x}_1 + \mu \nabla\psi(\hat{x}_0) - \mu \nabla\psi(\boldsymbol{r}_0), \\ \boldsymbol{\ell}_1^{\hat{\sigma}} &= -A^\top \hat{x}_0 + \mu \nabla\psi(\hat{x}_1) - \mu \nabla\psi(\boldsymbol{r}_1), \end{aligned} \quad (7)$$

$\boldsymbol{v}_i^{\hat{\sigma}}(I)$ and $\boldsymbol{m}_i^{\hat{\sigma}}(I)$ can be defined as in Eqs. (2)-(3) by replacing the original loss $\boldsymbol{\ell}_i^\sigma$ with $\boldsymbol{\ell}_i^{\hat{\sigma}}$ in Eq. (7). During the online learning process, algorithm generates a sequence of perturbed sequence-form strategy profiles $\{\hat{x}^t\}_{t \geq 1}$ (associated with the perturbed behavioral strategy profiles $\{\hat{\sigma}^t\}_{t \geq 1}$), and the corresponding sequence of transformed losses is denoted as $\{\hat{\boldsymbol{\ell}}^t = \boldsymbol{\ell}^{\hat{\sigma}^t}\}_{t \geq 1}$ based on Eq. (7). By continuously decreasing the value of $\gamma, \mu$ (with $\mu$ decreasing faster than $\gamma$) and updating $\boldsymbol{r}$ to $\hat{x}^{*,\gamma}$, (Bernasconi et al., 2024, Thm 4.3) rigorously proves that the sequence of NEs of perturbed regularized EFGs converges to the set of NEs of original EFG. Thus, establishing the last-iterate convergence for the perturbed regularized BSPP in Eq. (6) indicates the last-iterate convergence for the original BSPP in Eq. (1). Previous work (Meng et al., 2025a) has shown the last-iterate convergence

of CFR$^+$ algorithm for BSPP in Eq. (6). In the next section, we give a refined last-iterate convergence analysis for the predictive/optimistic versions of CFR$^+$ algorithm in solving BSPP of Eq. (6). The main technical challenges of our last-iterate convergence analysis are detailed in Remark D.9.

## 5. Convergence Analysis of RTPCFR$^+$

Within the Blackwell approachability framework (Farina et al., 2021b; Chakrabarti et al., 2024), PCFR$^+$ algorithm is equivalent to running optimistic online mirror descent (OOMD) algorithm (Rakhlin & Sridharan, 2013a;b) over the convex cone of simplex and then normalizing the output vector. Formally, at iteration $t$ and infoset $I \in \mathcal{I}_i$ , the update rule of PCFR$^+$ with step size $\eta > 0$ for learning an NE of the perturbed regularized BSPP in Eq. (6) is

$$
\boldsymbol{\theta}_I^{t+1} \in \arg\min_{\boldsymbol{\theta}_I \in \mathbb{R}_{\geq 0}^{|A(I)|}} \langle -\hat{\boldsymbol{m}}_i^t(I), \boldsymbol{\theta}_I \rangle + \frac{1}{\eta} D_\psi(\boldsymbol{\theta}_I \| \hat{\boldsymbol{\theta}}_I^t),
$$
$$
\hat{\boldsymbol{\theta}}_I^{t+1} \in \arg\min_{\boldsymbol{\theta}_I \in \mathbb{R}_{\geq 0}^{|A(I)|}} \langle -\hat{\boldsymbol{m}}_i^{t+1}(I), \boldsymbol{\theta}_I \rangle + \frac{1}{\eta} D_\psi(\boldsymbol{\theta}_I \| \hat{\boldsymbol{\theta}}_I^t).
$$

(8)

where the instantaneous counterfactual regret is denoted as $\hat{\boldsymbol{m}}_i^t(I) = \boldsymbol{m}_i^{\hat{\sigma}^t}(I)$ for conciseness, the distance generating function $\psi(\boldsymbol{\theta}) = \frac{1}{2}\|\boldsymbol{\theta}\|_2^2$. The output unnormalized strategy of the first update step in Eq. (8) has a closed-form solution $\boldsymbol{\theta}^{t+1} = [\hat{\boldsymbol{\theta}}^t + \eta\hat{\boldsymbol{m}}_i^t(I)]^+$. Then, the output behavioral strategy is $\sigma_i^{t+1}(I) = \boldsymbol{\theta}_I^{t+1}/\langle \boldsymbol{\theta}_I^{t+1}, \mathbf{1} \rangle$, and the perturbed behavioral strategy for solving Eq. (6) is computed as $\hat{\sigma}_i^{t+1}(I) = (1 - \alpha_I)\sigma_i^{t+1}(I) + \gamma\mathbf{1}$, where $\alpha_I = \gamma|A(I)|$. The corresponding perturbed sequence-form strategy can be determined as $\hat{\boldsymbol{x}}_i^{t+1}(I) = \pi_i^{\hat{\sigma}^{t+1}}(I)\hat{\sigma}_i^{t+1}(I)$. The aforementioned whole process results in the RTPCFR$^+$ algorithm for solving the perturbed regularized EFGs in Eq. (6), and its pseudocode is exhibited in Algorithm 1.

Besides, it is not difficult to see that the update rule of CFR$^+$ algorithm in Eq. (5) is a special case of PCFR$^+$ by setting $\hat{\boldsymbol{m}}_i^t(I) = 0$ in the first update of Eq. (8). Therefore, the following last-/average-iterate convergence results for RTPCFR$^+$ also hold for the reward transformation-based CFR$^+$ algorithm (i.e. RTCFR$^+$) in (Meng et al., 2025a).

### 5.1. Invariant Last-Iterate Convergence with Optimism

We first give the asymptotic last-iterate convergence of RTPCFR$^+$ algorithm as follow. The main differences between our proof techniques and that of the most related work (Meng et al., 2025a) are listed in Remark E.2.

**Theorem 5.1.** *Assume that all players follow the update rule of RTPCFR$^+$ with any $\hat{\boldsymbol{\theta}}_I^0 \in \mathbb{R}_{\geq 0}^{|A(I)|}$ and $\eta > 0$, then the strategy profiles $\{\hat{\boldsymbol{x}}^t\}_{t \geq 1}$ converges to the NE of perturbed regularized EFGs in Eq. (6) with any $\gamma > 0$ and $\mu > 0$.*

Im the perturbed regularized EFGs settings, Theorem 5.1 provides the first asymptotic last-iterate convergence for op-

timistic CFR algorithms that holds for any step size $\eta > 0$ and any initialization (which we call the ***invariant last-iterate convergence***, hereafter). Previous asymptotic last-iterate convergence with any step size $\eta > 0$ holds for the non-optimistic CFR variant, CFR$^+$ (Meng et al., 2025a). Another asymptotic last-iterate convergence in perturbed regularized EFGs within the CFR framework is shown in (Liu et al., 2023, Thm 5.3) for Reg-CFR algorithm, which uses a stabilized optimistic OMD algorithm as the local regret minimizer with adaptive step sizes. However, Reg-CFR (Liu et al., 2023, Eq.(5.2)) directly solves the constrained optimization problem over the $\gamma$-perturbed simplex, which requires additional projection to the $\gamma$-perturbed simplex for implementation and is less efficient than our RTPCFR$^+$ that has a closed-form behavioral strategy $\hat{\sigma}_i(I)$. We further give the following remark to compare the conditions for the last-iterate convergence of different algorithms in EFGs.

*Remark 5.2. Most last-iterate convergence results hold for online algorithms ran directly over the sequence-form strategy $\boldsymbol{x}_i \in \mathcal{X}_i$, and such algorithms require their step sizes to be small enough or satisfy certain constraint. For example, (1) the Optimistic OMD algorithm (Lee et al., 2021, Thm 4) and its regularized version Reg-DOMWU/Reg-DOGDA (Liu et al., 2023, Thm 4.1) require the step size $\eta \leq O(1/P)$ to achieve asymptotic or linear last-iterate convergence, where $P$ is the number of all infosets and is about $10^{16}$ in two-player Heads-up Limit Texas Hold'em. (2) The famous algorithm in multiplayer general-sum EFG setting, kernelized OMWU (Farina et al., 2022, Thm 5.5), requires its step size $\{\eta^t\}_{t \geq 1}$ to be adaptive or $\eta \leq 1/8$ (under the unique NE assumption) to achieve asymptotic/linear last-iterate convergence. (3) In contrary, by continually decreasing $\gamma, \mu$ as in (Bernasconi et al., 2024, Sect 4.2), our proposed RTPCFR$^+$ achieves the asymptotic last-iterate convergence for any step size $\eta > 0$, partially indicating the flexibility of RTPCFR$^+$ and the potential of RT framework in EFGs.*

### 5.2. Novel Average-Iterate Convergence for PCFR$^+$

Based on the invariant last-iterate convergence in Theorem 5.1 that holds for any initialization $\hat{\boldsymbol{\theta}}^0$ and any step size $\eta > 0$, we further establish the upper bound on cumulative regret $R_i^T$ and the average-iterate convergence of RTPCFR$^+$ for perturbed regularized EFGs with any $\gamma \geq 0$ and $\mu \geq 0$.

**Theorem 5.3.** *Assume that player $i$ follow the update rule of RTPCFR$^+$ to solve the perturbed regularized EFGs in Eq. (6) with any $\gamma \geq 0$ and $\mu \geq 0$. Then, for any initialization $\hat{\boldsymbol{\theta}}_I^0 \in \mathbb{R}_{\geq 0}^{|A(I)|}$ and step size $\eta > 0$, we have*

$$
R_i^T \leq \sum_{I \in \mathcal{I}_i} 2\sqrt{\frac{\|\hat{\boldsymbol{\theta}}_I^0\|_2^2}{\eta^2} + \frac{1}{2}\sum_{t=0}^{T-1} \|\hat{\boldsymbol{m}}_i^{t+1}(I) - \hat{\boldsymbol{m}}_i^t(I)\|_2^2}.
$$

The above regret bound holds not only for the strictly perturbed regularized EFG with $\gamma > 0, \mu > 0$, but also holds

for PCFR$^+$ algorithm in solving the original non-perturbed EFG with $\gamma = \mu = 0$. When $\hat{\boldsymbol{\theta}}_I^0 = \boldsymbol{0}$, it recovers the regret bound for PCFR$^+$ in (Farina et al., 2021b, Thm 3). To the best of our knowledge, this is the first time to derive regret bound for the state-of-the-art PCFR$^+$ algorithm without its original and more complicated Blackwell approachability analysis (Farina et al., 2021b, Prop 2). We then give the following remark to show the key differences between them.

*Remark 5.4.* *(Farina et al., 2021b, Eq. (24)) gives the regret bound for PCFR$^+$ at an inforset with any initialization $\hat{\boldsymbol{\theta}}_I^0 \in \mathbb{R}_{\geq 0}^{|A(I)|}$ and step size $\eta > 0$ as follow:*

$$\frac{1 + \|\hat{\boldsymbol{\theta}}_I^0\|_2^2}{\eta} + \eta \sum_{t=0}^{T-1} \|\hat{\boldsymbol{m}}_i^{t+1}(I) - \hat{\boldsymbol{m}}_i^t(I)\|_2^2, \qquad (9)$$

*which is of order $O(1 + \sum_t \|\hat{\boldsymbol{m}}_i^{t+1}(I) - \hat{\boldsymbol{m}}_i^t(I)\|_2^2)$ and is larger than our bound $O(\sqrt{1 + \sum_t \|\hat{\boldsymbol{m}}_i^{t+1}(I) - \hat{\boldsymbol{m}}_i^t(I)\|_2^2})$ in Thm 5.3. Even setting the optimal $\eta$ in Eq. (9), its lowest value $2\sqrt{(1 + \|\hat{\boldsymbol{\theta}}_I^0\|_2^2) \sum_{t=0}^{T-1} \|\hat{\boldsymbol{m}}_i^{t+1}(I) - \hat{\boldsymbol{m}}_i^t(I)\|_2^2}$ is larger than ours in Thm 5.3 when $\sum_{t=0}^{T-1} \|\hat{\boldsymbol{m}}_i^{t+1}(I) - \hat{\boldsymbol{m}}_i^t(I)\|_2^2 \geq 1/\eta^2$, validating the improvement of our new average-iterate convergence analysis.*

Although our average-iterate convergence holds for any step size $\eta > 0$ in solving perturbed regularized EFGs (including the case $\gamma = \mu = 0$), we give the following remark to point out the double-edged nature of this *infoset step size invariance* property (Chakrabarti et al., 2024, Sect 4).

*Remark 5.5.* *(1) Fast average-iterate convergence in EFGs always requires the step size of algorithm to satisfy certain constraints: (i) For the two-player zero-sum EFGs: Under the CFR framework, the fast average-iterate convergence $O(1/T)$ of conceptual RM$^+$ (with additional $O(T \log T)$ gradient computations) in (Farina et al., 2023, Sect J) requires the step size $\eta \leq O(1/P)$, where $P$ is the number of total information-action pairs of all players; For the first-order method with dilated regularization, to achieve $O(1/T)$ convergence rate, EGT (Hoda et al., 2010; Kroer et al., 2020; Farina et al., 2021a) needs to shrink step size at each iteration and OOMD (Farina et al., 2019c, Sect 2) requires the constant step size smaller than the reciprocal of the payoff matrix norm $\|\boldsymbol{A}\|$. (ii) For the multi-player general-sum EFGs: kernelized OMWU (Farina et al., 2022, Thm 5.5) requires step size $\eta \leq O(1/\log T)$ to achieve the near-optimal rate $O(\log^4 T/T)$, DLRC-OMWU uses dynamic step sizes to achieve $O(\log T/T)$ rate (Soleymani et al., 2025, Sect 4). (2) In contrast, our infoset-stepsize-invariant RTPCFR$^+$ has a suboptimal rate $O(\sqrt{\sum_{t=0}^{T-1} \|\hat{\boldsymbol{m}}_i^{t+1}(I) - \hat{\boldsymbol{m}}_i^t(I)\|_2^2}/T) = O(1/\sqrt{T})$, and (Farina et al., 2023, Fig 1) shows for PCFR$^+$ an empirical convergence rate of $\Omega(1/\sqrt{T})$. Therefore, combining the analysis in (1)-(2), although (Chakrabarti et al., 2024) demonstrates the* infoset step size invariance *is the key com-*

*ponent for the empirical success of PCFR$^+$ algorithm, we conjecture that such invariance property is also the greatest barrier to achieving theoretically fast average-iterate convergence rate for PCFR$^+$ and its variants enjoying this inforset step size invariance property, e.g. CFR$^+$.*

Nevertheless, our cumulative regret bound in Thm 5.3 is of order $O(\sqrt{T})$ for perturbed regularized EFGs, which is slower than the (optimal) constant bound $O(1)$ in (Liu et al., 2023, Thm 5.6) for Reg-CFR algorithm that uses more complex adaptive step size $\{\eta_I^t\}_{t \geq 1}$ at distinct infosets $I$. More comparisons between average regret bounds of different CFR variants are shown in Table 2 of App. C.

# 6. Convergence Analysis of RTPDCFR$^+$

Besides the aforementioned optimism approach (Syrgkanis et al., 2015), another famous technique in game to accelerate convergence to NEs of EFGs is discounting (Brown & Sandholm, 2019), which discounts players' regrets from earlier iterations. In particular, its CFR version, named DCFR (Brown & Sandholm, 2019), has shown promising advantage in solving large-scale EFGs (like heads-up no-limit Texas hold'em (HUNL)) over the vanilla CFR variants (e.g. CFR/CFR$^+$, see (Brown & Sandholm, 2019, Figs 1-5)) and even over PCFR$^+$ (see (Farina et al., 2021b, Fig 2)), the new state-of-the-art in solving small-to-medium-size EFGs.

In this section, we take into account the discounting technique under the reward transformation framework, and integrate such technique into the RTPCFR$^+$ algorithm, resulting in the RTPDCFR$^+$ algorithm. Notice that when applying RTPDCFR$^+$ to solve original EFGs (with $\gamma = \mu = 0$), such algorithm reduces to the PDCFR$^+$ or its non-optimistic version DCFR$^+$ (Xu et al., 2024), which combined with the weighted-average strategy has shown superior performance at almost all EFG benchmarks (including the large-scale HUNL Subgame, see (Xu et al., 2024, Fig 2)).

Formally, the update rule of PDCFR$^+$ for learning an NE of the perturbed regularized EFGs in Eq. (6) at iteration $t$ and infoset $I \in \mathcal{I}_i$ is (we write $\lambda_I^t$ as $\lambda^t$ for simplicity):

$$\begin{aligned}
\boldsymbol{\theta}_I^{t+1} &= [\lambda^{t+1}\hat{\boldsymbol{\theta}}_I^t + \hat{\boldsymbol{m}}_i^t(I)]^+, \\
\hat{\boldsymbol{\theta}}_I^{t+1} &= [\lambda^{t+1}\hat{\boldsymbol{\theta}}_I^t + \hat{\boldsymbol{m}}_i^{t+1}(I)]^+,
\end{aligned} \qquad (10)$$

where $\{\lambda^t\}_{t \geq 0}$ is the non-decreasing positive sequence with an upper bound $\lambda$ (i.e. $0 < \lambda^t \leq \lambda^{t+1} \leq \lambda$), and the output behavioral strategy is computed as $\sigma_i^{t+1}(I) = \boldsymbol{\theta}_I^{t+1}/\langle \boldsymbol{\theta}_I^{t+1}, \boldsymbol{1} \rangle$, $\hat{\sigma}_i^{t+1}(I) = (1 - \alpha_I)\sigma_i^{t+1}(I) + \gamma\boldsymbol{1}$. Actually, due to the linearity of the ReLU function $[\cdot]^+ = \max(\cdot, 0)$ and the $\ell_1$-normalizing operator, the update rule of PDCFR$^+$ in Eq. (10) is equivalent to running optimistic OMD algorithm in Eq. (8) with non-increasing step sizes $\{\eta^t\}_{t \geq 0}$ (i.e. $\lambda^t = 1/\eta^t$). The non-decreasing positive sequence $\{\lambda^t\}_{t \geq 0}$ is widely used in solving EFGs, for ex-

ample, in DCFR$^+$ it is $\lambda^t = \frac{t^\alpha}{1+t^\alpha} \leq 1$ (with $\alpha > 0$) and is called *discounting factor*; in Reg-CFR (Liu et al., 2023) and AdOGD (Zhang et al., 2025), the adaptive coefficient $\lambda_I^t = \sqrt{\sum_{s=1}^{t-1} \|\hat{m}_i^s(I) - \hat{m}_i^{s-1}(I)\|^2 + \delta}$ with $\delta > 0$.

### 6.1. Invariant Last-Iterate Convergence with Discounting

Then, we give the asymptotic last-iterate convergence for RTPDCFR$^+$. Such asymptotic last-iterate convergence also holds for the reward transformation-based DCFR$^+$ (Li et al., 2024), and holds for RT-based AdOGD algorithms (Zhang et al., 2025) if the adaptive step size of AdOGD at any information set is upper bounded.

**Theorem 6.1.** *Assume that all players follow the update rule of RTPDCFR$^+$ with any $\hat{\theta}_I^0 \in \mathbb{R}_{\geq 0}^{|A(I)|}$ and any non-decreasing upper-bounded sequence $\{\lambda^t\}_{t\geq 1}$, the strategy profile $\hat{x}^t$ converges to the set of NEs of the perturbed regularized EFGs in Eq. (6) with any $\gamma > 0$ and $\mu > 0$.*

We then have several observations: **(1)** The required (any) non-decreasing $\lambda^t$ (i.e. non-increasing $\eta^t$ in Eq. (8)) includes the (any) constant step size $\eta > 0$ in Theorem 5.1 as a special case, and so holds in a more general situation. According to our proof, it is the linearity of ReLU function in OOMD algorithm with $\ell_2$-regularization function (which does not exist in optimistic MWU algorithm) that makes the key contribution to the invariant last-iterate convergence of RTPDCF$^+$ for *any* non-decreasing sequence $\{\lambda^t\}_{t\geq 1}$. **(2)** Another insight is from the boundedness of $\lambda^t$. (Hsieh et al., 2021, Sect 3) actually shows that there exists a simple BSPP such that running OOMD (i.e. Eq. (10)) with any $\lambda^t < \sqrt{3}$ fails to converge, and they suggest a common remedy to set $\lambda^t \propto \sqrt{t}$ to increase regularization for convergence. In Theorem 6.1, we also increase regularization, but require an upper limit. The last-iterate convergence still holds, mainly due to the introduction of reward transformation. **(3)** By simultaneously decreasing the value of $\gamma, \mu$ (with $\mu$ vanishing faster) and updating $r$ to $\hat{x}^{*,\gamma}$ as in (Bernasconi et al., 2024, Sect 4), integrating non-increasing discounting technique into the optimistic OMD framework guarantees the last-iterate convergence to the set of NEs of original EFGs.

### 6.2. Refined Average-Iterate Convergence for PDCFR$^+$

In this section, we further give the average-iterate convergence of RTPDCFR$^+$ algorithm in solving the perturbed regularized EFGs. Specifically, we demonstrate the convergence of weighted-average strategy profile $\bar{x}^w = \sum_{t=1}^T w^t x^t / \sum_{t=1}^T w^t$ with $w^t > 0$ for any $t \geq 1$ as in (Xu et al., 2024). When $w^t = 1$, the weighted-average strategy reduces to the uniformly-average strategy $\sum_{t=1}^T x^t / T$.

**Theorem 6.2.** *Assume that player $i$ follow the update rule of RTPDCFR$^+$ to solve the perturbed regularized EFGs in Eq. (6) with any $\gamma \geq 0$ and $\mu \geq 0$. If $\{w^t\lambda^t\}_{t\geq 0}$ is a positive non-increasing sequence, then, for any ini-*

*tialization $\hat{\theta}_I^0 \in \mathbb{R}_{\geq 0}^{|A(I)|}$, the $w$-weighted regret $R_i^{T,w} = \sum_{t=0}^{T-1} w^{t+1} \langle \hat{\ell}_i^{t+1}, \hat{x}_i^{t+1} - \hat{x}_i \rangle$ is upper bounded by*

$$2\sum_{I \in \mathcal{I}_i} \sqrt{w^1\lambda^1\Big[w^1\lambda^1\|\hat{\theta}_I^0\|_2^2 + \sum_{t=0}^{T-1} \frac{w^{t+1}}{2\lambda^{t+1}}\|\hat{m}_i^{t+1}(I) - \hat{m}_i^t(I)\|_2^2\Big]}.$$

According to (Brown & Sandholm, 2014, Cor 2), Theorem 6.2 guarantees that the weighted-average sequence-form strategy profile $\bar{x}^w$ converges with a rate $O(\sum_{I \in \mathcal{I}} \sqrt{\sum_{t=0}^{T-1} \frac{w^{t+1}}{2\lambda^{t+1}}\|\hat{m}_i^{t+1}(I) - \hat{m}_i^t(I)\|_2^2}/(\sum_{t=1}^T w^t))$ to the NEs of perturbed regularized EFGs. Besides, when initialization $\hat{\theta}_I^0 = 0$, the above regret bound recovers the result in (Xu et al., 2024, Thm 4) for the original EFGs with $\gamma = \mu = 0$. Nevertheless, we should point out one limitation of the above result: even with our refined analysis from the last-iterate convergence for perturbed regularized EFGs (i.e. the invariance of output sequence strategy for any non-decreasing sequence $\{\lambda^t\}_{t\geq 1}$), the regret bounds in Theorem 6.2 and (Xu et al., 2024, Thm 4) only hold with the strictly decreasing weights $\{w^t\}_{t\geq 1}$, to ensure the non-increasing sequence $\{w^t\lambda^t\}_{t\geq 0}$. However, such weighted-average strategy is rarely used in practice, due to the growing importance of subsequent strategies in EFGs.

## 7. Experiments

**Experimental Settings.** We conduct experiments on nine small-to-medium-sized instances of five EFG benchmarks: Kuhn Poker, Leduc Poker, Goofspiel, Liar's Dice, and Battleship (see Table 3 for game details). **Source code is provided in supplementary material**. We compare our RTPCFR$^+$ with three groups of baselines: (1) Traditional CFR algorithms like CFR$^+$ (Tammelin et al., 2015), DCFR (Brown & Sandholm, 2019), and PCFR$^+$ (Farina et al., 2021b); (2) Recent algorithms that update the sequence-form strategy over the treeplex and enjoy last-iterate convergence, like OMWU and OGDA (Lee et al., 2021). (3) Recent algorithms that is proposed within the reward transformation framework to solve perturbed regularized EFGs at each information sets, with the last-iterate convergence guarantees, like Reg-CFR (Liu et al., 2023) and RTCFR$^+$ (Meng et al., 2025a). All algorithms are built on the open-source LiteEFG code (Liu et al., 2024), whose implementation is 100 times faster than the implementation from OpenSpiel (Lanctot et al., 2019). For RTPCFR$^+$, we set the initial values $\gamma = 1e{-}10$, $\mu = 1e{-}4$, the number of iterations $T_u = 100$ to update $r$. We set hyperparameters of other baselines as in their original papers (see App. J). All algorithms are ran for 20,000 ($N = 20000/T_u$) iterations. More experiments, including the last-iterate results of RTPDCFR$^+$ (Fig. 2), the last-iterate convergence of CFR variants in large-scale EFGs (Fig. 3), the comparison between last-iterate convergence of RTPCFR$^+$ and average-iterate convergence of CFR variants (Fig. 4), are deferred to Appendix J.

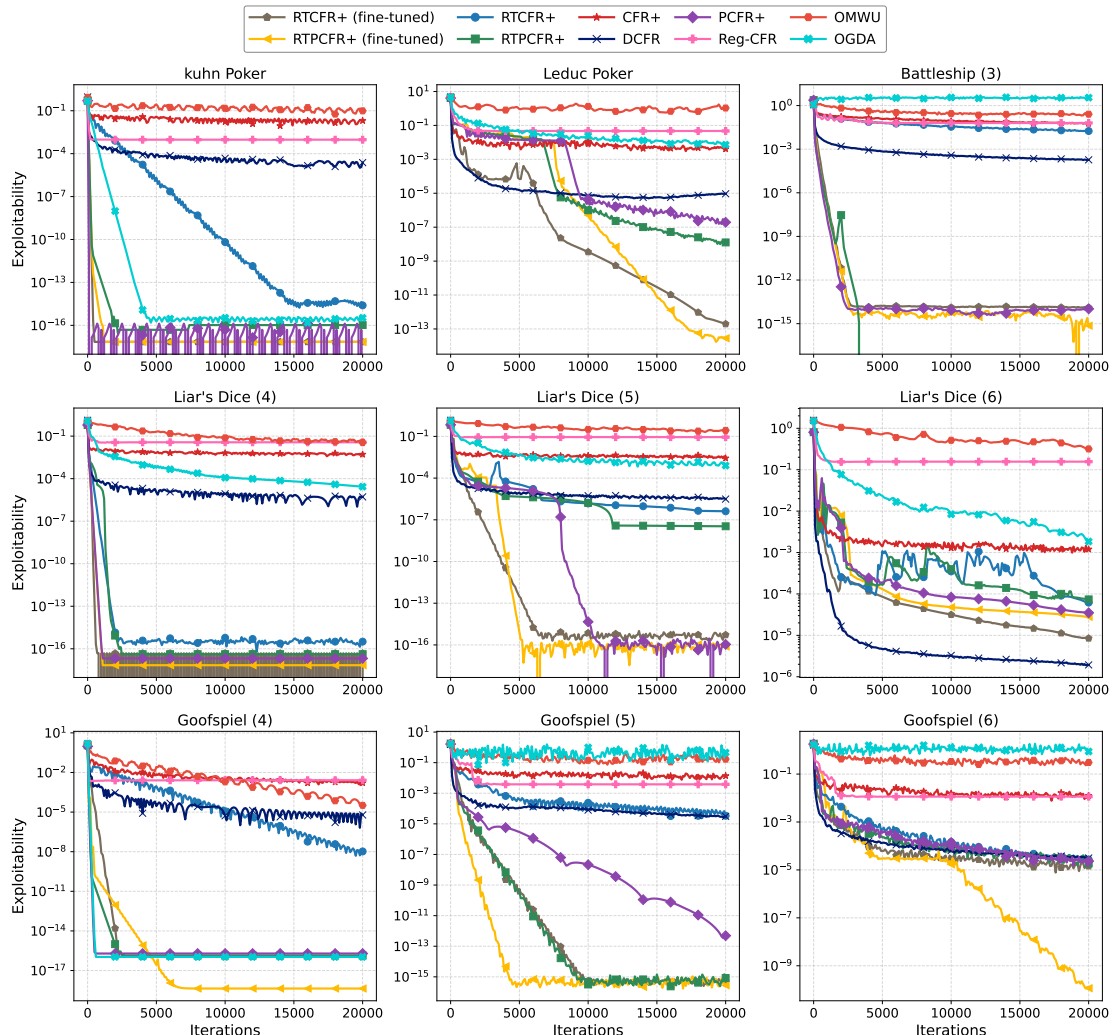

*Figure 1.* Last-iterate convergence of different algorithms in EFGs.

**Experimental Results.** According to Fig. 1, we can conclude that: **(1)** RTPCFR$^+$ (without fine-tuning hyperparameters) achieves faster convergence than the latest RT-based method RTCFR$^+$ in most benchmarks except Leduc Poker. In some EFGs like Battleship, Goofspiel, and Goofspiel (5), RTPCFR$^+$ outperforms RTCFR$^+$ by over five orders of magnitude in speed. **(2)** RTPCFR$^+$ (fine-tuned) also converges faster than RTCFR$^+$ (fine-tuned) in all experiments, and such improvement is up to five orders of magnitude in Goofspiel (6), demonstrating the strengths of optimism technique in RT framework. Besides, RTPCFR$^+$ (fine-tuned) achieves the best performance in 8 EFG benchmarks, except Liar's Dice (6), revealing the potential of reward transformation framework in solving EFGs. **(3)** RTPCFR$^+$ (without fine-tuning) obtains comparable performance with the famous PCFR$^+$. Specifically, in Leduc Poker, Battleship (3), Goofspiel (5), RTPCFR$^+$ yields faster last-iterate convergence rates than PCFR$^+$. In Liar's Dice (4), Goofspiel (4), and Goofspiel (6), RTPCFR$^+$ obtains virtually identical convergence. But in the remaining 3 EFGs, PCFR$^+$ outper-

forms RTPCFR$^+$. **(4)** RTPCFR$^+$ demonstrates the fastest convergence performance in all EFGs when compared with algorithms that enjoy theoretical last-iterate convergence.

## 8. Conclusions

We explore the last-iterate convergence for predictive and discounted variants of CFR$^+$ algorithm in solving perturbed regularized EFGs. Under the reward transformation framework, we provide for the proposed RTPCFR$^+$ and RTPDCFR$^+$ algorithms the asymptotic last-iterate convergence, that holds for any initialized cumulative counterfactual regret and any (non-increasing) step sizes at each infoset. Such invariant last-iterate convergence enables us to set the output sequence-form strategy after multiple iterations as the initialization of next iteration for efficient implementation, and leads to a novel average regret bound for PCFR$^+$. Extensive experiments validate the effectiveness of our theoretical results, showing the promise of optimism/discounting techniques in EFGs under reward transformation framework.

## Impact Statement

This paper aims to advance the field of game theory by developing efficient CFR-type algorithms for solving extensive-form games with last-iterate convergence guarantees. There are potential social implications associated with our work, but none that we believe need to be particularly emphasized here.

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

## A. Additional Related Work

**Last-Iterate Convergence in EFGs.** We first describe the last-iterate convergence results in solving normal-form games (NFGs) / matrix games, which is a strict subset of EFGs. It is first discovered in (two-player) zero-sum matrix games that the vanilla online gradient descent (OGD) and multiplicative weight update (MWU) fail to converge last iterate (Mertikopoulos et al., 2018b, Thm 4.3) with continuous-time feedback, even using shrinking step sizes (Bailey & Piliouras, 2018, Thm 3.5). In contrast, their optimistic versions, optimistic gradient descent ascent (OGDA) and optimist multiplicative weight update (OMWU) (Daskalakis & Panageas, 2019; Mertikopoulos et al., 2018a; Lei et al., 2021), are proved to enjoy asymptotic last-iterate convergence. (Wei et al., 2021, Thm 3) further reveals that OMWU achieves linear last-iterate convergence in bilinear games over the simplex with the same unique NE assumption as in (Daskalakis & Panageas, 2019), and OGDA yields such linear convergence even without the uniqueness assumption (Wei et al., 2021, Thm 8). (Hsieh et al., 2021, Thm 7) shows the asymptotic last-iterate convergence of dual-stabilized optimistic OMD (DS-OOMD) with adaptive step size. (Cai et al., 2025, Thm 1) proves the last-iterate convergence of the famous regret matching$^+$ (RM$^+$) algorithm under the unique strict NE assumption, and their experimental counterexample shows the non-convergence of RM$^+$ under the unique non-strict NE condition. Besides, (Cai et al., 2025; Meng et al., 2025b) further prove the last-iterate convergence results for the smooth versions of RM$^+$, such as smooth predictive RM$^+$ in (Cai et al., 2025, Thm 5) and (Meng et al., 2025b, Thm 5.3). Extending the theoretical results (or algorithms) from NFG to EFG while maintaining efficient implementation is nontrivial, because directly doing so comes at the cost of an exponential blowup of the strategy space. (Lee et al., 2021, Thm 4) for the first time demonstrates that optimistic OMD (OOMD) that runs over the sequence-form strategy space (Von Stengel, 1996) of EFG achieve asymptotic last-iterate convergence. With additional unique NE assumption, vanilla OMWU and OGDA (both of which are special cases of OOMD by setting the Bregman divergence as KL-divergence and Euclidean distance, respectively) yield sublinear and linear last-convergence rates (Lee et al., 2021, Thm 6), respectively. Besides, running OOMD with a dilated entropy function is much more efficient (i.e having a closed-form solution at each decision point) and still maintains a linear last-iterate convergence rate, e.g. see (Lee et al., 2021, Thm 7) and (Farina et al., 2022, Thm 5.5).

**Counterfactual Regret Minimization (CFR) Algorithms in EFGs .** In extensive-form game, the core idea of CFR method is to decompose the (global) regret of each player as the summation of (local) counterfactual regrets at all information sets of the game tree (Zinkevich et al., 2007; Farina et al., 2019b). The next step is to utilize a no-regret online algorithm (Hazan et al., 2016; Shalev-Shwartz et al., 2012) to minimize the counterfactual regret at each information set. The first CFR algorithm (Zinkevich et al., 2007) uses regret matching (RM) (Hart & Mas-Colell, 2000) as the local regret minimizer, which is a parameter-free algorithm that outputs a distribution over actions in an infoset in proportion to the positive cumulative counterfactual regret on those actions. Subsequent works along this line mainly focus on developing new RM-type regret minimizer, such as RM$^+$ (Tammelin et al., 2015), discounted RM (Brown & Sandholm, 2019), predictive RM$^+$ (Farina et al., 2021b), smooth predictive RM$^+$ (Farina et al., 2023, Alg 2), conceptual RM$^+$ (Farina et al., 2023, Alg 3), predictive and discounted RM$^+$ (Xu et al., 2024), leading to CFR$^+$, DCFR, PCFR$^+$, smooth PCFR$^+$, conceptual smooth CFR$^+$, and PDCFR$^+$ algorithms, respectively. Most of these RM-based CFR algorithms enjoys an average-iterate convergence rate $O(1/\sqrt{T})$ in two-player zero-sum EFGs, except that smooth PCFR$^+$ (Farina et al., 2023, Thm 4.2) and conceptual smooth CFR$^+$ (Farina et al., 2023, Thm 5.5) achieving fast average-iterate convergence rate $O(1/T)$ (although conceptual smooth CFR$^+$, also called clairvoyant CFR$^+$, requires additional $O(T \log T)$ gradient computations to achieve this rate). Besides RM-type CFR algorithms, there also exist other CFR variants that take optimistic OMD as local regret minimizer, such as (Farina et al., 2019a) choosing OMWU and (Liu et al., 2023, Eq (5.2)) selecting dual-stabilized OOMD as local regret minimizers, and both of them achieve $O(1/T^{3/4})$ average-iterate convergence rate. We list the update rules and the average-iterate convergence rates of different CFR variants in Table 1 and Table 2 respectively, for the reader's reference.

**Other Average-Iterate Convergence Results in EFGs.** In contrast to CFR algorithms that use no-regret learning algorithm to minimize local counterfactual regrets in decision points and to output behavioral strategy, another group directly runs online algorithm over the whole treeplex to output sequence-form strategy and to minimize player's individual regret. (Farina et al., 2019c) directly runs optimistic OMD over treeplex and obtains $O(1/T)$ average-iterate convergence. (Farina et al., 2022) develop a kernelized OMWU algorithm that can be implemented efficiently with a per-iteration complexity linear in the number of sequences (i.e. information-action pairs) of the player, and can achieve $O(\log^4(T)/T)$ per-player regret in multiplayer general-sum game. (Bai et al., 2022, Thm 8) proves that running kernelized OMWU is equivalent to running OMD with dilated KL-divergence over the whole treeplex. (Fan et al., 2024, Thm 5.4) proposes new treeplex primal-dual norms and originally designs an optimistic OMD algorithm that is implemented (in theory but not in practice) over the terminal observation points in treeplex with per-player regret $O(1/\sqrt{T})$.

## B. Pseudocode for RTPCFR+ and RTPDCFR+

---

**Algorithm 1** RTCFR$^+$ with Optimism and Discounting for Perturbed Regularized EFGs

---

1: **Input:** $N, T_u, \mu, \gamma, \boldsymbol{r}, \alpha$ ($\alpha$ is only for RTPDCFR$^+$)
2: $\hat{\boldsymbol{\theta}}_I^1 \leftarrow \boldsymbol{0}, \eta \leftarrow 1, \forall I \in \mathcal{I}$
3: **for** each $n \in [1, 2, \cdots, N]$ **do**
4:    Build the perturbed regularized EFGs in (6) via $\mu, \gamma$, and $\boldsymbol{r}$
5:    **for** each $t \in [1, 2, \cdots, T_u]$ **do**
6:      **Option I (RTPCFR$^+$):** Update $\boldsymbol{\theta}_I^{t+1} = [\hat{\boldsymbol{\theta}}_I^t + \hat{\boldsymbol{m}}_i^t(I)]^+, \quad \hat{\boldsymbol{\theta}}_I^{t+1} = [\hat{\boldsymbol{\theta}}_I^t + \hat{\boldsymbol{m}}_i^{t+1}(I)]^+$
7:      **Option II (RTPDCFR$^+$):** Update $\boldsymbol{\theta}_I^{t+1} = [\frac{t^\alpha}{1+t^\alpha}\hat{\boldsymbol{\theta}}_I^t + \hat{\boldsymbol{m}}_i^t(I)]^+, \quad \hat{\boldsymbol{\theta}}_I^{t+1} = [\frac{t^\alpha}{1+t^\alpha}\hat{\boldsymbol{\theta}}_I^t + \hat{\boldsymbol{m}}_i^{t+1}(I)]^+$
8:      Output probability vector: $\sigma_i^{t+1}(I) = \boldsymbol{\theta}_I^{t+1}/\langle\boldsymbol{\theta}_I^{t+1}, \boldsymbol{1}\rangle$
9:      Output perturbed behavioral strategy: $\hat{\sigma}_i^{t+1}(I) = (1 - \alpha_I)\sigma_i^{t+1}(I) + \gamma\boldsymbol{1}$
10:      Update perturbed sequence-form strategy $\hat{\boldsymbol{x}}^{t+1}$ as : $\hat{\boldsymbol{x}}_i^{t+1}(I) = \pi_i^{\hat{\sigma}^{t+1}}(I)\hat{\sigma}_i^{t+1}(I), \forall I \in \mathcal{I}$
11:    **end for**
12:    Update hyperparameters: $\mu \leftarrow \mu * 0.3, \gamma \leftarrow \gamma * 0.5, \boldsymbol{r} \leftarrow \hat{\boldsymbol{x}}^{T_u+1}$
13:    Update initialized cumulative counterfactual regrets: $\hat{\boldsymbol{\theta}}_I^1 \leftarrow \hat{\boldsymbol{\theta}}_I^{T_u+1}, \forall I \in \mathcal{I}$
14: **end for**
15: **Return** $\hat{\boldsymbol{x}}^{T_u+1}$

---

## C. Update Rules and Average-Iterate Convergence Rates of differnet CFR Algorithms in EFGs

*Table 1.* Update rule of different CFR algorithms. $m_i^t(I)$ is the immediate counterfactual regret for player $i$ at infoset $I \in \mathcal{I}_i$.

| Algorithm | Cumulative Counterfactual Regret $\hat{\theta}_I^t$ | Behavioral Strategy $\sigma_I^{t+1}$ |
|:---:|:---:|:---:|
| CFR | $\hat{\boldsymbol{\theta}}_I^{t-1} + \boldsymbol{m}_i^t(I)$ | $[\hat{\boldsymbol{\theta}}_I^t]^+/\left\|[\hat{\boldsymbol{\theta}}_I^t]^+\right\|_1$ |
| CFR+ | $[\hat{\boldsymbol{\theta}}_I^{t-1} + \boldsymbol{m}_i^t(I)]^+$ | $\hat{\boldsymbol{\theta}}_I^t/\left\|\hat{\boldsymbol{\theta}}_I^t\right\|_1$ |
| Linear CFR | $\hat{\boldsymbol{\theta}}_I^{t-1} + t \cdot \boldsymbol{m}_i^t(I)$ | $[\hat{\boldsymbol{\theta}}_I^t]^+/\left\|[\hat{\boldsymbol{\theta}}_I^t]^+\right\|_1$ |
| DCFR | $\hat{\boldsymbol{\theta}}_I^t = \hat{\boldsymbol{\theta}}_I^{t-1} \odot \boldsymbol{d}_I^{t-1} + \boldsymbol{m}_i^t(I)$, where $\boldsymbol{d}_I^{t-1}[a] = \begin{cases} \dfrac{(t-1)^\alpha}{(t-1)^\alpha + 1} & \text{if } \hat{\boldsymbol{\theta}}_I^{t-1}[a] > 0 \\ \dfrac{(t-1)^\beta}{(t-1)^\beta + 1} & \text{otherwise} \end{cases}$ | $[\hat{\boldsymbol{\theta}}_I^t]^+/\left\|[\hat{\boldsymbol{\theta}}_I^t]^+\right\|_1$ |
| DCFR+ | $\hat{\boldsymbol{\theta}}_I^t = \left[\hat{\boldsymbol{\theta}}_I^{t-1}\dfrac{(t-1)^\alpha}{(t-1)^\alpha + 1} + \boldsymbol{m}_i^t(I)\right]^+$ | $\hat{\boldsymbol{\theta}}_I^t/\left\|\hat{\boldsymbol{\theta}}_I^t\right\|_1$ |
| PCFR+ | $[\hat{\boldsymbol{\theta}}_I^{t-1} + \boldsymbol{m}_i^t(I)]^+$ | $\boldsymbol{\theta}_I^{t+1}/\left\|\boldsymbol{\theta}_I^{t+1}\right\|_1$, where $\boldsymbol{\theta}_I^{t+1} = \left[\hat{\boldsymbol{\theta}}_I^t + \boldsymbol{m}_i^t(I)\right]^+$ |
| PDCFR+ | $\hat{\boldsymbol{\theta}}_I^t = \left[\hat{\boldsymbol{\theta}}_I^{t-1}\dfrac{(t-1)^\alpha}{(t-1)^\alpha + 1} + \boldsymbol{m}_i^t(I)\right]^+$ | $\boldsymbol{\theta}_I^{t+1}/\left\|\boldsymbol{\theta}_I^{t+1}\right\|_1$, where $\boldsymbol{\theta}_I^{t+1} = \left[\hat{\boldsymbol{\theta}}_I^t\dfrac{t^\alpha}{t^\alpha + 1} + \boldsymbol{m}_i^t(I)\right]^+$ |

*Table 2.* Comparisons of per-player individual regret (i.e. average-iterate convergence rate ) of different CFR variants for two-player zero-sum extensive-form games.

| Algorithm | Local Regret Minimizer | Per-Player Individual Regret |
|---|---|---|
| CFR
(Zinkevich et al., 2007, NeurIPS) | RM | $O(1/\sqrt{T})$ |
| CFR+
(Tammelin et al., 2015, IJCAI) | RM+ | $O(1/\sqrt{T})$ |
| DCFR
(Brown & Sandholm, 2019, AAAI) | Discounted RM | $O(1/\sqrt{T})$ |
| PCFR+
(Farina et al., 2021b, AAAI) | Predictive RM+ | $O\left(\sum_{I\in\mathcal{I}_i}\sqrt{\sum_{t=1}^{T}\|\hat{\boldsymbol{m}}_i^t(I)-\hat{\boldsymbol{m}}_i^{t-1}(I)\|_2^2}\,/\,T\right)$ |
| Opt FTRL
(Farina et al., 2019a, ICML) | Stable-Predictive OptMWU | $O(1/T^{3/4})$ |
| Reg-CFR
(Liu et al., 2023, ICLR) | DS-OptOMD | $O(1/T^{3/4})$ |
| Conceptual Smooth CFR+
(Farina et al., 2023, NeurIPS) | Conceptual RM+ | $O(1/T)$
Cost: $O(T\log T)$ gradient computations |
| PDCFR+
(Xu et al., 2024, IJCAI) | Predictive Discounted RM+ | $O\left(\sum_{I\in\mathcal{I}_i}\sqrt{\sum_{t=1}^{T}\|\hat{\boldsymbol{m}}_i^t(I)-\hat{\boldsymbol{m}}_i^{t-1}(I)\|_2^2}\,/\,T\right)$ |
| Invariant-PCFR+
our Theorem 5.3 | Predictive RM$^+$ | $O\left(\sum_{I\in\mathcal{I}_i}\sqrt{\frac{\|\hat{\boldsymbol{\theta}}_I^0\|_2^2}{\eta^2}+\sum_{t=1}^{T}\|\hat{\boldsymbol{m}}_i^t(I)-\hat{\boldsymbol{m}}_i^{t-1}(I)\|_2^2}\,/\,T\right)$ |

## D. Auxiliary Results

**Lemma D.1** (Lemma 4 in (Farina et al., 2021b)). *Let $\mathcal{D} \subseteq \mathbb{R}^d$ be a convex and closed set. Consider the update rule of online mirror descent (OMD) algorithm with step size $\eta > 0$ and convex distance generating function $\psi$: $\boldsymbol{\theta}^* \in \arg\min_{\boldsymbol{\theta} \in \mathcal{D}} \left\{ \langle \boldsymbol{g}, \boldsymbol{\theta} \rangle + \frac{1}{\eta} D_\psi(\boldsymbol{\theta} || \boldsymbol{\theta}^0) \right\}$, we have the following result for any $\boldsymbol{\theta} \in \mathcal{D}$:*

$$\eta \langle \boldsymbol{g}, \boldsymbol{\theta}^* - \boldsymbol{\theta} \rangle \leq D_\psi(\boldsymbol{\theta} || \boldsymbol{\theta}^0) - D_\psi(\boldsymbol{\theta} || \boldsymbol{\theta}^*) - D_\psi(\boldsymbol{\theta}^* || \boldsymbol{\theta}^0) \tag{11}$$

**Lemma D.2** (Adapted from Lemma D.4 in (Sokota et al., 2023)). *For any $\boldsymbol{x} \in \mathcal{X}$, $\mu \geq 0$, and $\gamma \geq 0$,*

$$\sum_{i \in \mathcal{N}} \langle \boldsymbol{\ell}_i^{\boldsymbol{x}}, \boldsymbol{x}_i - \boldsymbol{x}_i^{*,\mu,\gamma,\boldsymbol{r}} \rangle \geq \sum_{i \in \mathcal{N}} \langle \boldsymbol{\ell}_i^{\boldsymbol{x}} - \boldsymbol{\ell}_i^{\boldsymbol{x}^{*,\mu,\gamma,\boldsymbol{r}}}, \boldsymbol{x}_i - \boldsymbol{x}_i^{*,\mu,\gamma,\boldsymbol{r}} \rangle \geq \mu \| \boldsymbol{x} - \boldsymbol{x}^{*,\mu,\gamma,\boldsymbol{r}} \|_2^2, \tag{12}$$

*where $\boldsymbol{\ell}_0^{\boldsymbol{x}} = \boldsymbol{A}\boldsymbol{x}_1 + \mu\nabla\psi(\boldsymbol{x}_0) - \mu\nabla\psi(\boldsymbol{r}_0)$ and $\boldsymbol{\ell}_1^{\boldsymbol{x}} = -\boldsymbol{A}^T\boldsymbol{x}_0 + \mu\nabla\psi(\boldsymbol{x}_1) - \mu\nabla\psi(\boldsymbol{r}_1)$.*

**Lemma D.3** (Lemmas D.2-D.3 in (Meng et al., 2025a)). *For any $\boldsymbol{x}, \boldsymbol{x}' \in \mathcal{X}$, $i \in \mathcal{N}$, $I \in \mathcal{I}_i$, $\mu \geq 0$, and $\gamma \geq 0$,*

$$\|\hat{\boldsymbol{v}}_i^\sigma(I)\|_2 \leq \|\hat{\boldsymbol{v}}_i^\sigma(I)\|_1 \leq P + 2\mu D \tag{13}$$

$$\|\hat{\boldsymbol{v}}_i^\sigma(I) - \hat{\boldsymbol{v}}_i^{\sigma'}(I)\|_2 \leq (L + \mu)\|\boldsymbol{x} - \boldsymbol{x}'\|_1 + (P + 2\mu D)\|\sigma_i - \sigma_i'\|_1, \tag{14}$$

*where $\hat{\boldsymbol{v}}_i^\sigma(I) = [\hat{\boldsymbol{v}}_i^\sigma(I, a) | a \in A(I)]$, $\hat{\boldsymbol{v}}_i^\sigma(I, a) = -\hat{\boldsymbol{\ell}}_i^{\boldsymbol{x}} + \sum_{I' \in C_i(I,a)} \langle \hat{\boldsymbol{v}}_i^\sigma(I'), \sigma_i(I') \rangle$ with $\hat{\boldsymbol{\ell}}_0^{\boldsymbol{x}} = \boldsymbol{A}\boldsymbol{x}_1 + \mu\nabla\psi(\boldsymbol{x}_0) - \mu\nabla\psi(\boldsymbol{r}_0)$ and $\hat{\boldsymbol{\ell}}_1^{\boldsymbol{x}} = -\boldsymbol{A}^T\boldsymbol{x}_0 + \mu\nabla\psi(\boldsymbol{x}_1) - \mu\nabla\psi(\boldsymbol{r}_1)$, as well as $\sigma$ and $\sigma'$ are the behavioral strategy profiles associated with $\boldsymbol{x}$ and $\boldsymbol{x}'$, respectively.*

**Lemma D.4** (Lemma D.4 in (Meng et al., 2025a)). *For any $\hat{\boldsymbol{x}}, \hat{\boldsymbol{x}}' \in \mathcal{X}^\gamma$ with $\gamma > 0$, $i \in \mathcal{N}$, $I \in \mathcal{I}_i$, and $\mu \geq 0$,*

$$\|\hat{\sigma}_i - \hat{\sigma}_i'\|_1 \leq \frac{A_{max}C_{max} + 1}{\gamma^H}\|\hat{\boldsymbol{x}}_i - \hat{\boldsymbol{x}}_i'\|_1, \tag{15}$$

*where $\hat{\sigma}$ and $\hat{\sigma}'$ are the behavioral strategy profiles associated with $\hat{\boldsymbol{x}}$ and $\hat{\boldsymbol{x}}'$, respectively.*

**Lemma D.5** (Lemma 4.4 in (Meng et al., 2025a)). *$\forall i \in \mathcal{N}$, $I \in \mathcal{I}_i$, and $\boldsymbol{\theta}_I \in \mathbb{R}_{\geq 0}^{|A(I)|}$, $\langle -\hat{\boldsymbol{m}}_i^{*,\mu,\gamma,\boldsymbol{r}}(I), \boldsymbol{\theta}_I \rangle \geq 0$.*

The following lemma establishes the equivalence relation between individual regret $\langle \boldsymbol{\ell}_i, \boldsymbol{x}_i - \boldsymbol{x}_i' \rangle$ and weighted summation of counterfactual regrets $\{\boldsymbol{m}_i^\sigma(I)\}_{I \in \mathcal{I}_i}$. Such result is very essential and we give the whole proof details for the completeness of this work.

**Lemma D.6** (Lemma 2.2 in (Liu et al., 2022)). *For any $\boldsymbol{x}, \boldsymbol{x}' \in \mathcal{X}$, $\boldsymbol{\ell} \in \mathbb{R}^{|\mathcal{X}|}$, $i \in \mathcal{N}$, $\mu \geq 0$, and $\gamma \geq 0$, define*

$$\begin{aligned}
\boldsymbol{m}_i^\sigma(I) &= \boldsymbol{v}_i^\sigma(I) - \langle \boldsymbol{v}_i^\sigma(I), \sigma_i(I) \rangle \mathbf{1}, \\
\boldsymbol{v}_i^\sigma(I, a) &= -\boldsymbol{\ell}_i^\sigma(I, a) + \sum_{I' \in C_i(I,a)} \langle \boldsymbol{v}_i^\sigma(I'), \sigma_i(I') \rangle, \\
\boldsymbol{\ell}_0^\sigma &= \boldsymbol{A}\boldsymbol{x}_1^\sigma + \mu\nabla\psi(\boldsymbol{x}_0^\sigma) - \mu\nabla\psi(\boldsymbol{r}_0), \quad \boldsymbol{\ell}_1^\sigma = -\boldsymbol{A}^\top\boldsymbol{x}_0^\sigma + \mu\nabla\psi(\boldsymbol{x}_1^\sigma) - \mu\nabla\psi(\boldsymbol{r}_1).
\end{aligned} \tag{16}$$

*Then, we have the following result for any player $i \in \mathcal{N}$:*

$$\langle \boldsymbol{\ell}_i, \boldsymbol{x}_i - \boldsymbol{x}_i' \rangle = \sum_{I \in \mathcal{I}_i} \pi_i^{\sigma'}(I) \langle \boldsymbol{m}_i^\sigma(I), \sigma_i'(I) \rangle \tag{17}$$

*Proof.* Plug the definition of instantaneous counterfactual regret $\boldsymbol{m}_i^\sigma(I)$ into the right-hand side of the above equality,

$$\begin{aligned}
&\sum_{I \in \mathcal{I}_i} \pi_i^{\sigma'}(I) \langle \boldsymbol{m}_i^\sigma(I), \sigma_i'(I) \rangle \\
&= \sum_{I \in \mathcal{I}_i} \pi_i^{\sigma'}(I) \langle \boldsymbol{v}_i^\sigma(I) - \langle \boldsymbol{v}_i^\sigma(I), \sigma_i(I) \rangle \mathbf{1}, \sigma_i'(I) \rangle
\end{aligned}$$

$$= \sum_{I \in \mathcal{I}_i} \pi_i^{\sigma'}(I) \langle \boldsymbol{v}_i^{\sigma}(I), \sigma_i'(I) - \sigma_i(I) \rangle \tag{18}$$

$$\overset{(i)}{=} \sum_{I \in \mathcal{I}_i} \pi_i^{\sigma'}(I) \Big\langle -\boldsymbol{\ell}_i^{\sigma}(I) + \Big( \sum_{I' \in C_i(I,a)} \langle \boldsymbol{v}_i^{\sigma}(I'), \sigma_i(I') \rangle \Big)_{a \in A(I)}, \sigma_i'(I) \Big\rangle - \sum_{I \in \mathcal{I}_i} \pi_i^{\sigma'}(I) \langle \boldsymbol{v}_i^{\sigma}(I), \sigma_i(I) \rangle$$

$$= \sum_{I \in \mathcal{I}_i} \pi_i^{\sigma'}(I) \langle -\boldsymbol{\ell}_i^{\sigma}(I), \sigma_i'(I) \rangle + \sum_{I \in \mathcal{I}_i} \sum_{a \in A(I)} \sum_{I' \in C_i(I,a)} \pi_i^{\sigma'}(I) \sigma_i'(I,a) \langle \boldsymbol{v}_i^{\sigma}(I'), \sigma_i(I') \rangle - \sum_{I \in \mathcal{I}_i} \pi_i^{\sigma'}(I) \langle \boldsymbol{v}_i^{\sigma}(I), \sigma_i(I) \rangle$$

$$\overset{(ii)}{=} \sum_{I \in \mathcal{I}_i} \pi_i^{\sigma'}(I) \langle -\boldsymbol{\ell}_i^{\sigma}(I), \sigma_i'(I) \rangle + \sum_{I \in \mathcal{I}_i} \sum_{a \in A(I)} \sum_{I' \in C_i(I,a)} \pi_i^{\sigma'}(I') \langle \boldsymbol{v}_i^{\sigma}(I'), \sigma_i(I') \rangle - \sum_{I \in \mathcal{I}_i} \pi_i^{\sigma'}(I) \langle \boldsymbol{v}_i^{\sigma}(I), \sigma_i(I) \rangle$$

$$\overset{(iii)}{=} \sum_{I \in \mathcal{I}_i} \pi_i^{\sigma'}(I) \langle -\boldsymbol{\ell}_i^{\sigma}(I), \sigma_i'(I) \rangle + \sum_{I \in \mathcal{I}_i \backslash \{r\}} \pi_i^{\sigma'}(I) \langle \boldsymbol{v}_i^{\sigma}(I), \sigma_i(I) \rangle - \sum_{I \in \mathcal{I}_i} \pi_i^{\sigma'}(I) \langle \boldsymbol{v}_i^{\sigma}(I), \sigma_i(I) \rangle$$

$$= \sum_{I \in \mathcal{I}_i} \pi_i^{\sigma'}(I) \langle -\boldsymbol{\ell}_i^{\sigma}(I), \sigma_i'(I) \rangle - \sum_{I \in \{r\}} \pi_i^{\sigma'}(I) \langle \boldsymbol{v}_i^{\sigma}(I), \sigma_i(I) \rangle$$

$$\overset{(iv)}{=} \sum_{I \in \mathcal{I}_i} \pi_i^{\sigma'}(I) \langle -\boldsymbol{\ell}_i^{\sigma}(I), \sigma_i'(I) \rangle - \sum_{I \in \{r\}} \pi_i^{\sigma}(I) \langle \boldsymbol{v}_i^{\sigma}(I), \sigma_i(I) \rangle$$

$$\overset{(v)}{=} \sum_{I \in \mathcal{I}_i} \pi_i^{\sigma'}(I) \langle -\boldsymbol{\ell}_i^{\sigma}(I), \sigma_i'(I) \rangle - \sum_{I \in \{r\}} \Big[ \pi_i^{\sigma}(I) \langle -\boldsymbol{\ell}_i^{\sigma}(I) + \Big( \sum_{I' \in C_i(I,a)} \langle \boldsymbol{v}_i^{\sigma}(I'), \sigma_i(I') \rangle \Big)_{a \in A(I)}, \sigma_i(I) \rangle \Big]$$

$$= \sum_{I \in \mathcal{I}_i} \pi_i^{\sigma'}(I) \langle -\boldsymbol{\ell}_i^{\sigma}(I), \sigma_i'(I) \rangle - \sum_{I \in \{r\}} \Big[ \pi_i^{\sigma}(I) \langle -\boldsymbol{\ell}_i^{\sigma}(I), \sigma_i(I) \rangle + \sum_{a \in A(I)} \sum_{I' \in C_i(I,a)} \pi_i^{\sigma}(I) \langle \boldsymbol{v}_i^{\sigma}(I'), \sigma_i(I') \rangle \sigma_i(I,a) \Big]$$

$$\overset{(vi)}{=} \sum_{I \in \mathcal{I}_i} \pi_i^{\sigma'}(I) \langle -\boldsymbol{\ell}_i^{\sigma}(I), \sigma_i'(I) \rangle - \sum_{I \in \{r\}} \Big[ \pi_i^{\sigma}(I) \langle -\boldsymbol{\ell}_i^{\sigma}(I), \sigma_i(I) \rangle + \sum_{a \in A(I)} \sum_{I' \in C_i(I,a)} \pi_i^{\sigma}(I') \langle \boldsymbol{v}_i^{\sigma}(I'), \sigma_i(I') \rangle \Big]$$

$$\overset{(vii)}{=} \sum_{I \in \mathcal{I}_i} \pi_i^{\sigma'}(I) \langle -\boldsymbol{\ell}_i^{\sigma}(I), \sigma_i'(I) \rangle - \sum_{I \in \mathcal{I}_i} \pi_i^{\sigma}(I) \langle -\boldsymbol{\ell}_i^{\sigma}(I), \sigma_i(I) \rangle$$

$$\overset{(viii)}{=} \sum_{I \in \mathcal{I}_i} \langle -\boldsymbol{\ell}_i^{\sigma}(I), \boldsymbol{x}_i^{\sigma'}(I) \rangle - \sum_{I \in \mathcal{I}_i} \langle -\boldsymbol{\ell}_i^{\sigma}(I), \boldsymbol{x}_i^{\sigma}(I) \rangle$$

$$= \langle \boldsymbol{\ell}_i^{\sigma}, \boldsymbol{x}_i^{\sigma} - \boldsymbol{x}_i^{\sigma'} \rangle$$

where the equality $(i)$ and $(v)$ hold due to the definition of counterfactual value vector $\boldsymbol{v}_i^{\sigma}(I)$, $(ii)$ and $(vi)$ due to the fact that $\pi_i^{\sigma}(I)\sigma_i(I,a) = \pi_i^{\sigma}(I')$ for any $I' \in C_i(I,a)$ and any $\sigma$, $(iii)$ due to Lemma D.8 by setting $f(\sigma_i(I), \pi_i^{\sigma}(I)) = \pi_i^{\sigma}(I)\langle \boldsymbol{v}_i^{\sigma}(I), \sigma_i(I) \rangle$; $(iv)$ holds since in the root decision node $r \in \mathcal{I}_i$ we have $\pi_i^{\sigma}(r) = \pi_i^{\sigma'}(r) = 1$, $(vii)$ due to the recursion of expression $\pi_i^{\sigma}(I)\langle \boldsymbol{v}_i^{\sigma}(I), \sigma_i(I) \rangle$, $(viii)$ due to the fact that the equality $\pi_i^{\sigma}(I)\sigma_i(I) = \boldsymbol{x}_i^{\sigma}(I)$ holds for any $\sigma$ and information set $I$. $\qquad \square$

Applying Eq. (18) in the proof for Lemma D.6 obtains the following lemma.

**Corollary D.7** (Lemma 4.3 in (Meng et al., 2025a)). *For any $\boldsymbol{x}, \boldsymbol{x}' \in \mathcal{X}$, $\boldsymbol{\ell} \in \mathbb{R}^{|\mathcal{X}|}$, $i \in \mathcal{N}$, $\mu \geq 0$, and $\gamma \geq 0$,*

$$\langle \boldsymbol{\ell}_i, \boldsymbol{x}_i - \boldsymbol{x}_i' \rangle = \sum_{I \in \mathcal{I}_i} \pi_i^{\sigma'}(I) \langle -\boldsymbol{v}_i^{\sigma}(I,a), \sigma_i(I) - \sigma_i'(I) \rangle. \tag{19}$$

**Lemma D.8.** *For any behavioral strategy $\sigma$, any player $i \in \mathcal{N}$, any information set $I \in \mathcal{I}_i$, define any measurable function $f$ over the product space between the simplex space $\Delta^{|A(I)|}$ and unit internal $[0,1]$ as $f : \Delta^{|A(I)|} \times [0,1] \to \mathbb{R}$ as $f(\sigma_i(I), \pi_i^{\sigma}(I))$. Then we have*

$$\sum_{I \in \mathcal{I}_i} \sum_{a \in A(I)} \sum_{I' \in C_i(I,a)} f(\sigma_i(I'), \pi_i^{\sigma}(I')) = \sum_{I \in \mathcal{I}_i \backslash \{r\}} f(\sigma_i(I), \pi_i^{\sigma}(I)) \tag{20}$$

*Proof.* For any information set $I' \in \mathcal{I}_i \backslash \{r\}$, there exists only one information set $I \in \mathcal{I}_i$ and only one action $a \in A(I)$, such that $I' \in C_i(I,a)$, due to the perfect recall property of the extensive-form game. Besides, for any information set

$I$, and any action $a \in A(I)$, information set $I \notin C_i(I, a)$; and the root information set $r$ does not belong to any $C_i(I, a)$. Therefore, for any information set $I'(I' \notin \{r\})$, the term $f(\sigma_i(I'), \pi_i^\sigma(I'))$ in the left-hand side of Eq. (20) appears only once, thus the equality in Eq. (20) holds. The proof is finished. $\square$

*Remark D.9 (**Two Technical Challenges in our Refined Last-Iterate Convergence Analysis**). We explain the two main technical challenges in our last-iterate convergence analysis.*

1. *When demonstrating the asymptotic last-iterate convergence for RTPCFR$^+$: If we directly apply the first-order optimality condition to the first OMD update step in Eq. (8) (as what previous work does in their (Meng et al., 2025a, Sect D)), ultimately we can not get a telescoping sum. Instead, we choose to apply the first-order optimality condition to the second OMD update step in Eq. (8). Such choice can achieve a telescoping sum, but results in an extra term $-\eta\langle \hat{\boldsymbol{m}}_i^{t+1}(I), \boldsymbol{\theta}^{t+1} - \hat{\boldsymbol{\theta}}_I^t \rangle$. To deal with the extra term, we decompose it as $-\eta\langle \hat{\boldsymbol{m}}_i^{t+1}(I) - \hat{\boldsymbol{m}}_i^t(I), \boldsymbol{\theta}^{t+1} - \hat{\boldsymbol{\theta}}_I^t \rangle - \eta\langle \hat{\boldsymbol{m}}_i^t(I), \boldsymbol{\theta}^{t+1} - \hat{\boldsymbol{\theta}}_I^t \rangle$, and then bound the first term with Cauchy-Schwartz inequality and bound the second term with the first-order optimality condition of the first OMD update step in Eq. (8), finally leading to the term $\|\hat{\boldsymbol{m}}_i^{t+1}(I) - \hat{\boldsymbol{m}}_i^t(I)\|_2$ that needs to be upper bounded with the smoothness property of instantaneous counterfactual regret proposed in Lemma E.1.*

2. *When demonstrating the asymptotic last-iterate convergence for RTPDCFR$^+$, the main challenge lies in proving the invariance (i.e. for any non-increasing step sizes $\{\eta^t\}_{t\geq0}$) of last-iterate results. To solve this challenge, for any non-increasing step sizes $\{\eta^t\}_{t\geq0}$, we construct a novel sequence of step sizes $\eta^{t+1'} = \frac{\eta^{t'}}{\eta^t}\eta^{t+1}$ with $\eta^{0'} \leq \frac{\mu}{3|\mathcal{I}|\kappa^2}$, and prove the non-increasing property as well as the existence of the limit point of $\{\eta^{t+1'}\}_{t\geq0}$ by applying monotone convergence theorem (see Lemma G.2). The above analysis is critical to prove $\hat{\boldsymbol{\theta}}_I^{t'} = \frac{\eta^{t'}}{\eta^t}\hat{\boldsymbol{\theta}}_I^t$ by induction in Lemma G.3, finally resulting in the invariant last-iterate convergence for RTPDCFR$^+$.* $\square$

# E. Proof for the Asymptotic Last-Iterate Convergence of RTPCFR+ Algorithm

**Lemma E.1.** *Define the instantaneous regret associated with any behavioral strategy profile $\sigma = \{\sigma_1, \sigma_2\}$:*

$$
\begin{aligned}
\hat{\boldsymbol{m}}_i^\sigma(I) &= \hat{\boldsymbol{v}}_i^\sigma(I) - \langle \hat{\boldsymbol{v}}_i^\sigma(I), \sigma_i(I)\rangle \mathbf{1}, \\
\hat{\boldsymbol{v}}_i^\sigma(I, a) &= -\hat{\boldsymbol{\ell}}_i^\sigma(I, a) + \sum_{I' \in C_i(I, a)} \langle \hat{\boldsymbol{v}}_i^\sigma(I'), \hat{\sigma}_i(I)\rangle, \\
\hat{\boldsymbol{\ell}}_0^\sigma &= \boldsymbol{A}\hat{\boldsymbol{x}}_1^\sigma + \mu\nabla\psi(\hat{\boldsymbol{x}}_0^\sigma) - \mu\nabla\psi(\boldsymbol{r}_0), \quad \hat{\boldsymbol{\ell}}_1^\sigma = -\boldsymbol{A}^\top\hat{\boldsymbol{x}}_0^\sigma + \mu\nabla\psi(\hat{\boldsymbol{x}}_1^\sigma) - \mu\nabla\psi(\boldsymbol{r}_1).
\end{aligned}
\tag{21}
$$

*Let $\kappa = \left[(A_{max} + 1)(L + \mu) + (2A_{max} + 1)\frac{(P + 2\mu D)}{1 - \gamma A_{max}}\frac{A_{max}C_{max} + 1}{\gamma^H}\right]\sqrt{|\mathcal{I}|A_{max}}$ Then for any strategy profile $\sigma, \sigma'$, any player $i \in \mathcal{N}$, any information set $I \in \mathcal{I}_i$, we have:*

$$
\|\hat{\boldsymbol{m}}_i^\sigma(I) - \hat{\boldsymbol{m}}_i^{\sigma'}(I)\|_2 \leq \kappa\|\hat{\boldsymbol{x}}^\sigma - \hat{\boldsymbol{x}}^{\sigma'}\|_2
\tag{22}
$$

*Proof.*

$$
\begin{aligned}
\|\hat{\boldsymbol{m}}_i^\sigma(I) - \hat{\boldsymbol{m}}_i^{\sigma'}(I)\|_2 &= \|\hat{\boldsymbol{v}}_i^\sigma(I) - \langle \hat{\boldsymbol{v}}_i^\sigma(I), \sigma_i(I)\rangle\mathbf{1} - \hat{\boldsymbol{v}}_i^{\sigma'}(I) + \langle \hat{\boldsymbol{v}}_i^{\sigma'}(I), \sigma_i'(I)\rangle\mathbf{1}\|_2 \\
&\overset{(i)}{\leq} \|\hat{\boldsymbol{v}}_i^\sigma(I) - \hat{\boldsymbol{v}}_i^{\sigma'}(I)\|_2 + \left|\langle \hat{\boldsymbol{v}}_i^\sigma(I), \sigma_i(I)\rangle - \langle \hat{\boldsymbol{v}}_i^{\sigma'}(I), \sigma_i'(I)\rangle\right|\|\mathbf{1}\|_2 \\
&\overset{(ii)}{\leq} \|\hat{\boldsymbol{v}}_i^\sigma(I) - \hat{\boldsymbol{v}}_i^{\sigma'}(I)\|_2 + \left|\langle \hat{\boldsymbol{v}}_i^\sigma(I) - \hat{\boldsymbol{v}}_i^{\sigma'}(I), \sigma_i(I)\rangle + \langle \hat{\boldsymbol{v}}_i^{\sigma'}(I), \sigma_i(I) - \sigma_i'(I)\rangle\right|\sqrt{|A(I)|} \\
&\overset{(iii)}{\leq} \|\hat{\boldsymbol{v}}_i^\sigma(I) - \hat{\boldsymbol{v}}_i^{\sigma'}(I)\|_2 + \sqrt{|A(I)|}\left(\|\hat{\boldsymbol{v}}_i^\sigma(I) - \hat{\boldsymbol{v}}_i^{\sigma'}(I)\|_2\|\sigma_i(I)\|_2 + \|\hat{\boldsymbol{v}}_i^{\sigma'}(I)\|_2\|\sigma_i(I) - \sigma_i'(I)\|_2\right) \\
&\overset{(iv)}{\leq} \|\hat{\boldsymbol{v}}_i^\sigma(I) - \hat{\boldsymbol{v}}_i^{\sigma'}(I)\|_2 + \sqrt{|A(I)|}\left(\|\hat{\boldsymbol{v}}_i^\sigma(I) - \hat{\boldsymbol{v}}_i^{\sigma'}(I)\|_2 + (P + 2\mu D)\|\sigma_i(I) - \sigma_i'(I)\|_2\right) \\
&= (|A(I)| + 1)\|\hat{\boldsymbol{v}}_i^\sigma(I) - \hat{\boldsymbol{v}}_i^{\sigma'}(I)\|_2 + \sqrt{|A(I)|}(P + 2\mu D)\|\sigma_i(I) - \sigma_i'(I)\|_2 \\
&\overset{(v)}{\leq} (|A(I)| + 1)\left[(L + \mu)\|\hat{\boldsymbol{x}}^\sigma - \hat{\boldsymbol{x}}^{\sigma'}\|_1 + (P + 2\mu D)\|\sigma_i - \sigma_i'\|_1\right] + \sqrt{|A(I)|}(P + 2\mu D)\|\sigma_i(I) - \sigma_i'(I)\|_2
\end{aligned}
\tag{23}
$$

$$=\left(|A(I)|+1\right)\left[(L+\mu)\|\hat{\boldsymbol{x}}^\sigma-\hat{\boldsymbol{x}}^{\sigma'}\|_1+\frac{(P+2\mu D)}{1-\alpha_I}\|\hat{\sigma}_i-\hat{\sigma}'_i\|_1\right]+\frac{|A(I)|(P+2\mu D)}{1-\alpha_I}\|\hat{\sigma}_i(I)-\hat{\sigma}'_i(I)\|_2$$

$$\leq\left(|A(I)|+1\right)\left[(L+\mu)\|\hat{\boldsymbol{x}}^\sigma-\hat{\boldsymbol{x}}^{\sigma'}\|_1+\frac{(P+2\mu D)}{1-\alpha_I}\|\hat{\sigma}_i-\hat{\sigma}'_i\|_1\right]+\frac{|A(I)|(P+2\mu D)}{1-\alpha_I}\|\hat{\sigma}_i-\hat{\sigma}'_i\|_1$$

$$=\left(|A(I)|+1\right)(L+\mu)\|\hat{\boldsymbol{x}}^\sigma-\hat{\boldsymbol{x}}^{\sigma'}\|_1+\left(2|A(I)|+1\right)\frac{(P+2\mu D)}{1-\alpha_I}\|\hat{\sigma}_i-\hat{\sigma}'_i\|_1$$

$$\overset{(vi)}{\leq}\left(|A(I)|+1\right)(L+\mu)\|\hat{\boldsymbol{x}}^\sigma-\hat{\boldsymbol{x}}^{\sigma'}\|_1+\left(2|A(I)|+1\right)\frac{(P+2\mu D)}{1-\alpha_I}\frac{A_{max}C_{max}+1}{\gamma^H}\|\hat{\boldsymbol{x}}_i^\sigma-\hat{\boldsymbol{x}}_i^{\sigma'}\|_1$$

$$\leq\left[\left(|A(I)|+1\right)(L+\mu)+\left(2|A(I)|+1\right)\frac{(P+2\mu D)}{1-\alpha_I}\frac{A_{max}C_{max}+1}{\gamma^H}\right]\|\hat{\boldsymbol{x}}^\sigma-\hat{\boldsymbol{x}}^{\sigma'}\|_1$$

$$\leq\kappa\|\hat{\boldsymbol{x}}^\sigma-\hat{\boldsymbol{x}}^{\sigma'}\|_2$$

where $(i)$ holds due to the triangle inequality, $(ii)$ due to the fact $\|\mathbf{1}\|_2\leq\|\mathbf{1}\|_1\leq|A(I)|$, $(iii)$ due to Cauchy-Schwartz inequality $\langle\boldsymbol{a},\boldsymbol{b}\rangle\leq\|\boldsymbol{a}\|_2\|\boldsymbol{b}\|_2$, $(iv)$ due to the boundedness of counterfactual value $\hat{\boldsymbol{v}}_i^{\sigma'}(I)$ in Eq. (13) and the boundedness of $\ell_1$-norm of a simplex vector $\sigma_i(I)$, $(v)$ due to Lemma D.3, $(vi)$ due to Lemma D.4. The last inequality holds since for any $d$-dimensional vector $\boldsymbol{x}$, we have $\|\boldsymbol{x}\|_1\leq\sqrt{d}\|\boldsymbol{x}\|_2$. $\qquad\square$

*Remark* E.2 (**Three main differences between our last-iterate convergence analysis and that of the most related work**). *Our proposed RTPCFR$^+$ algorithm can be considered as the generalization of RTCFR$^+$ algorithm (Meng et al., 2025a), and our proof for the last-iterate convergence of RTPCFR$^+$ has three main differences from that of (Meng et al., 2025a, Thm 4.1).*

1. *The first difference is that we aim to bound $\eta\langle-\hat{\boldsymbol{m}}_i^{t+1}(I),\hat{\boldsymbol{\theta}}_I^{t+1}-\boldsymbol{\theta}\rangle$ in the second update step in optimistic OMD, while (Meng et al., 2025a, Eq (7)) chooses to bound $\eta\langle\hat{\boldsymbol{m}}_i^t(I),\boldsymbol{\theta}-\boldsymbol{\theta}_I^{t+1}\rangle$ in non-optimistic OMD. Such difference leads to an additional term in our upper bound $-\eta\langle\hat{\boldsymbol{m}}_i^{t+1}(I),\boldsymbol{\theta}^{t+1}-\hat{\boldsymbol{\theta}}_I^t\rangle$ (i.e. term ④ in Eq. (28)). To bound this term, we decompose it as $-\eta\langle\hat{\boldsymbol{m}}_i^{t+1}(I),\boldsymbol{\theta}^{t+1}-\hat{\boldsymbol{\theta}}_I^t\rangle=\underbrace{-\eta\langle\hat{\boldsymbol{m}}_i^{t+1}(I)-\hat{\boldsymbol{m}}_i^t(I),\boldsymbol{\theta}^{t+1}-\hat{\boldsymbol{\theta}}_I^t\rangle}_{(i)}\underbrace{-\eta\langle\hat{\boldsymbol{m}}_i^t(I),\boldsymbol{\theta}^{t+1}-\hat{\boldsymbol{\theta}}_I^t\rangle}_{(ii)}$, and use the first-order optimality condition of the first update step in OOMD to bound $(ii)$: $-\eta\langle\hat{\boldsymbol{m}}_i^t(I),\boldsymbol{\theta}^{t+1}-\hat{\boldsymbol{\theta}}_I^t\rangle$. Such bounding is not that technical, but highlights the key difference between the update rules of optimist OMD and that of vanilla OMD.*

2. *The second difference lies the upper bound on $(i)$ in the first step, and leads to an extra term $\|\hat{\boldsymbol{m}}_i^{t+1}(I)-\hat{\boldsymbol{m}}_i^t(I)\|_2^2$, and we upper bound it with our Lemma E.1. Such bounding process does not exist in the proof for RTCFR$^+$.*

3. *The third main difference lies in the proof for the invariance of RTPCFR$^+$ algorithm to any step size $\eta>0$. We first need to use induction to prove that $\hat{\boldsymbol{\theta}}_I^{t+1'}=\frac{\eta'}{\eta}\hat{\boldsymbol{\theta}}_I^{t+1}$ for any $t$, and then this result to prove $\boldsymbol{\theta}_I^{t+1'}=\frac{\eta'}{\eta}\boldsymbol{\theta}_I^{t+1}$, which is critical to the step size invariance of RTPCFR$^+$. In contrast, (Meng et al., 2025a, Sect D) directly demonstrates that $\boldsymbol{\theta}_I^{t'}=\frac{\eta'}{\eta}\boldsymbol{\theta}_I^t$.* $\qquad\square$

*Proof of Theorem 5.1.* The update rule of optimistic OMD is as follow:

$$\boldsymbol{\theta}_I^{t+1}\in\arg\min_{\boldsymbol{\theta}_I\in\mathbb{R}_{\geq0}^{|A(I)|}}\left\{\langle-\hat{\boldsymbol{m}}_i^t(I),\boldsymbol{\theta}_I\rangle+\frac{1}{\eta}D_\psi(\boldsymbol{\theta}_I\|\hat{\boldsymbol{\theta}}_I^t)\right\},\tag{24}$$

$$\hat{\boldsymbol{\theta}}_I^{t+1}\in\arg\min_{\boldsymbol{\theta}_I\in\mathbb{R}_{\geq0}^{|A(I)|}}\left\{\langle-\hat{\boldsymbol{m}}_i^{t+1}(I),\boldsymbol{\theta}_I\rangle+\frac{1}{\eta}D_\psi(\boldsymbol{\theta}_I\|\hat{\boldsymbol{\theta}}_I^t)\right\}.\tag{25}$$

We apply Lemma D.1 for the update rule of optimistic OMD in Eq. (25), obtaining that for any $\boldsymbol{\theta}\in\mathbb{R}_{\geq0}^{|A(I)|}$:

$$\eta\langle-\hat{\boldsymbol{m}}_i^{t+1}(I),\hat{\boldsymbol{\theta}}_I^{t+1}-\boldsymbol{\theta}\rangle\leq D_\psi(\boldsymbol{\theta}\|\hat{\boldsymbol{\theta}}_I^t)-D_\psi(\boldsymbol{\theta}\|\hat{\boldsymbol{\theta}}_I^{t+1})-D_\psi(\hat{\boldsymbol{\theta}}_I^{t+1}\|\hat{\boldsymbol{\theta}}_I^t).\tag{26}$$

Setting $\boldsymbol{\theta}=\sigma_i^{*,\gamma}(I)$ in the above inequality, we have

$$\eta\langle\hat{\boldsymbol{m}}_i^{t+1}(I),\sigma_i^{*,\gamma}(I)-\hat{\boldsymbol{\theta}}_I^{t+1}\rangle\leq D_\psi(\sigma_i^{*,\gamma}(I)\|\hat{\boldsymbol{\theta}}_I^t)-D_\psi(\sigma_i^{*,\gamma}(I)\|\hat{\boldsymbol{\theta}}_I^{t+1})-D_\psi(\hat{\boldsymbol{\theta}}_I^{t+1}\|\hat{\boldsymbol{\theta}}_I^t).\tag{27}$$

Adding $\eta\langle \hat{\boldsymbol{m}}_i^{t+1}(I), -\boldsymbol{\theta}_I^{t+1}\rangle$ into both sides of Eq. (27), and rearranging the terms, we have

$$
\begin{aligned}
&\eta\langle \hat{\boldsymbol{m}}_i^{t+1}(I), \sigma_i^{*,\gamma}(I) - \boldsymbol{\theta}_I^{t+1}\rangle\\
\leq & D_\psi(\sigma_i^{*,\gamma}(I)||\hat{\boldsymbol{\theta}}_I^t) - D_\psi(\sigma_i^{*,\gamma}(I)||\hat{\boldsymbol{\theta}}_I^{t+1}) - D_\psi(\hat{\boldsymbol{\theta}}_I^{t+1}||\hat{\boldsymbol{\theta}}_I^t) - \eta\langle \hat{\boldsymbol{m}}_i^{t+1}(I), \boldsymbol{\theta}_I^{t+1} - \hat{\boldsymbol{\theta}}_I^{t+1}\rangle\\
= & \underbrace{D_\psi(\sigma_i^{*,\gamma}(I)||\hat{\boldsymbol{\theta}}_I^t) - D_\psi(\sigma_i^{*,\gamma}(I)||\hat{\boldsymbol{\theta}}_I^{t+1})}_{①} \underbrace{- D_\psi(\hat{\boldsymbol{\theta}}_I^{t+1}||\hat{\boldsymbol{\theta}}_I^t) - \eta\langle \hat{\boldsymbol{m}}_i^{t+1}(I) - \hat{\boldsymbol{m}}_i^{*,\gamma}(I), \hat{\boldsymbol{\theta}}_I^t - \hat{\boldsymbol{\theta}}_I^{t+1}\rangle}_{②}\\
& \underbrace{- \eta\langle \hat{\boldsymbol{m}}_i^{*,\gamma}(I), \hat{\boldsymbol{\theta}}_I^t - \hat{\boldsymbol{\theta}}_I^{t+1}\rangle}_{③} \underbrace{- \eta\langle \hat{\boldsymbol{m}}_i^{t+1}(I), \boldsymbol{\theta}^{t+1} - \hat{\boldsymbol{\theta}}_I^t\rangle}_{④}.
\end{aligned}
\tag{28}
$$

It is not difficult to see that, when taking a summation of the right-hand-side of Eq. (28) over $t \in [T]$, both term ① and term ③ can lead to the telescoping sum. Therefore, it remains to bound terms ② and ④ respectively.

**(I) Bounding ② in Eq. (28)**

$$
\begin{aligned}
& - D_\psi(\hat{\boldsymbol{\theta}}_I^{t+1}||\hat{\boldsymbol{\theta}}_I^t) - \eta\langle \hat{\boldsymbol{m}}_i^{t+1}(I) - \hat{\boldsymbol{m}}_i^{*,\gamma}(I), \hat{\boldsymbol{\theta}}_I^t - \hat{\boldsymbol{\theta}}_I^{t+1}\rangle\\
\leq & - D_\psi(\hat{\boldsymbol{\theta}}_I^{t+1}||\hat{\boldsymbol{\theta}}_I^t) + \eta\Big[\|\hat{\boldsymbol{m}}_i^{t+1}(I) - \hat{\boldsymbol{m}}_i^{*,\gamma}(I)\|_2 \cdot \|\hat{\boldsymbol{\theta}}_I^t - \hat{\boldsymbol{\theta}}_I^{t+1}\|_2\Big]\\
\leq & - D_\psi(\hat{\boldsymbol{\theta}}_I^{t+1}||\hat{\boldsymbol{\theta}}_I^t) + \eta\Big[\frac{\eta\|\hat{\boldsymbol{m}}_i^{t+1}(I) - \hat{\boldsymbol{m}}_i^{*,\gamma}(I)\|_2^2}{2} + \frac{\|\hat{\boldsymbol{\theta}}_I^t - \hat{\boldsymbol{\theta}}_I^{t+1}\|_2^2}{2\eta}\Big]\\
= & \frac{\eta^2}{2}\|\hat{\boldsymbol{m}}_i^{t+1}(I) - \hat{\boldsymbol{m}}_i^{*,\gamma}(I)\|_2^2,
\end{aligned}
\tag{29}
$$

where the first and the second inequalities in Eq. (29) are due to Cauchy-Schwartz and mean-value inequality $ab \leq \frac{\eta a^2}{2} + \frac{b^2}{2\eta}$ for any $\eta > 0$, respectively.

**(II) Bounding ④ in Eq. (28)**

$$
\begin{aligned}
& - \eta\langle \hat{\boldsymbol{m}}_i^{t+1}(I), \boldsymbol{\theta}^{t+1} - \hat{\boldsymbol{\theta}}_I^t\rangle\\
= & - \eta\langle \hat{\boldsymbol{m}}_i^{t+1}(I) - \hat{\boldsymbol{m}}_i^t(I), \boldsymbol{\theta}^{t+1} - \hat{\boldsymbol{\theta}}_I^t\rangle - \eta\langle \hat{\boldsymbol{m}}_i^t(I), \boldsymbol{\theta}^{t+1} - \hat{\boldsymbol{\theta}}_I^t\rangle\\
\overset{(i)}{\leq} & \eta\Big[\frac{\eta}{4}\|\hat{\boldsymbol{m}}_i^{t+1}(I) - \hat{\boldsymbol{m}}_i^t(I)\|_2^2 + \frac{1}{\eta}\|\boldsymbol{\theta}^{t+1} - \hat{\boldsymbol{\theta}}_I^t\|_2^2\Big] + \Big[D_\psi(\hat{\boldsymbol{\theta}}_I^t||\hat{\boldsymbol{\theta}}_I^t) - D_\psi(\hat{\boldsymbol{\theta}}_I^t||\boldsymbol{\theta}_I^{t+1}) - D_\psi(\boldsymbol{\theta}_I^{t+1}||\hat{\boldsymbol{\theta}}_I^t)\Big]\\
= & \eta\Big[\frac{\eta}{4}\|\hat{\boldsymbol{m}}_i^{t+1}(I) - \hat{\boldsymbol{m}}_i^t(I)\|_2^2 + \frac{1}{\eta}\|\boldsymbol{\theta}^{t+1} - \hat{\boldsymbol{\theta}}_I^t\|_2^2\Big] + \Big[-\|\boldsymbol{\theta}^{t+1} - \hat{\boldsymbol{\theta}}_I^t\|_2^2\Big]\\
= & \frac{\eta^2}{4}\|\hat{\boldsymbol{m}}_i^{t+1}(I) - \hat{\boldsymbol{m}}_i^t(I)\|_2^2,
\end{aligned}
\tag{30}
$$

where in the inequality $(i)$ of Eq. (30), the first part holds due to Cauchy-Schwartz/mean-value inequality (cf. Eq. (29)), and the second part due to the application of Lemma D.1 to the OMD update in Eq. (24).

Combining the results in Eq. (29) and Eq. (30), and plugging them into Eq. (28), we have

$$
\begin{aligned}
& \eta\langle \hat{\boldsymbol{m}}_i^{t+1}(I), \sigma_i^{*,\gamma}(I) - \boldsymbol{\theta}_I^{t+1}\rangle - \frac{\eta^2}{2}\|\hat{\boldsymbol{m}}_i^{t+1}(I) - \hat{\boldsymbol{m}}_i^{*,\gamma}(I)\|_2^2 - \frac{\eta^2}{4}\|\hat{\boldsymbol{m}}_i^{t+1}(I) - \hat{\boldsymbol{m}}_i^t(I)\|_2^2\\
\leq & D_\psi(\sigma_i^{*,\gamma}(I)||\hat{\boldsymbol{\theta}}_I^t) - D_\psi(\sigma_i^{*,\gamma}(I)||\hat{\boldsymbol{\theta}}_I^{t+1}) - \eta\langle \hat{\boldsymbol{m}}_i^{*,\gamma}(I), \hat{\boldsymbol{\theta}}_I^t - \hat{\boldsymbol{\theta}}_I^{t+1}\rangle.
\end{aligned}
\tag{31}
$$

Notice that $\langle \hat{\boldsymbol{m}}_i^{t+1}(I), \boldsymbol{\theta}_I^{t+1}\rangle = \langle \hat{\boldsymbol{m}}_i^{t+1}(I), \sigma_i^{t+1}\rangle\|\boldsymbol{\theta}_I^{t+1}\|_1 = \langle \hat{\boldsymbol{v}}_i^{t+1}(I) - \langle \hat{\boldsymbol{v}}_i^{t+1}(I), \sigma_i^{t+1}(I)\rangle\mathbf{1}, \sigma_i^{t+1}\rangle\|\boldsymbol{\theta}_I^{t+1}\|_1 = 0$, then the first term in the left-hand side of Eq. (31) can be simplified as

$$
\begin{aligned}
\langle \hat{\boldsymbol{m}}_i^{t+1}(I), \sigma_i^{*,\gamma}(I) - \boldsymbol{\theta}_I^{t+1}\rangle & = \langle \hat{\boldsymbol{v}}_i^{t+1}(I) - \langle \hat{\boldsymbol{v}}_i^{t+1}(I), \sigma_i^{t+1}(I)\rangle\mathbf{1}, \sigma_i^{*,\gamma}(I)\rangle\\
& = \langle \hat{\boldsymbol{v}}_i^{t+1}(I), \sigma_i^{*,\gamma}(I) - \sigma_i^{t+1}(I)\rangle\\
& = \frac{1}{1 - \alpha_I}\langle \hat{\boldsymbol{v}}_i^{t+1}(I), \hat{\sigma}_i^{*,\gamma}(I) - \hat{\sigma}_i^{t+1}(I)\rangle
\end{aligned}
\tag{32}
$$

Then, let $\beta_I = \pi_i^{\hat{\sigma}^{*,\gamma}}(I)$, $\zeta_I = \beta_I(1 - \alpha_I)$ (obviously $\zeta_I < 1$), multiplying both sides in Eq. (31) by $\zeta_I$, we have

$$
\begin{aligned}
&\eta\beta_I\langle -\hat{\boldsymbol{v}}_i^{t+1}(I), \hat{\sigma}_i^{t+1}(I) - \hat{\sigma}_i^{*,\gamma}(I)\rangle - \frac{\eta^2\zeta_I}{2}\|\hat{\boldsymbol{m}}_i^{t+1}(I) - \hat{\boldsymbol{m}}_i^{*,\gamma}(I)\|_2^2 - \frac{\eta^2\zeta_I}{4}\|\hat{\boldsymbol{m}}_i^{t+1}(I) - \hat{\boldsymbol{m}}_i^t(I)\|_2^2 \\
&\leq \zeta_I\Big[D_\psi(\sigma_i^{*,\gamma}(I)\|\hat{\boldsymbol{\theta}}_I^t) - D_\psi(\sigma_i^{*,\gamma}(I)\|\hat{\boldsymbol{\theta}}_I^{t+1}) - \eta\langle \hat{\boldsymbol{m}}_i^{*,\gamma}(I), \hat{\boldsymbol{\theta}}_I^t - \hat{\boldsymbol{\theta}}_I^{t+1}\rangle\Big].
\end{aligned}
\tag{33}
$$

Taking summation of both sides in Eq. (33) over $t = 0, 1, \cdots, T-1$, $i \in \mathcal{N}$, $I \in \mathcal{I}_i$, we have

$$
\begin{aligned}
&\sum_{i\in\mathcal{N}}\sum_{I\in\mathcal{I}_i}\zeta_I\Big[D_\psi(\sigma_i^{*,\gamma}(I)\|\hat{\boldsymbol{\theta}}_I^0) - \eta\langle \hat{\boldsymbol{m}}_i^{*,\gamma}(I), \hat{\boldsymbol{\theta}}_I^0\rangle\Big] \\
&\stackrel{(i)}{\geq} \sum_{i\in\mathcal{N}}\sum_{I\in\mathcal{I}_i}\zeta_I\Big[D_\psi(\sigma_i^{*,\gamma}(I)\|\hat{\boldsymbol{\theta}}_I^0) - D_\psi(\sigma_i^{*,\gamma}(I)\|\hat{\boldsymbol{\theta}}_I^T) - \eta\langle \hat{\boldsymbol{m}}_i^{*,\gamma}(I), \hat{\boldsymbol{\theta}}_I^0 - \hat{\boldsymbol{\theta}}_I^T\rangle\Big] \\
&\geq \sum_{t=0}^{T-1}\sum_{i\in\mathcal{N}}\sum_{I\in\mathcal{I}_i}\Big[\eta\beta_I\langle -\hat{\boldsymbol{v}}_i^{t+1}(I), \hat{\sigma}_i^{t+1}(I) - \hat{\sigma}_i^{*,\gamma}(I)\rangle - \frac{\eta^2\zeta_I}{2}\|\hat{\boldsymbol{m}}_i^{t+1}(I) - \hat{\boldsymbol{m}}_i^{*,\gamma}(I)\|_2^2 - \frac{\eta^2\zeta_I}{4}\|\hat{\boldsymbol{m}}_i^{t+1}(I) - \hat{\boldsymbol{m}}_i^t(I)\|_2^2\Big] \\
&\geq \sum_{t=0}^{T-1}\sum_{i\in\mathcal{N}}\sum_{I\in\mathcal{I}_i}\Big[\eta\beta_I\langle -\hat{\boldsymbol{v}}_i^{t+1}(I), \hat{\sigma}_i^{t+1}(I) - \hat{\sigma}_i^{*,\gamma}(I)\rangle - \frac{\eta^2}{2}\|\hat{\boldsymbol{m}}_i^{t+1}(I) - \hat{\boldsymbol{m}}_i^{*,\gamma}(I)\|_2^2 - \frac{\eta^2}{4}\|\hat{\boldsymbol{m}}_i^{t+1}(I) - \hat{\boldsymbol{m}}_i^t(I)\|_2^2\Big] \\
&\stackrel{(ii)}{=} \sum_{t=0}^{T-1}\sum_{i\in\mathcal{N}}\Big[\eta\langle \hat{\boldsymbol{\ell}}_i^{t+1}, \hat{\boldsymbol{x}}_i^{t+1} - \hat{\boldsymbol{x}}_i^{*,\gamma}\rangle - \sum_{I\in\mathcal{I}_i}\frac{\eta^2}{2}\|\hat{\boldsymbol{m}}_i^{t+1}(I) - \hat{\boldsymbol{m}}_i^{*,\gamma}(I)\|_2^2 - \sum_{I\in\mathcal{I}_i}\frac{\eta^2}{4}\|\hat{\boldsymbol{m}}_i^{t+1}(I) - \hat{\boldsymbol{m}}_i^t(I)\|_2^2\Big] \\
&\stackrel{(iii)}{\geq} \sum_{t=0}^{T-1}\sum_{i\in\mathcal{N}}\Big[\eta\mu\|\hat{\boldsymbol{x}}_i^{t+1} - \hat{\boldsymbol{x}}_i^{*,\gamma}\|_2^2 - \sum_{I\in\mathcal{I}_i}\frac{\eta^2}{2}\|\hat{\boldsymbol{m}}_i^{t+1}(I) - \hat{\boldsymbol{m}}_i^{*,\gamma}(I)\|_2^2 - \sum_{I\in\mathcal{I}_i}\frac{\eta^2}{4}\|\hat{\boldsymbol{m}}_i^{t+1}(I) - \hat{\boldsymbol{m}}_i^t(I)\|_2^2\Big] \\
&\stackrel{(iv)}{\geq} \sum_{t=0}^{T-1}\sum_{i\in\mathcal{N}}\Big[\eta\mu\|\hat{\boldsymbol{x}}_i^{t+1} - \hat{\boldsymbol{x}}_i^{*,\gamma}\|_2^2 - \sum_{I\in\mathcal{I}_i}\frac{\eta^2\kappa^2}{2}\|\hat{\boldsymbol{x}}^{t+1} - \hat{\boldsymbol{x}}^{*,\gamma}\|_2^2 - \sum_{I\in\mathcal{I}_i}\frac{\eta^2\kappa^2}{4}\|\hat{\boldsymbol{x}}^{t+1} - \hat{\boldsymbol{x}}^t\|_2^2\Big] \\
&= \sum_{t=0}^{T-1}\Big[\eta\mu\|\hat{\boldsymbol{x}}^{t+1} - \hat{\boldsymbol{x}}^{*,\gamma}\|_2^2 - |\mathcal{I}|\frac{\eta^2\kappa^2}{2}\|\hat{\boldsymbol{x}}^{t+1} - \hat{\boldsymbol{x}}^{*,\gamma}\|_2^2 - |\mathcal{I}|\frac{\eta^2\kappa^2}{4}\|\hat{\boldsymbol{x}}^{t+1} - \hat{\boldsymbol{x}}^t\|_2^2\Big] \\
&\stackrel{(v)}{\geq} \sum_{t=0}^{T-1}\Big[\eta\mu\|\hat{\boldsymbol{x}}^{t+1} - \hat{\boldsymbol{x}}^{*,\gamma}\|_2^2 - |\mathcal{I}|\frac{\eta^2\kappa^2}{2}\|\hat{\boldsymbol{x}}^{t+1} - \hat{\boldsymbol{x}}^{*,\gamma}\|_2^2 - |\mathcal{I}|\frac{\eta^2\kappa^2}{2}(\|\hat{\boldsymbol{x}}^{*,\gamma} - \hat{\boldsymbol{x}}^t\|_2^2 + \|\hat{\boldsymbol{x}}^{t+1} - \hat{\boldsymbol{x}}^{*,\gamma}\|_2^2)\Big] \\
&= \sum_{t=0}^{T-1}\Big[\eta\mu\|\hat{\boldsymbol{x}}^{t+1} - \hat{\boldsymbol{x}}^{*,\gamma}\|_2^2 - |\mathcal{I}|\eta^2\kappa^2\|\hat{\boldsymbol{x}}^{t+1} - \hat{\boldsymbol{x}}^{*,\gamma}\|_2^2 - |\mathcal{I}|\frac{\eta^2\kappa^2}{2}\|\hat{\boldsymbol{x}}^{*,\gamma} - \hat{\boldsymbol{x}}^t\|_2^2\Big] \\
&= \sum_{t=0}^{T-1}\Big[(\frac{2\eta\mu}{3} - |\mathcal{I}|\eta^2\kappa^2)\|\hat{\boldsymbol{x}}^{t+1} - \hat{\boldsymbol{x}}^{*,\gamma}\|_2^2\Big] + \sum_{t=0}^{T-1}\Big[(\frac{\eta\mu}{3})\|\hat{\boldsymbol{x}}^{t+1} - \hat{\boldsymbol{x}}^{*,\gamma}\|_2^2 - |\mathcal{I}|\frac{\eta^2\kappa^2}{2}\|\hat{\boldsymbol{x}}^{*,\gamma} - \hat{\boldsymbol{x}}^t\|_2^2\Big] \\
&= \sum_{t=0}^{T-1}\Big[(\frac{2\eta\mu}{3} - |\mathcal{I}|\eta^2\kappa^2)\|\hat{\boldsymbol{x}}^{t+1} - \hat{\boldsymbol{x}}^{*,\gamma}\|_2^2\Big] + \sum_{t=1}^{T-1}\Big[(\frac{\eta\mu}{3} - \frac{|\mathcal{I}|\eta^2\kappa^2}{2})\|\hat{\boldsymbol{x}}^t - \hat{\boldsymbol{x}}^{*,\gamma}\|_2^2\Big] + \frac{\eta\mu}{3}\|\hat{\boldsymbol{x}}^T - \hat{\boldsymbol{x}}^{*,\gamma}\|_2^2 - \frac{|\mathcal{I}|\eta^2\kappa^2}{2}\|\hat{\boldsymbol{x}}^{*,\gamma} - \hat{\boldsymbol{x}}^0\|_2^2.
\end{aligned}
\tag{34}
$$

where $(i)$ due to Lemma D.5 and the non-negativity of Bregman divergence, $(ii)$ due to Corollary D.7, $(iii)$ due to the strong monotonicity in Lemma D.2, $(iv)$ due to the smoothness of instantaneous regret in Lemma E.1, $(v)$ due to the mean-value inequality $\|\boldsymbol{a} + \boldsymbol{b}\|_2^2 \leq 2\|\boldsymbol{a}\|_2^2 + 2\|\boldsymbol{b}\|_2^2$. Rearranging the terms in Eq. (34), we have

$$
\begin{aligned}
&\sum_{i\in\mathcal{N}}\sum_{I\in\mathcal{I}_i}\zeta_I\Big[D_\psi(\sigma_i^{*,\gamma}(I)\|\hat{\boldsymbol{\theta}}_I^0) - \eta\langle \hat{\boldsymbol{m}}_i^{*,\gamma}(I), \hat{\boldsymbol{\theta}}_I^0\rangle\Big] + \frac{|\mathcal{I}|\eta^2\kappa^2}{2}\|\hat{\boldsymbol{x}}^{*,\gamma} - \hat{\boldsymbol{x}}^0\|_2^2 \\
&\geq \sum_{t=0}^{T-1}\Big[(\frac{2\eta\mu}{3} - |\mathcal{I}|\eta^2\kappa^2)\|\hat{\boldsymbol{x}}^{t+1} - \hat{\boldsymbol{x}}^{*,\gamma}\|_2^2\Big] + \sum_{t=1}^{T-1}\Big[(\frac{\eta\mu}{3} - \frac{|\mathcal{I}|\eta^2\kappa^2}{2})\|\hat{\boldsymbol{x}}^t - \hat{\boldsymbol{x}}^{*,\gamma}\|_2^2\Big] + \frac{\eta\mu}{3}\|\hat{\boldsymbol{x}}^T - \hat{\boldsymbol{x}}^{*,\gamma}\|_2^2
\end{aligned}
\tag{35}
$$

Setting $\frac{\eta\mu}{3} - \frac{|\mathcal{I}|\eta^2\kappa^2}{2} \geq \frac{\eta\mu}{6}$, i.e., $0 < \eta \leq \frac{\mu}{3|\mathcal{I}|\kappa^2}$, Eq. (35) leads to for any $T > 0$:

$$\frac{\eta\mu}{3}\sum_{t=0}^{T-1}\|\hat{\boldsymbol{x}}^{t+1} - \hat{\boldsymbol{x}}^{*,\gamma}\|_2^2 \leq \sum_{i \in \mathcal{N}}\sum_{I \in \mathcal{I}_i}\zeta_I\Big[D_\psi(\sigma_i^{*,\gamma}(I)\|\hat{\boldsymbol{\theta}}_I^0) - \eta\langle\hat{\boldsymbol{m}}_i^{*,\gamma}(I), \hat{\boldsymbol{\theta}}_I^0\rangle\Big] + \frac{|\mathcal{I}|\eta^2\kappa^2}{2}\|\hat{\boldsymbol{x}}^{*,\gamma} - \hat{\boldsymbol{x}}^0\|_2^2 = O(1).$$

Therefore, the sequence $\{\hat{\boldsymbol{x}}^t\}_{t\geq 1}$ converges to $\hat{\boldsymbol{x}}^{*,\gamma}$, showing the asymptotic last-iterate convergence of RTPCFR+ algorithm.

We next prove the following claim, which, combining with the aforementioned last-iterate convergence of optimistic OMD with step size $\eta \leq \mu/(3|\mathcal{I}|\kappa^2)$, finishes the whole proof for Theorem 5.1.

**Claim E.3.** *For any information set $I \in \mathcal{I}_i$, any initialization $\hat{\boldsymbol{\theta}}_I^0 \in \mathbb{R}_{\geq 0}^{|A(I)|}$, any step size $\eta > 0$, the output behavioral strategy $\hat{\sigma}_i^t$ of the optimistic OMD algorithm converges asymptotically to the Nash equilibrium $\hat{\sigma}^{*,\gamma}$.*

It suffices to show that there exists an initialization $\hat{\boldsymbol{\theta}}_I^{0'}$ and step size $\eta' = \mu/(3|\mathcal{I}|\kappa^2)$, such that the output strategy of optimistic OMD with $\hat{\boldsymbol{\theta}}_I^{0'}$ and $\eta' = \mu/(3|\mathcal{I}|\kappa^2)$ (whose output sequence strategy $\{\hat{\boldsymbol{x}}^t\}_{t\geq 1}$ converges to $\hat{\boldsymbol{x}}^{*,\gamma}$) is the same as that of OOMD with any fixed initialization $\hat{\boldsymbol{\theta}}_I^0 \in \mathbb{R}_{\geq 0}^{|A(I)|}$ and any step size $\eta > 0$.

Actually, $\forall$ fixed initialization $\hat{\boldsymbol{\theta}}_I^0$, let $\hat{\boldsymbol{\theta}}_I^{0'} = \frac{\eta'}{\eta}\hat{\boldsymbol{\theta}}_I^0$. We demonstrate by induction the following relationship between unnormalized strategy $\{\hat{\boldsymbol{\theta}}_I^{t'}\}_{t\geq 0}$ and $\{\hat{\boldsymbol{\theta}}_I^t\}_{t\geq 0}$ generated by running OOMD with $(\hat{\boldsymbol{\theta}}_I^{0'}, \eta')$ and $(\hat{\boldsymbol{\theta}}_I^0, \eta)$ respectively: $\hat{\boldsymbol{\theta}}_I^{t'} = \frac{\eta'}{\eta}\hat{\boldsymbol{\theta}}_I^t$, for $t \geq 0$. The base case when $t = 0$ holds obviously due to the construction of initialization $\hat{\boldsymbol{\theta}}_I^{0'} = \frac{\eta'}{\eta}\hat{\boldsymbol{\theta}}_I^0$. Assume that claim holds when for iteration $t \geq 0$, then for the iteration $t + 1$, we have

$$\hat{\boldsymbol{\theta}}_I^{t+1'} = [\hat{\boldsymbol{\theta}}_I^{t'} + \eta'\hat{\boldsymbol{m}}_i^{t+1}(I)]^+ = [\frac{\eta'}{\eta}\hat{\boldsymbol{\theta}}_I^t + \eta'\hat{\boldsymbol{m}}_i^{t+1}(I)]^+ = \frac{\eta'}{\eta}[\hat{\boldsymbol{\theta}}_I^t + \eta\hat{\boldsymbol{m}}_i^{t+1}(I)]^+ = \frac{\eta'}{\eta}\hat{\boldsymbol{\theta}}_I^{t+1},$$

where the above first inequality holds due to the update rule of OOMD, the second equality due to the inductive hypothesis. Therefore, $\hat{\boldsymbol{\theta}}_I^{t'} = \frac{\eta'}{\eta}\hat{\boldsymbol{\theta}}_I^t$, for $t \geq 0$, and so we further have that the output behavioral strategy of OOMD is

$$\boldsymbol{\theta}_I^{t+1'} = [\hat{\boldsymbol{\theta}}_I^{t'} + \eta'\hat{\boldsymbol{m}}_i^t(I)]^+ = [\frac{\eta'}{\eta}\hat{\boldsymbol{\theta}}_I^t + \eta'\hat{\boldsymbol{m}}_i^{t+1}(I)]^+ = \frac{\eta'}{\eta}[\hat{\boldsymbol{\theta}}_I^t + \eta\hat{\boldsymbol{m}}_i^{t+1}(I)]^+ = \frac{\eta'}{\eta}\boldsymbol{\theta}_I^{t+1}$$

$$\sigma_i^{t+1'} = \frac{\boldsymbol{\theta}_I^{t+1'}}{\langle\boldsymbol{\theta}_I^{t+1'}, \mathbf{1}\rangle} = \frac{\frac{\eta'}{\eta}\boldsymbol{\theta}_I^{t+1}}{\frac{\eta'}{\eta}\langle\boldsymbol{\theta}_I^{t+1}, \mathbf{1}\rangle} = \sigma_i^{t+1},$$

which finishes the proof of Claim E.3. $\qquad\qquad\square$

# F. Proof for the Average-Iterate Convergence of RTPCFR+ Algorithm

**Theorem F.1** (Equivalent Restatement of Theorem 5.3). *Assuming all players follow the update rule of PCFR$^+$ with any $\hat{\boldsymbol{\theta}}_I^0 \in \mathbb{R}_{\geq 0}^{|A(I)|}$ and $\eta > 0$, then the uniformly average strategy profile $\bar{\boldsymbol{x}}^T = \sum_{t=1}^T \boldsymbol{x}^t/T$ converges with a rate $O(1/\sqrt{T})$ to the set of NEs of the perturbed regularized EFGs in (6) for any $\gamma \geq 0$ and $\mu \geq 0$.*

*Proof.* We aim to give an upper bound for the $i$-th player's cumulative regret $R_i^T = \max_{\hat{\boldsymbol{x}}_i \in \mathcal{X}_i^\gamma}\sum_{t=1}^T\langle\hat{\boldsymbol{\ell}}_i^t, \hat{\boldsymbol{x}}_i^t - \hat{\boldsymbol{x}}_i\rangle$. We start from Eq. (26) and set $\boldsymbol{\theta} = \sigma_i(I)$, obtaining that:

$$\eta\langle\hat{\boldsymbol{m}}_i^{t+1}(I), \sigma_i(I) - \boldsymbol{\theta}_I^{t+1}\rangle$$

$$\leq D_\psi(\sigma_i(I)\|\hat{\boldsymbol{\theta}}_I^t) - D_\psi(\sigma_i(I)\|\hat{\boldsymbol{\theta}}_I^{t+1}) - D_\psi(\hat{\boldsymbol{\theta}}_I^{t+1}\|\hat{\boldsymbol{\theta}}_I^t) - \eta\langle\hat{\boldsymbol{m}}_i^{t+1}(I), \boldsymbol{\theta}_I^{t+1} - \hat{\boldsymbol{\theta}}_I^{t+1}\rangle$$

$$= D_\psi(\sigma_i(I)\|\hat{\boldsymbol{\theta}}_I^t) - D_\psi(\sigma_i(I)\|\hat{\boldsymbol{\theta}}_I^{t+1}) - D_\psi(\hat{\boldsymbol{\theta}}_I^{t+1}\|\hat{\boldsymbol{\theta}}_I^t) - \eta\langle\hat{\boldsymbol{m}}_i^{t+1}(I) - \hat{\boldsymbol{m}}_i^t(I), \boldsymbol{\theta}_I^{t+1} - \hat{\boldsymbol{\theta}}_I^{t+1}\rangle - \eta\langle\hat{\boldsymbol{m}}_i^t(I), \boldsymbol{\theta}_I^{t+1} - \hat{\boldsymbol{\theta}}_I^{t+1}\rangle$$

$$\leq D_\psi(\sigma_i(I)\|\hat{\boldsymbol{\theta}}_I^t) - D_\psi(\sigma_i(I)\|\hat{\boldsymbol{\theta}}_I^{t+1}) - D_\psi(\hat{\boldsymbol{\theta}}_I^{t+1}\|\hat{\boldsymbol{\theta}}_I^t)$$

$$+ \eta\Big[\frac{\eta}{2}\|\hat{\boldsymbol{m}}_i^{t+1}(I) - \hat{\boldsymbol{m}}_i^t(I)\|_2^2 + \frac{\|\boldsymbol{\theta}_I^{t+1} - \hat{\boldsymbol{\theta}}_I^{t+1}\|_2^2}{2\eta}\Big] + \Big[D_\psi(\hat{\boldsymbol{\theta}}_I^{t+1}\|\hat{\boldsymbol{\theta}}_I^t) - D_\psi(\hat{\boldsymbol{\theta}}_I^{t+1}\|\boldsymbol{\theta}_I^{t+1}) - D_\psi(\boldsymbol{\theta}_I^{t+1}\|\hat{\boldsymbol{\theta}}_I^t)\Big]$$

$$= D_\psi(\sigma_i(I)\|\hat{\boldsymbol{\theta}}_I^t) - D_\psi(\sigma_i(I)\|\hat{\boldsymbol{\theta}}_I^{t+1}) + \frac{\eta^2}{2}\|\hat{\boldsymbol{m}}_i^{t+1}(I) - \hat{\boldsymbol{m}}_i^t(I)\|_2^2 - D_\psi(\boldsymbol{\theta}_I^{t+1}\|\hat{\boldsymbol{\theta}}_I^t),$$

$$\tag{36}$$

where the second inequality holds due to Cauchy-Schwartz/mean-value inequality (in the first square brackets) and Lemma D.1 (in the second square brackets). By using $\langle \hat{m}_i^{t+1}(I), \theta_I^{t+1}\rangle = \langle \hat{m}_i^{t+1}(I), \sigma_i^{t+1}(I)\rangle \|\theta_I^{t+1}\|_1 = 0$, we can further simplify the left-hand side of Eq. (36) as

$$\langle \hat{m}_i^{t+1}(I), \sigma_i(I) - \theta_I^{t+1}\rangle = \langle \hat{m}_i^{t+1}(I), \sigma_i(I)\rangle = \langle \hat{v}_i^{t+1}(I) - \langle \hat{v}_i^{t+1}(I), \sigma_i^{t+1}(I)\rangle \mathbf{1}, \sigma_i(I)\rangle$$
$$= \langle \hat{v}_i^{t+1}(I), \sigma_i(I) - \sigma_i^{t+1}(I)\rangle = \frac{1}{1 - \alpha_I} \langle \hat{v}_i^{t+1}(I), \hat{\sigma}_i(I) - \hat{\sigma}_i^{t+1}(I)\rangle. \tag{37}$$

Then, multiplying both sides in Eq. (36) with $\zeta_I = (1 - \alpha_I)\pi_i^{\hat{\sigma}}(I)$, we have

$$\pi_i^{\hat{\sigma}}(I)\langle -\hat{v}_i^{t+1}(I), \hat{\sigma}_i^{t+1}(I) - \hat{\sigma}_i(I)\rangle$$
$$\leq \frac{\zeta_I}{\eta} \Big[ D_\psi(\sigma_i(I)\|\hat{\theta}_I^t) - D_\psi(\sigma_i(I)\|\hat{\theta}_I^{t+1}) - D_\psi(\theta_I^{t+1}\|\hat{\theta}_I^t) + \frac{\eta^2}{2}\|\hat{m}_i^{t+1}(I) - \hat{m}_i^t(I)\|_2^2 \Big] \tag{38}$$

Taking summation of both sides in Eq. (38) over $t = 0, 1, \cdots, T - 1$, $I \in \mathcal{I}_i$, and applying Lemma D.7, we have

$$\sum_{t=0}^{T-1}\langle \hat{\ell}_i^{t+1}, \hat{x}_i^{t+1} - \hat{x}_i\rangle$$
$$\leq \sum_{t=0}^{T-1}\sum_{I \in \mathcal{I}_i} \frac{\zeta_I}{\eta} \Big[ D_\psi(\sigma_i(I)\|\hat{\theta}_I^t) - D_\psi(\sigma_i(I)\|\hat{\theta}_I^{t+1}) - D_\psi(\theta_I^{t+1}\|\hat{\theta}_I^t) + \frac{\eta^2}{2}\|\hat{m}_i^{t+1}(I) - \hat{m}_i^t(I)\|_2^2 \Big] \tag{39}$$
$$\leq \sum_{I \in \mathcal{I}_i} \zeta_I \Big[ \frac{D_\psi(\sigma_i(I)\|\hat{\theta}_I^0) - D_\psi(\sigma_i(I)\|\hat{\theta}_I^T)}{\eta} + \frac{\eta}{2}\sum_{t=0}^{T-1}\|\hat{m}_i^{t+1}(I) - \hat{m}_i^t(I)\|_2^2 \Big].$$

According to the proof for Claim E.3, we know that, as long as we set another initialization $\hat{\theta}_I^{0'} = \frac{\eta'}{\eta}\hat{\theta}_I^0$, then for any step size $\eta' > 0$, the output sequence strategy $\{\hat{x}_i^{t'}\}_{t \geq 1}$ of OOMD algorithm (with initialization $\hat{\theta}_I^{0'}$ and step size $\eta'$) is the same as that $\{\hat{x}_i^t\}_{t \geq 1}$ of OOMD algorithm (with initialization $\hat{\theta}_I^0$ and step size $\eta$). Applying such result into Eq. (39), we have for any $\eta' > 0$:

$$\sum_{t=0}^{T-1}\langle \hat{\ell}_i^{t+1}, \hat{x}_i^{t+1} - \hat{x}_i\rangle \leq \sum_{I \in \mathcal{I}_i} \zeta_I \Big[ \frac{D_\psi(\sigma_i(I)\|\hat{\theta}_I^{0'}) - D_\psi(\sigma_i(I)\|\hat{\theta}_I^{T'})}{\eta'} + \frac{\eta'}{2}\sum_{t=0}^{T-1}\|\hat{m}_i^{t+1}(I) - \hat{m}_i^t(I)\|_2^2 \Big]$$
$$= \sum_{I \in \mathcal{I}_i} \zeta_I \Big[ \frac{D_\psi(\sigma_i(I)\|\frac{\eta'}{\eta}\hat{\theta}_I^0) - D_\psi(\sigma_i(I)\|\hat{\theta}_I^{T'})}{\eta'} + \frac{\eta'}{2}\sum_{t=0}^{T-1}\|\hat{m}_i^{t+1}(I) - \hat{m}_i^t(I)\|_2^2 \Big]$$
$$\leq \sum_{I \in \mathcal{I}_i} \zeta_I \Big[ \frac{\|\sigma_i(I) - \frac{\eta'}{\eta}\hat{\theta}_I^0\|_2^2}{2\eta'} + \frac{\eta'}{2}\sum_{t=0}^{T-1}\|\hat{m}_i^{t+1}(I) - \hat{m}_i^t(I)\|_2^2 \Big]$$
$$\leq \sum_{I \in \mathcal{I}_i} \zeta_I \Big[ \frac{\|\sigma_i(I)\|_2^2 + \|\frac{\eta'}{\eta}\hat{\theta}_I^0\|_2^2}{\eta'} + \frac{\eta'}{2}\sum_{t=0}^{T-1}\|\hat{m}_i^{t+1}(I) - \hat{m}_i^t(I)\|_2^2 \Big]$$
$$\leq \sum_{I \in \mathcal{I}_i} \zeta_I \Big[ \frac{1}{\eta'} + \frac{\eta'\|\hat{\theta}_I^0\|_2^2}{\eta^2} + \frac{\eta'}{2}\sum_{t=0}^{T-1}\|\hat{m}_i^{t+1}(I) - \hat{m}_i^t(I)\|_2^2 \Big]$$
$$\leq \sum_{I \in \mathcal{I}_i} 2\zeta_I \sqrt{\frac{\|\hat{\theta}_I^0\|_2^2}{\eta^2} + \frac{1}{2}\sum_{t=0}^{T-1}\|\hat{m}_i^{t+1}(I) - \hat{m}_i^t(I)\|_2^2} \tag{40}$$

where the third inequality holds due to the mean-value inequality $\|a + b\|^2 \leq 2\|a\|^2 + 2\|b\|^2$, the fourth inequality since for the simplex vector $\|\sigma_i(I)\|_1 = 1$, the last inequality due to the basic inequality $\frac{a}{\eta} + b\eta \leq 2\sqrt{ab}$ for any $\eta > 0$. $\qquad\square$

## G. Proof for the Asymptotic Last-Iterate Convergence of RTPDCFR+ Algorithm

***Proof of Theorem 6.1***. we first consider non-increasing sequence of step size $\{\eta^t\}_{t\geq 0}$, i.e. $\lim_{t\to\infty} \eta^t = \eta, \eta \in \left(0, \frac{\mu}{3|\mathcal{I}|\kappa^2}\right)$.
The update rule of OOMD with bounded non-increasing learning rate $\{\eta^t\}_{t\geq 0}$ is as follow:

$$\boldsymbol{\theta}_I^{t+1} \in \arg\min_{\boldsymbol{\theta}_I \in \mathbb{R}_{\geq 0}^{|A(I)|}} \left\{ \langle -\hat{\boldsymbol{m}}_i^t(I), \boldsymbol{\theta}_I \rangle + \frac{1}{\eta^{t+1}} D_\psi(\boldsymbol{\theta}_I \| \hat{\boldsymbol{\theta}}_I^t) \right\}, \tag{41}$$

$$\hat{\boldsymbol{\theta}}_I^{t+1} \in \arg\min_{\boldsymbol{\theta}_I \in \mathbb{R}_{\geq 0}^{|A(I)|}} \left\{ \langle -\hat{\boldsymbol{m}}_i^{t+1}(I), \boldsymbol{\theta}_I \rangle + \frac{1}{\eta^{t+1}} D_\psi(\boldsymbol{\theta}_I \| \hat{\boldsymbol{\theta}}_I^t) \right\}. \tag{42}$$

By replacing the constant step size $\eta$ with $\eta^{t+1}$ in Eqs. (27)-(33), we can obtain

$$\eta^{t+1}\beta_I \langle -\hat{\boldsymbol{v}}_i^{t+1}(I), \hat{\sigma}_i^{t+1}(I) - \hat{\sigma}_i^{*,\gamma}(I) \rangle - \frac{(\eta^{t+1})^2\zeta_I}{2} \|\hat{\boldsymbol{m}}_i^{t+1}(I) - \hat{\boldsymbol{m}}_i^{*,\gamma}(I)\|_2^2 - \frac{(\eta^{t+1})^2\zeta_I}{4}\|\hat{\boldsymbol{m}}_i^{t+1}(I) - \hat{\boldsymbol{m}}_i^t(I)\|_2^2$$
$$\leq \zeta_I \left[ D_\psi(\sigma_i^{*,\gamma}(I)\|\hat{\boldsymbol{\theta}}_I^t) - D_\psi(\sigma_i^{*,\gamma}(I)\|\hat{\boldsymbol{\theta}}_I^{t+1}) - \eta^{t+1}\langle \hat{\boldsymbol{m}}_i^{*,\gamma}(I), \hat{\boldsymbol{\theta}}_I^t - \hat{\boldsymbol{\theta}}_I^{t+1} \rangle \right]. \tag{43}$$

Then, taking summation of both sides in Eq. (43) over $t = 0, 1, \cdots, T-1, i \in \mathcal{N}, I \in \mathcal{I}_i$, we have

$$\sum_{t=0}^{T-1}\sum_{i\in\mathcal{N}}\sum_{I\in\mathcal{I}_i}\left[\eta^{t+1}\beta_I\langle -\hat{\boldsymbol{v}}_i^{t+1}(I),\hat{\sigma}_i^{t+1}(I)-\hat{\sigma}_i^{*,\gamma}(I)\rangle - \frac{(\eta^{t+1})^2\zeta_I}{2}\|\hat{\boldsymbol{m}}_i^{t+1}(I)-\hat{\boldsymbol{m}}_i^{*,\gamma}(I)\|_2^2 - \frac{(\eta^{t+1})^2\zeta_I}{4}\|\hat{\boldsymbol{m}}_i^{t+1}(I)-\hat{\boldsymbol{m}}_i^t(I)\|_2^2\right]$$

$$\leq \sum_{i\in\mathcal{N}}\sum_{I\in\mathcal{I}_i}\zeta_I\left[D_\psi(\sigma_i^{*,\gamma}(I)\|\hat{\boldsymbol{\theta}}_I^0) - D_\psi(\sigma_i^{*,\gamma}(I)\|\hat{\boldsymbol{\theta}}_I^T) - \langle\hat{\boldsymbol{m}}_i^{*,\gamma}(I), \sum_{t=0}^{T-1}\eta^{t+1}(\hat{\boldsymbol{\theta}}_I^t - \hat{\boldsymbol{\theta}}_I^{t+1})\rangle\right] \tag{44}$$

$$= \sum_{i\in\mathcal{N}}\sum_{I\in\mathcal{I}_i}\zeta_I\left[D_\psi(\sigma_i^{*,\gamma}(I)\|\hat{\boldsymbol{\theta}}_I^0) - D_\psi(\sigma_i^{*,\gamma}(I)\|\hat{\boldsymbol{\theta}}_I^T) - \langle\hat{\boldsymbol{m}}_i^{*,\gamma}(I), \eta^1\hat{\boldsymbol{\theta}}_I^0 - \eta^T\hat{\boldsymbol{\theta}}_I^T + \sum_{t=1}^{T-1}(\eta^{t+1}-\eta^t)\hat{\boldsymbol{\theta}}_I^t\rangle\right]$$

$$\leq \sum_{i\in\mathcal{N}}\sum_{I\in\mathcal{I}_i}\zeta_I\left[D_\psi(\sigma_i^{*,\gamma}(I)\|\hat{\boldsymbol{\theta}}_I^0) - \langle\hat{\boldsymbol{m}}_i^{*,\gamma}(I), \eta^1\hat{\boldsymbol{\theta}}_I^0\rangle\right],$$

where the last inequality holds due to the non-negativity of Bregman divergence, and the non-increasing property of sequence $\{\eta^t\}_{t\geq 0}$ (i.e. $\eta^{t+1} - \eta^t \leq 0$) and Lemma D.5. The left-hand side of Eq. (44) can be further lower bounded as in Eq. (34):

$$\sum_{t=0}^{T-1}\sum_{i\in\mathcal{N}}\sum_{I\in\mathcal{I}_i}\left[\eta^{t+1}\beta_I\langle -\hat{\boldsymbol{v}}_i^{t+1}(I),\hat{\sigma}_i^{t+1}(I)-\hat{\sigma}_i^{*,\gamma}(I)\rangle - \frac{(\eta^{t+1})^2\zeta_I}{2}\|\hat{\boldsymbol{m}}_i^{t+1}(I)-\hat{\boldsymbol{m}}_i^{*,\gamma}(I)\|_2^2 - \frac{(\eta^{t+1})^2\zeta_I}{4}\|\hat{\boldsymbol{m}}_i^{t+1}(I)-\hat{\boldsymbol{m}}_i^t(I)\|_2^2\right]$$

$$\geq \sum_{t=0}^{T-1}\left[\left(\frac{2\eta^{t+1}\mu}{3} - |\mathcal{I}|(\eta^{t+1})^2\kappa^2\right)\|\hat{\boldsymbol{x}}^{t+1} - \hat{\boldsymbol{x}}^{*,\gamma}\|_2^2\right] + \sum_{t=0}^{T-1}\left[\left(\frac{\eta^{t+1}\mu}{3}\right)\|\hat{\boldsymbol{x}}^{t+1} - \hat{\boldsymbol{x}}^{*,\gamma}\|_2^2 - |\mathcal{I}|\frac{(\eta^{t+1})^2\kappa^2}{2}\|\hat{\boldsymbol{x}}^{*,\gamma} - \hat{\boldsymbol{x}}^t\|_2^2\right]$$

$$= \sum_{t=0}^{T-1}\left[\left(\frac{2\eta^{t+1}\mu}{3} - |\mathcal{I}|(\eta^{t+1})^2\kappa^2\right)\|\hat{\boldsymbol{x}}^{t+1} - \hat{\boldsymbol{x}}^{*,\gamma}\|_2^2\right] + \sum_{t=1}^{T-1}\left[\left(\frac{\eta^t\mu}{3} - \frac{|\mathcal{I}|(\eta^{t+1})^2\kappa^2}{2}\right)\|\hat{\boldsymbol{x}}^t - \hat{\boldsymbol{x}}^{*,\gamma}\|_2^2\right]$$

$$+ \frac{\eta^T\mu}{3}\|\hat{\boldsymbol{x}}^T - \hat{\boldsymbol{x}}^{*,\gamma}\|_2^2 - \frac{|\mathcal{I}|(\eta^1)^2\kappa^2}{2}\|\hat{\boldsymbol{x}}^{*,\gamma} - \hat{\boldsymbol{x}}^0\|_2^2$$

$$\geq \sum_{t=0}^{T-1}\left[\left(\frac{2\eta^{t+1}\mu}{3} - |\mathcal{I}|(\eta^{t+1})^2\kappa^2\right)\|\hat{\boldsymbol{x}}^{t+1} - \hat{\boldsymbol{x}}^{*,\gamma}\|_2^2\right] + \sum_{t=1}^{T-1}\left[\left(\frac{\eta^{t+1}\mu}{3} - \frac{|\mathcal{I}|(\eta^{t+1})^2\kappa^2}{2}\right)\|\hat{\boldsymbol{x}}^t - \hat{\boldsymbol{x}}^{*,\gamma}\|_2^2\right] - \frac{|\mathcal{I}|(\eta^1)^2\kappa^2}{2}\|\hat{\boldsymbol{x}}^{*,\gamma} - \hat{\boldsymbol{x}}^0\|_2^2,$$

$$\tag{45}$$

where the last inequality holds due to the fact that $\eta^t \geq \eta^{t+1}$. Combining the upper bound in Eq. (44) and lower bound in Eq. (45), and setting $\eta^{t+1} \in (\eta, \frac{\mu}{3|\mathcal{I}|\kappa^2}]$ we have

$$
\begin{aligned}
O(1) &= \sum_{i \in \mathcal{N}} \sum_{I \in \mathcal{I}_i} \zeta_I \Big[ D_\psi(\sigma_i^{*,\gamma}(I) \| \hat{\boldsymbol{\theta}}_I^0) - \langle \hat{\boldsymbol{m}}_i^{*,\gamma}(I), \eta^1 \hat{\boldsymbol{\theta}}_I^0 \rangle \Big] + \frac{|\mathcal{I}|(\eta^1)^2 \kappa^2}{2} \| \hat{\boldsymbol{x}}^{*,\gamma} - \hat{\boldsymbol{x}}^0 \|_2^2 \\
&\geq \sum_{t=0}^{T-1} \Big[ \Big( \frac{2\eta^{t+1}\mu}{3} - |\mathcal{I}|(\eta^{t+1})^2 \kappa^2 \Big) \| \hat{\boldsymbol{x}}^{t+1} - \hat{\boldsymbol{x}}^{*,\gamma} \|_2^2 \Big] + \sum_{t=1}^{T-1} \Big[ \Big( \frac{\eta^{t+1}\mu}{3} - \frac{|\mathcal{I}|(\eta^{t+1})^2 \kappa^2}{2} \Big) \| \hat{\boldsymbol{x}}^t - \hat{\boldsymbol{x}}^{*,\gamma} \|_2^2 \Big] \qquad (46) \\
&\geq \sum_{t=0}^{T-1} \Big[ \Big( \frac{\eta\mu}{6} \Big) \| \hat{\boldsymbol{x}}^{t+1} - \hat{\boldsymbol{x}}^{*,\gamma} \|_2^2 \Big].
\end{aligned}
$$

Therefore, when running (weighted) OOMD algorithm in Eqs. (41)-(42), with non-increasing step sizes $\{\eta^t\}_{t \geq 0}$ that satisfies $\eta^t \in (\eta, \frac{\mu}{3|\mathcal{I}|\kappa^2}]$, the output sequence $\{\hat{\boldsymbol{x}}^t\}_{t \geq 1}$ satisfies the inequality for any $T \geq 1$:

$$
\sum_{t=1}^{T} \| \hat{\boldsymbol{x}}^t - \hat{\boldsymbol{x}}^{*,\gamma} \|_2^2 \leq \frac{6}{\eta\mu} \Big[ \sum_{i \in \mathcal{N}} \sum_{I \in \mathcal{I}_i} \zeta_I \Big[ D_\psi(\sigma_i^{*,\gamma}(I) \| \hat{\boldsymbol{\theta}}_I^0) - \langle \hat{\boldsymbol{m}}_i^{*,\gamma}(I), \eta^1 \hat{\boldsymbol{\theta}}_I^0 \rangle \Big] + \frac{|\mathcal{I}|(\eta^1)^2 \kappa^2}{2} \| \hat{\boldsymbol{x}}^{*,\gamma} - \hat{\boldsymbol{x}}^0 \|_2^2 \Big] = O(1).
$$

Therefore, $\{\hat{\boldsymbol{x}}^t\}_{t \geq 1}$ converges to the Nash equilibrium $\hat{\boldsymbol{x}}^{*,\gamma}$, completing the proof of the first part. $\qquad \square$

We next prove the following claim, which, combining with the aforementioned last-iterate convergence of (weighted) optimistic OMD with step size $\eta^t \in (\eta, \frac{\mu}{3|\mathcal{I}|\kappa^2}]$, finishes the whole proof for Theorem 6.1.

**Claim G.1.** *For any information set $I \in \mathcal{I}_i$, any initialization $\hat{\boldsymbol{\theta}}_I^0 \in \mathbb{R}_{\geq 0}^{|A(I)|}$, any non-increasing step sizes $\{\eta^t\}_{t \geq 0}$ with $\lim_{t \to \infty} \eta^t = \eta > 0$, the output behavior strategy $\hat{\sigma}_i^t$ of the weighted OOMD algorithm converges asymptotically to the Nash equilibrium $\hat{\sigma}^{*,\gamma}$.*

For any initialization $\hat{\boldsymbol{\theta}}_I^0 \in \mathbb{R}_{\geq 0}^{|A(I)|}$, non-increasing step sizes $\{\eta^t\}_{t \geq 0}$, we construct another initialization $\hat{\boldsymbol{\theta}}_I^{0'} = \frac{\eta^{0'}}{\eta^0} \hat{\boldsymbol{\theta}}_I^0$ and sequence of step size $\{\eta^{t'}\}_{t \geq 0}$ which satisfies $\eta^{t+1'} = \frac{\eta^{t'}}{\eta^t} \eta^{t+1}$ and $\eta^{0'} \leq \frac{\mu}{3|\mathcal{I}|\kappa^2}$. We then prove the following lemma.

**Lemma G.2.** *For any initialization $\hat{\boldsymbol{\theta}}_I^0 \in \mathbb{R}_{\geq 0}^{|A(I)|}$, any non-increasing step sizes $\{\eta^t\}_{t \geq 0}$ with $\lim_{t \to \infty} \eta^t = \eta > 0$, we have that: the constructed sequence of step sizes $\{\eta^{t'}\}_{t \geq 0}$ is non-increasing and $\lim_{t \to \infty} \eta^{t'} = \eta^{0'} \frac{\eta}{\eta^0} \in (0, \frac{\mu}{3|\mathcal{I}|\kappa^2}]$.*

*Proof.* Due to the non-increasing property of $\{\eta^t\}_{t \geq 0}$, we have $\eta^{t+1'} = \frac{\eta^{t'}}{\eta^t} \eta^{t+1} \leq \eta^{t'}$ and so $\{\eta^{t'}\}_{t \geq 0}$ is non-increasing and non-negative. From the monotone convergence theorem, the limit of sequence $\{\eta^{t'}\}_{t \geq 0}$ exists, and can be computed as:

$$
\lim_{t \to \infty} \eta^{t+1'} = \lim_{t \to \infty} \frac{\eta^{t+1'}}{\eta^{t'}} \frac{\eta^{t'}}{\eta^{t-1'}} \cdots \frac{\eta^{1'}}{\eta^{0'}} \eta^{0'} = \lim_{t \to \infty} \frac{\eta^{t+1}}{\eta^t} \frac{\eta^t}{\eta^{t-1}} \cdots \frac{\eta^1}{\eta^0} \eta^{0'} = \lim_{t \to \infty} \eta^{t+1} \frac{\eta^{0'}}{\eta^0} = \frac{\eta^{0'}}{\eta^0} \eta. \qquad \square
$$

From Lemma G.2, the constructed non-increasing step sizes $\{\eta^{t'}\}_{t \geq 0}$ satisfy $\eta^{t'} \in (\eta^{0'} \frac{\eta}{\eta^0}, \frac{\mu}{3|\mathcal{I}|\kappa^2}]$, so that the output strategy $\hat{\sigma}_i^t$ of weighted optimistic OMD with $\hat{\boldsymbol{\theta}}_I^{0'}$ and $\{\eta^{t'}\}_{t \geq 0}$ converges to Nash equilibrium $\hat{\sigma}_i^{*,\gamma}$. We next give the following lemma, which completes the whole proof for Claim G.1.

**Lemma G.3.** *The output behavioral strategy of weighted OOMD algorithm with any (fixed) initialization $\hat{\boldsymbol{\theta}}_I^0$ and step sizes $\{\eta^t\}_{t \geq 0}$, is the same as that of weighted OOMD algorithm with the constructed initialization $\hat{\boldsymbol{\theta}}_I^{0'}$ and step size $\{\eta^{t'}\}_{t \geq 0}$*

*Proof.* We first demonstrate by induction that, for any iteration $t \geq 0$: $\hat{\boldsymbol{\theta}}_I^{t'} = \frac{\eta^{t'}}{\eta^t} \hat{\boldsymbol{\theta}}_I^t$.

The base case when $t = 0$ holds trivially due to the construction of initialization $\hat{\boldsymbol{\theta}}_I^{0'} = \frac{\eta^{0'}}{\eta^0} \hat{\boldsymbol{\theta}}_I^0$. Assume the proposition holds

for the iteration $t$. Consider the iteration $t + 1$ of weighted OOMD algorithm:

$$\hat{\boldsymbol{\theta}}^{t+1'} = [\hat{\boldsymbol{\theta}}^{t'} + \eta^{t+1'}\hat{\boldsymbol{m}}_i^{t+1}(I)]^+ = [\frac{\eta^{t'}}{\eta^t}\hat{\boldsymbol{\theta}}_I^t + \eta^{t+1'}\hat{\boldsymbol{m}}_i^{t+1}(I)]^+$$

$$= [\frac{\eta^{t'}}{\eta^t}\hat{\boldsymbol{\theta}}_I^t + \frac{\eta^{t'}}{\eta^t}\eta^{t+1}\hat{\boldsymbol{m}}_i^{t+1}(I)]^+ = \frac{\eta^{t'}}{\eta^t}[\hat{\boldsymbol{\theta}}_I^t + \eta^{t+1}\hat{\boldsymbol{m}}_i^{t+1}(I)]^+ \qquad (47)$$

$$= \frac{\eta^{t+1'}}{\eta^{t+1}}[\hat{\boldsymbol{\theta}}_I^t + \eta^{t+1}\hat{\boldsymbol{m}}_i^{t+1}(I)]^+ = \frac{\eta^{t+1'}}{\eta^{t+1}}\hat{\boldsymbol{\theta}}^{t+1},$$

where the second equality holds due to the inductive hypothesis for iteration $t$, the third and the fifth equality due to the construction of step size $\eta^{t+1'}$. Therefore, for any $t \geq 0$: $\hat{\boldsymbol{\theta}}_I^{t'} = \frac{\eta^{t'}}{\eta^t}\hat{\boldsymbol{\theta}}_I^t$.

Then, for the output unnormalized strategy $\boldsymbol{\theta}_I^{t+1'}$ of weighted OOMD algorithm, we have

$$\boldsymbol{\theta}_I^{t+1'} = [\hat{\boldsymbol{\theta}}^{t'} + \eta^{t+1'}\hat{\boldsymbol{m}}_i^t(I)]^+ = [\frac{\eta^{t'}}{\eta^t}\hat{\boldsymbol{\theta}}^t + \frac{\eta^{t'}}{\eta^t}\eta^{t+1}\hat{\boldsymbol{m}}_i^t(I)]^+ = \frac{\eta^{t'}}{\eta^t}[\hat{\boldsymbol{\theta}}^t + \eta^{t+1}\hat{\boldsymbol{m}}_i^t(I)]^+ = \frac{\eta^{t'}}{\eta^t}\boldsymbol{\theta}_I^{t+1}. \qquad (48)$$

Finally, the behavioral strategy of the weighted OOMD algorithm with constructed initialization and step size is computed as follow, and is the same as the original behavioral strategy:

$$\sigma_i^{t+1'} = \frac{\boldsymbol{\theta}_I^{t+1'}}{\langle\boldsymbol{\theta}_I^{t+1'}, \mathbf{1}\rangle} = \frac{\frac{\eta^{t'}}{\eta^t}\boldsymbol{\theta}_I^{t+1}}{\frac{\eta^{t'}}{\eta^t}\langle\boldsymbol{\theta}_I^{t+1}, \mathbf{1}\rangle} = \sigma_i^{t+1}. \qquad \square$$

## H. Proof for the Average-Iterate Convergence of RTPDCFR+ Algorithm

**Theorem H.1** (**Equivalent Restatement of Theorem 6.2**). *Assuming all players follow the update rule of PDCFR$^+$ with any initialization $\hat{\boldsymbol{\theta}}_I^0 \in \mathbb{R}_{\geq 0}^{|A(I)|}$ and any sequence of positive step sizes $\{\eta^t\}_{t\geq 0}$. If $\{\frac{w^t}{\eta^t}\}_{t\geq 0}$ is a positive non-increasing sequence, then the weighted average strategy profile $\bar{\boldsymbol{x}}_{\boldsymbol{w}}^T = \frac{\sum_{t=1}^T w^t \boldsymbol{x}^t}{\sum_{t=1}^T w^t}$ converges with a rate of $O(\frac{|\mathcal{I}_i|\sqrt{\sum_{t=0}^{T-1} w^{t+1}\eta^{t+1}\|\hat{\boldsymbol{m}}_i^{t+1}(I)-\hat{\boldsymbol{m}}_i^t(I)\|_2^2}}{\sum_{t=0}^{T-1} w^{t+1}})$ to the set of NEs of the perturbed regularized EFGs in (6) for any $\gamma \geq 0$ and $\mu \geq 0$.*

*Proof.* We aim to give an upper bound for the $i$-th player's weighted cumulative regret $R_i^T = \max_{\hat{\boldsymbol{x}}_i \in \boldsymbol{\mathcal{X}}_i^\gamma} \sum_{t=1}^T w^t\langle\hat{\boldsymbol{\ell}}_i^t, \hat{\boldsymbol{x}}_i^t - \hat{\boldsymbol{x}}_i\rangle$. Similar to the proof for Theorem F.1, by replacing $\eta$ in Eq. (26) with step size $\eta^{t+1}$ and setting $\boldsymbol{\theta} = \sigma_i(I)$, we have

$$\eta^{t+1}\langle\hat{\boldsymbol{m}}_i^{t+1}(I), \sigma_i(I) - \boldsymbol{\theta}_I^{t+1}\rangle$$

$$\leq D_\psi(\sigma_i(I)\|\hat{\boldsymbol{\theta}}_I^t) - D_\psi(\sigma_i(I)\|\hat{\boldsymbol{\theta}}_I^{t+1}) - D_\psi(\hat{\boldsymbol{\theta}}_I^{t+1}\|\hat{\boldsymbol{\theta}}_I^t) - \eta^{t+1}\langle\hat{\boldsymbol{m}}_i^{t+1}(I), \boldsymbol{\theta}_I^{t+1} - \hat{\boldsymbol{\theta}}_I^{t+1}\rangle$$

$$= D_\psi(\sigma_i(I)\|\hat{\boldsymbol{\theta}}_I^t) - D_\psi(\sigma_i(I)\|\hat{\boldsymbol{\theta}}_I^{t+1}) - D_\psi(\hat{\boldsymbol{\theta}}_I^{t+1}\|\hat{\boldsymbol{\theta}}_I^t) - \eta^{t+1}\langle\hat{\boldsymbol{m}}_i^{t+1}(I) - \hat{\boldsymbol{m}}_i^t(I), \boldsymbol{\theta}_I^{t+1} - \hat{\boldsymbol{\theta}}_I^{t+1}\rangle - \eta^{t+1}\langle\hat{\boldsymbol{m}}_i^t(I), \boldsymbol{\theta}_I^{t+1} - \hat{\boldsymbol{\theta}}_I^{t+1}\rangle$$

$$\leq D_\psi(\sigma_i(I)\|\hat{\boldsymbol{\theta}}_I^t) - D_\psi(\sigma_i(I)\|\hat{\boldsymbol{\theta}}_I^{t+1}) - D_\psi(\hat{\boldsymbol{\theta}}_I^{t+1}\|\hat{\boldsymbol{\theta}}_I^t)$$

$$+ \eta^{t+1}\Big[\frac{\eta^{t+1}}{2}\|\hat{\boldsymbol{m}}_i^{t+1}(I) - \hat{\boldsymbol{m}}_i^t(I)\|_2^2 + \frac{\|\boldsymbol{\theta}_I^{t+1} - \hat{\boldsymbol{\theta}}_I^{t+1}\|_2^2}{2\eta^{t+1}}\Big] + \Big[D_\psi(\hat{\boldsymbol{\theta}}_I^{t+1}\|\hat{\boldsymbol{\theta}}_I^t) - D_\psi(\hat{\boldsymbol{\theta}}_I^{t+1}\|\boldsymbol{\theta}_I^{t+1}) - D_\psi(\boldsymbol{\theta}_I^{t+1}\|\hat{\boldsymbol{\theta}}_I^t)\Big]$$

$$= D_\psi(\sigma_i(I)\|\hat{\boldsymbol{\theta}}_I^t) - D_\psi(\sigma_i(I)\|\hat{\boldsymbol{\theta}}_I^{t+1}) + \frac{(\eta^{t+1})^2}{2}\|\hat{\boldsymbol{m}}_i^{t+1}(I) - \hat{\boldsymbol{m}}_i^t(I)\|_2^2 - D_\psi(\boldsymbol{\theta}_I^{t+1}\|\hat{\boldsymbol{\theta}}_I^t),$$

$$(49)$$

Multiplying both sides in Eq. (49) with $w^{t+1}\zeta_I = w^{t+1}(1 - \alpha_I)\pi_i^{\hat{\sigma}}(I)$, and applying Lemma D.7 to simplify the left-hand side of Eq. (49) we have

$$w^{t+1}\pi_i^{\hat{\sigma}}(I)\langle-\hat{\boldsymbol{v}}_i^{t+1}(I), \hat{\sigma}_i^{t+1}(I) - \hat{\sigma}_i(I)\rangle$$

$$\leq \frac{w^{t+1}\zeta_I}{\eta^{t+1}}\Big[D_\psi(\sigma_i(I)\|\hat{\boldsymbol{\theta}}_I^t) - D_\psi(\sigma_i(I)\|\hat{\boldsymbol{\theta}}_I^{t+1}) + \frac{(\eta^{t+1})^2}{2}\|\hat{\boldsymbol{m}}_i^{t+1}(I) - \hat{\boldsymbol{m}}_i^t(I)\|_2^2 - D_\psi(\boldsymbol{\theta}_I^{t+1}\|\hat{\boldsymbol{\theta}}_I^t)\Big], \qquad (50)$$

Then, taking summation of both sides in Eq. (50) over $t = [T-1]$, $I \in \mathcal{I}_i$, and applying Lemma D.7, we have

$$\sum_{t=0}^{T-1} w^{t+1} \langle \hat{\boldsymbol{\ell}}_i^{t+1}, \hat{\boldsymbol{x}}_i^{t+1} - \hat{\boldsymbol{x}}_i \rangle$$

$$\leq \sum_{t=0}^{T-1} \sum_{I \in \mathcal{I}_i} \frac{\zeta_I w^{t+1}}{\eta^{t+1}} \Big[ D_\psi(\sigma_i(I)||\hat{\boldsymbol{\theta}}_I^t) - D_\psi(\sigma_i(I)||\hat{\boldsymbol{\theta}}_I^{t+1}) - D_\psi(\boldsymbol{\theta}_I^{t+1}||\hat{\boldsymbol{\theta}}_I^t) + \frac{(\eta^{t+1})^2}{2} \|\hat{\boldsymbol{m}}_i^{t+1}(I) - \hat{\boldsymbol{m}}_i^t(I)\|_2^2 \Big]$$

$$\leq \sum_{I \in \mathcal{I}_i} \zeta_I \Big[ \sum_{t=0}^{T-1} \frac{w^{t+1}}{\eta^{t+1}} \Big( D_\psi(\sigma_i(I)||\hat{\boldsymbol{\theta}}_I^t) - D_\psi(\sigma_i(I)||\hat{\boldsymbol{\theta}}_I^{t+1}) \Big) + \sum_{t=0}^{T-1} \frac{(w^{t+1}\eta^{t+1})}{2} \|\hat{\boldsymbol{m}}_i^{t+1}(I) - \hat{\boldsymbol{m}}_i^t(I)\|_2^2 \Big]$$

$$= \sum_{I \in \mathcal{I}_i} \zeta_I \Big[ \sum_{t=1}^{T-1} \Big( \frac{w^{t+1}}{\eta^{t+1}} - \frac{w^t}{\eta^t} \Big) D_\psi(\sigma_i(I)||\hat{\boldsymbol{\theta}}_I^t) + \frac{w^1}{\eta^1} D_\psi(\sigma_i(I)||\hat{\boldsymbol{\theta}}_I^0) - \frac{w^T}{\eta^T} D_\psi(\sigma_i(I)||\hat{\boldsymbol{\theta}}_I^T) + \sum_{t=0}^{T-1} \frac{(w^{t+1}\eta^{t+1})}{2} \|\hat{\boldsymbol{m}}_i^{t+1}(I) - \hat{\boldsymbol{m}}_i^t(I)\|_2^2 \Big]$$

$$\leq \sum_{I \in \mathcal{I}_i} \zeta_I \Big[ \frac{w^1}{\eta^1} D_\psi(\sigma_i(I)||\hat{\boldsymbol{\theta}}_I^0) + \sum_{t=0}^{T-1} \frac{(w^{t+1}\eta^{t+1})}{2} \|\hat{\boldsymbol{m}}_i^{t+1}(I) - \hat{\boldsymbol{m}}_i^t(I)\|_2^2 \Big].$$

$$(51)$$

where the third inequality holds due to the non-increasing property of sequence $\{\frac{w^t}{\eta^t}\}_{t\geq 0}$ and the non-negativity of Bregman divergence $D_\psi(\sigma_i(I)||\hat{\boldsymbol{\theta}}_I^T)$. Then, according to the analysis in Lemma G.3, for weighted OOMD algorithm running with any (fixed) initialization $\hat{\boldsymbol{\theta}}_I^0$ and step sizes $\{\eta^t\}_{t\geq 0}$, another weighted OOMD algorithm running with the initialization $\hat{\boldsymbol{\theta}}_I^{0'} = \frac{\eta^{0'}}{\eta^0} \hat{\boldsymbol{\theta}}_I^0$ (for any $\eta^{0'} > 0$) and the sequence of step sizes $\{\eta^{t+1'} = \frac{\eta^{t'}}{\eta^t} \eta^{t+1}\}_{t\geq 0}$ outputs the same sequence-form strategy $\hat{\boldsymbol{x}}_i^{t+1}$. Then, replacing $\eta^{t+1}$ and $\hat{\boldsymbol{\theta}}_I^0$ in Eq. (51) with $\eta^{t+1'}$ and $\hat{\boldsymbol{\theta}}_I^{0'}$, we obtain that for any $\eta^{0'} > 0$:

$$\sum_{t=0}^{T-1} w^{t+1} \langle \hat{\boldsymbol{\ell}}_i^{t+1}, \hat{\boldsymbol{x}}_i^{t+1} - \hat{\boldsymbol{x}}_i \rangle$$

$$\leq \sum_{I \in \mathcal{I}_i} \zeta_I \Big[ \frac{w^1}{\eta^{1'}} D_\psi(\sigma_i(I)||\hat{\boldsymbol{\theta}}_I^{0'}) + \sum_{t=0}^{T-1} \frac{(w^{t+1}\eta^{t+1'})}{2} \|\hat{\boldsymbol{m}}_i^{t+1}(I) - \hat{\boldsymbol{m}}_i^t(I)\|_2^2 \Big]$$

$$\leq \sum_{I \in \mathcal{I}_i} \zeta_I \Big[ \frac{w^1(\|\sigma_i(I)\|_2^2 + \|\hat{\boldsymbol{\theta}}_I^{0'}\|_2^2)}{\eta^{1'}} + \sum_{t=0}^{T-1} \frac{(w^{t+1}\eta^{t+1'})}{2} \|\hat{\boldsymbol{m}}_i^{t+1}(I) - \hat{\boldsymbol{m}}_i^t(I)\|_2^2 \Big]$$

$$= \sum_{I \in \mathcal{I}_i} \zeta_I \Big[ \frac{w^1 \|\sigma_i(I)\|_2^2}{\eta^{1'}} + \frac{w^1 \|\hat{\boldsymbol{\theta}}_I^{0'}\|_2^2}{\eta^{1'}} + \sum_{t=0}^{T-1} \frac{(w^{t+1}\eta^{t+1'})}{2} \|\hat{\boldsymbol{m}}_i^{t+1}(I) - \hat{\boldsymbol{m}}_i^t(I)\|_2^2 \Big]$$

$$= \sum_{I \in \mathcal{I}_i} \zeta_I \Big[ \frac{\eta^0 w^1 \|\sigma_i(I)\|_2^2}{\eta^1 \eta^{0'}} + \frac{w^1 \eta^{0'} \|\hat{\boldsymbol{\theta}}_I^0\|_2^2}{\eta^1 \eta^0} + \sum_{t=0}^{T-1} \frac{(w^{t+1}\eta^{t+1}\eta^{0'})}{2\eta^0} \|\hat{\boldsymbol{m}}_i^{t+1}(I) - \hat{\boldsymbol{m}}_i^t(I)\|_2^2 \Big]$$

$$\leq \sum_{I \in \mathcal{I}_i} \zeta_I 2 \sqrt{\frac{\eta^0 w^1}{\eta^1} \Big[ \frac{w^1 \|\hat{\boldsymbol{\theta}}_I^0\|_2^2}{\eta^1 \eta^0} + \sum_{t=0}^{T-1} \frac{(w^{t+1}\eta^{t+1})}{2\eta^0} \|\hat{\boldsymbol{m}}_i^{t+1}(I) - \hat{\boldsymbol{m}}_i^t(I)\|_2^2 \Big]} \quad (52)$$

$$= \sum_{I \in \mathcal{I}_i} \zeta_I 2 \sqrt{\frac{w^1}{\eta^1} \Big[ \frac{w^1 \|\hat{\boldsymbol{\theta}}_I^0\|_2^2}{\eta^1} + \sum_{t=0}^{T-1} \frac{(w^{t+1}\eta^{t+1})}{2} \|\hat{\boldsymbol{m}}_i^{t+1}(I) - \hat{\boldsymbol{m}}_i^t(I)\|_2^2 \Big]} \quad (53)$$

where the second inequality holds due to the mean-value inequality $\|\boldsymbol{a}+\boldsymbol{b}\|_2^2 \leq 2\|\boldsymbol{a}\|_2^2 + 2\|\boldsymbol{b}\|_2^2$, the second equality holds due to the fact that $\hat{\boldsymbol{\theta}}_I^{0'} = \frac{\eta^{0'}}{\eta^0} \hat{\boldsymbol{\theta}}_I^0$ and $\frac{\eta^{t+1'}}{\eta^{t+1}} = \frac{\eta^{t'}}{\eta^t} = \frac{\eta^{0'}}{\eta^0}$, the last inequality due to $\|\hat{\boldsymbol{\theta}}_I^0\|_2 \leq \|\hat{\boldsymbol{\theta}}_I^0\|_1 = 1$ and the basic inequality $\frac{a}{\eta} + b\eta \leq 2\sqrt{ab}$ for any $\eta > 0$. The whole proof is completed. $\qquad \square$

*Table 3.* Sizes of different extensive-form games, where #Histories is the total number of histories in the game tree, #Infosets the number of all information sets, #Terminal histories the number of leaf nodes, #Depth the depth of game tree, and #Max size of infosets is the largest number of histories contained at an infoset.

| Game | #Histories | #Infosets | #Terminal histories | Depth | Max size of infosets |
|---|---|---|---|---|---|
| Kuhn Poker | 58 | 12 | 30 | 6 | 2 |
| Leduc Poker | 9,457 | 936 | 5,520 | 12 | 5 |
| Liar's Dice (4) | 8,181 | 1,024 | 4,080 | 12 | 4 |
| Liar's Dice (5) | 51,181 | 5,120 | 25,575 | 14 | 5 |
| Liar's Dice (6) | 294,883 | 24,576 | 147,420 | 16 | 6 |
| Goofspiel (4) | 1,077 | 162 | 576 | 7 | 14 |
| Goofspiel (5) | 26,931 | 2,124 | 14,400 | 9 | 46 |
| Goofspiel (6) | 969,523 | 34,482 | 518,400 | 11 | 230 |
| Battleship (3) | 732,607 | 81,027 | 552,132 | 9 | 7 |
| HUNL Subgame (3) | 398,112,843 | 69,184 | 261,126,360 | 10 | 1,980 |
| HUNL Subgame (4) | 244,005,483 | 43,240 | 158,388,120 | 8 | 1,980 |

## I. Details of Extensive-Form Games (EFGs)

**Kuhn Poker** (Kuhn, 1950) is a simplified two-player zero-sum poker game with a three-card deck and one chance to bet for each player. **Leduc Poker** (Southey et al., 2005) is a larger game with a 6-card deck and two betting rounds. In **Liar's Dice (x)** ($x=4, 5$) (Lisý et al., 2015), each player gets an $x$-sided dice, which they will roll at the beginning of each hand and take turn placing bets on the outcome. ***Goofspiel (x)*** ($x=4, 5$) (Ross, 1971) is a card game where each player has $x$ cards and their goal is to score points by bidding simultaneously in $x$ rounds. In **Battleship (3)** (Farina et al., 2019d), each player conceals a $1 \times 2$ ship on her own $2 \times x$ grids and then alternates turns launching three attacks against the opponent's vessel. **HUNL Subgame (x)** ($x=3, 4$) refers to a heads-up no-limit Texas hold'em (HUNL) subgame constructed by the superhuman poker agent Libratus (Brown & Sandholm, 2019). The sizes of different Extensive-Form Games (EFGs) are shown in Table 3. More details of the rule of these EFGs can be found in (Xu et al., 2024, Appendix B).

## J. Additional Experiments

*Table 4.* Hyperparameters used in RTPCFR$^+$(fine-tuned).

| | Kuhn Poker | Leduc Poker | Battleship (3) | Liar's Dice (4) | Liar's Dice (5) |
|---|---|---|---|---|---|
| $\mu$ | 0.005 | 0.001 | 0.00005 | 0.001 | 0.0005 |
| $T_u$ | 50 | 200 | 10 | 10 | 50 |
| | Liar's Dice (6) | Goofspiel (4) | Goofspiel (5) | Goofspiel (6) | |
| $\mu$ | 0.0001 | 0.0001 | 0.005 | 0.005 | |
| $T_u$ | 200 | 200 | 100 | 500 | |

*Table 5.* Hyperparameters used in RTPDCFR$^+$(fine-tuned).

| | Kuhn Poker | Leduc Poker | Battleship (3) | Liar's Dice (4) | Liar's Dice (5) |
|---|---|---|---|---|---|
| $\mu$ | 0.1 | 0.0005 | 0.005 | 0.00005 | 0.0001 |
| $T_u$ | 10 | 100 | 10 | 10 | 50 |
| $\alpha$ | 1.3 | 2.5 | 2.8 | 1.5 | 2.0 |
| | Liar's Dice (6) | Goofspiel (4) | Goofspiel (5) | Goofspiel (6) | |
| $\mu$ | 0.00005 | 0.01 | 0.01 | 0.005 | |
| $T_u$ | 500 | 10 | 50 | 500 | |
| $\alpha$ | 2.8 | 3.0 | 2.8 | 3.0 | |

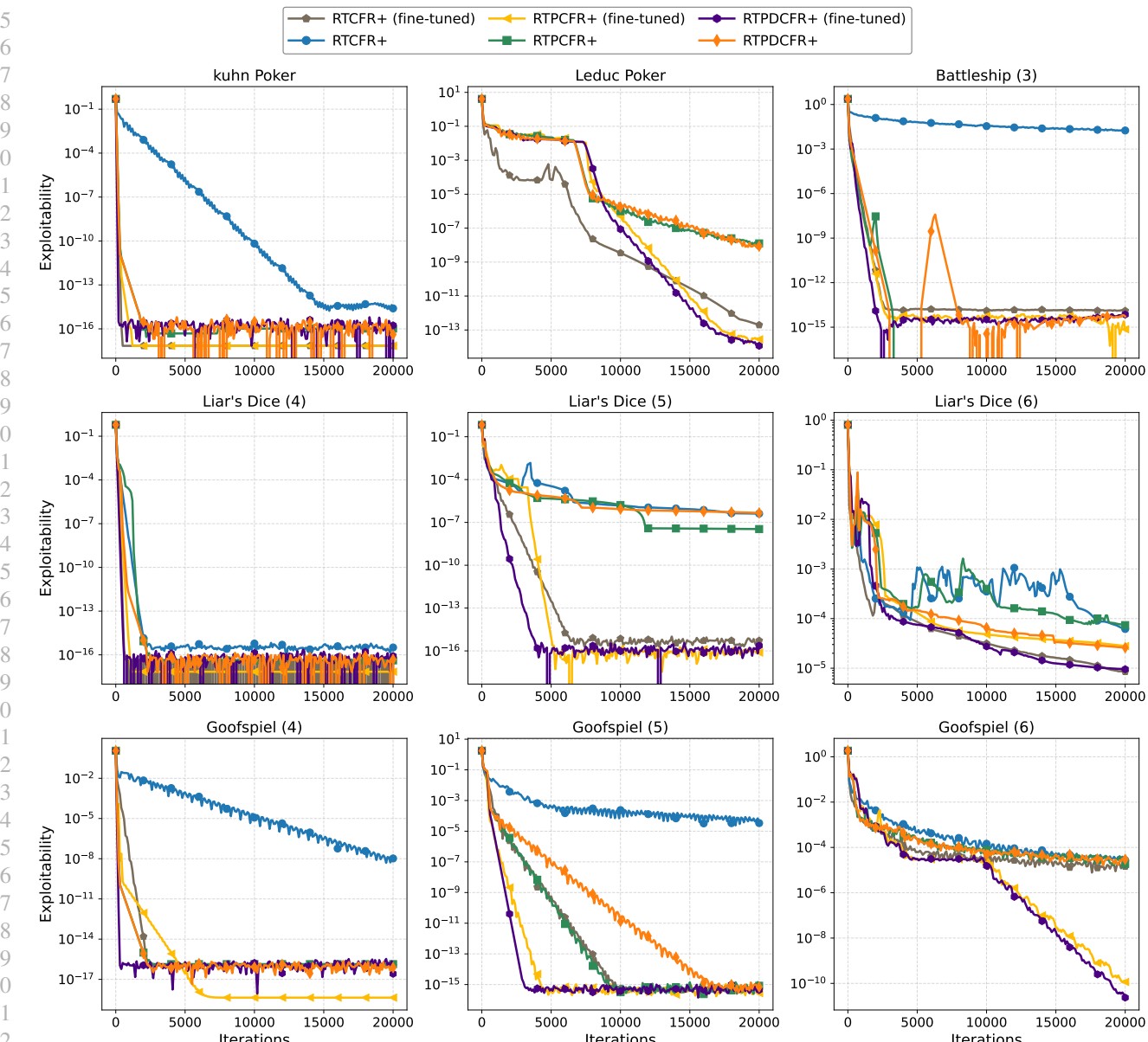

*Figure 2.* Last-iterate convergence of different reward transformation-based CFR methods in EFGs.

**Hyperparameter Setting of Baselines in EFGs.** For OMWU and OGDA, we set the step size $\eta$ to 0.5 and 0.1, respectively. For Reg-CFR, we use the same hyperparameters as in their original paper (Liu et al., 2023, Appendix B). For RTCFR$^+$ (Meng et al., 2025a), we set the initial values of $\gamma$ and $\mu$ as 1e−10 and 1e−3, respectively, and set the number of iterations $T_u$ required to update $\gamma$ and $r$ to 100. For our RTPDCFR$^+$, the hyperparameter $\alpha$ in discounting factor $\frac{t^\alpha}{1+t^\alpha}$ is set as $\alpha = 2.8$, the initial values $\gamma = 1e-10$, $\mu = 5e-5$, and the number of iterations is set as $T_u = 100$ to update $r$. The optimal hyperparameters used in RTPCFR$^+$ and RTPDCFR$^+$ (i.e. corresponding to RTPCFR$^+$ (fine-tuned) and RTPDCFR$^+$ (fine-tuned) respectively) over nine small-to-medium-size EFG benchmarks are shown in Tables 4-5.

**Empirical Last-Iterate Convergence Rates of different RT-based CFR Algorithms**. We compare the last-iterate convergence of three reward transformation-based algorithms: RTCFR$^+$ (Meng et al., 2025a), RTPCFR$^+$, RTPDCFR$^+$, as well their fine-tuned versions in Figure 2. We can see that: **(1)** Our proposed RTPCFR$^+$ and RTPDCFR$^+$ (without fine-tuning) can achieve faster convergence than the latest RTPCFR$^+$ (without fine-tuning) algorithm in most EFG benchmarks (expect Leduc Poker), demonstrating the benefits of optimism and discounting technique in solving EFGs. **(2)** RTPCFR$^+$ (fine-tuned) and RTPDCFR$^+$ (fine-tuned) also outperform RTCFR$^+$ (fine-tuned) in most EFGs. Specifically, RTPDCFR$^+$ (fine-tuned) yields faster convergence rates and lower exploitabilities in all nine EFG benchmarks. RTPCFR$^+$ (fine-tuned)

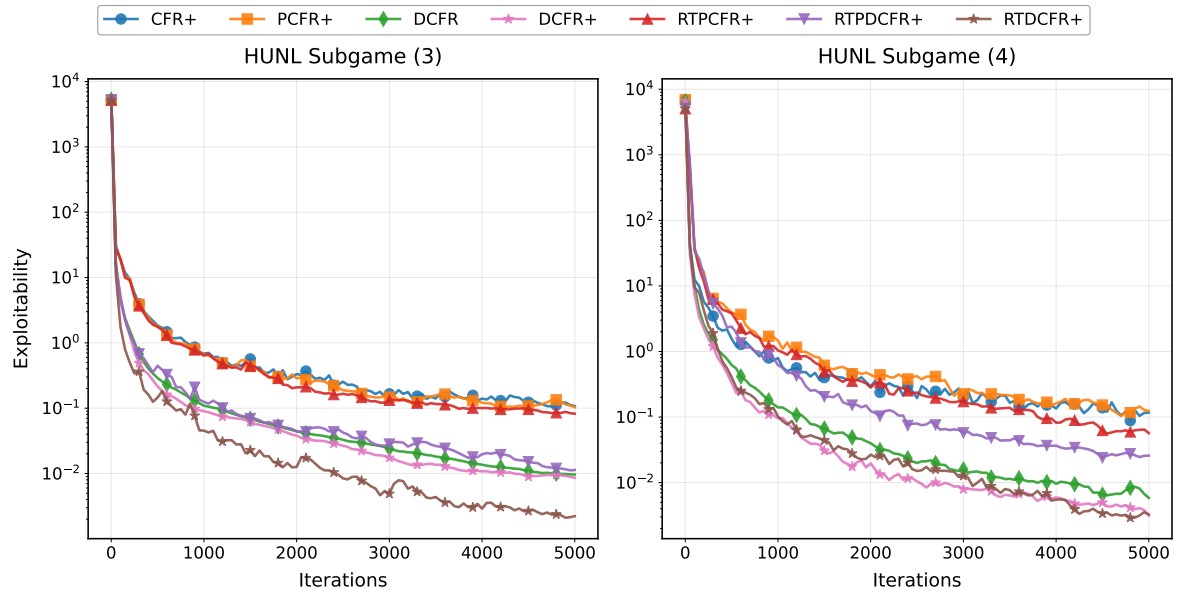

*Figure 3.* Last-iterate convergence of different CFR algorithms in large-size EFGs.

enjoys faster convergence rates and lower exploitabilities in 8 EFG benchmarks, except Liar's Dice (6). In particular, RTPCFR$^+$ (fine-tuned) obtains much lower exploitability than RTCFR$^+$ (fine-tuned) in Goofspiel (4). **(3)** For the comparison between RTPCFR$^+$ and RTPDCFR$^+$, RTPDCFR$^+$ can achieve lower exploitabilities in medium-size EFG benchmarks, like Liar's Dice (6) and Goofspiel (6), revealing the potential of discounting technique in solving large-scale EFGs. RTPDCFR$^+$ (fine-tuned) also obtains lower exploitabilities in these two EFGs than the fine-tuned version of RTPCFR$^+$. However, in Goofspiel (4), RTPCFR$^+$ (fine-tuned) attains a much lower exploitability than RTPDCFR$^+$ (fine-tuned). This indicates that, RTPCFR$^+$ seemingly has an edge in solving small-sized EFGs.

**Last-Iterate Convergence of Different CFR Algorithms in Large-size Games (i.e. Subgame (3) & (4)).** Previous work (Meng et al., 2025a, Figs 6-7) has shown that, the last-iterate convergence performance of the state-of-the-art DCFR in Subgame is almost identical to the average-iterate convergence performance. Therefore, we conduct experiments on large-size EFG benchmarks, HUNL Subgame (3) and Subgame (4) (Brown & Sandholm, 2019), to compare the empirical last-iterate convergence of different CFR algorithms. The experimental results are shown in Figure 3. We can observe that: **(1)** In such large-scale EFGs, the previous state-of-the-art DCFR (Brown & Sandholm, 2019) still achieves better faster convergence rates and lower exploitabilities than our proposed RTPCFR$^+$ algorithm (with $\mu = 0.005, T_u = 1000$). **(2)** Our proposed RTPDCFR$^+$ algorithm (with $\alpha = 1.4, \mu = 0.0005, T_u = 1000$) can achieve comparable result with DCFR in Subgame (3), but performs worse than DCFR in Subgame (4). **(3)** However, our proposed RTDCFR$^+$ (with $\alpha = 1.3, \mu = 0.0005, T_u = 1000$ in Subgame (3) and with $\alpha = 1.3, \mu = 0.01, T_u = 1000$ in Subgame (4)), the non-predictive version of RTPDCFR$^+$, yields lower exploitabilities than DCFR/DCFR$^+$ in both Subgame (3) and Subgame (4). Besides, RTDCFR$^+$ enjoys a faster convergence rate than DCFR/DCFR$^+$ in Subgame (3). Such performance demonstrates the strengths of discounting technique in solving large-scale EFGs, and somewhat indicates that optimism technique doesn't help much in solving large-size EFGs.

**Comparison with Average-Iterate Convergence of CFR algorithms.** We compare the last-iterate convergence of our RTPCFR$^+$ with the average-iterate convergence of traditional CFR algorithms in EFGs. The experimental results are shown in Figure 4. We can observe that: **(1)** In most EFG benchmarks (except Liar's Dice (6) and Goofspiel (6)), our RTPCFR$^+$ achieves significantly faster convergence rates to the NEs of original EFGs, than the average-iterate convergence rate of the state-of-the-art PCFR$^+$ algorithm (Farina et al., 2021b). In Kuhn Poker, Battleship (3), Liar's Dice (4), Goofspiel (4), and Goofspiel (5) these five EFG settings, our RTPCFR$^+$ achieves at least five orders of magnitude faster convergence rate than the average-iterate convergence rate of PCFR$^+$. **(2)** In Liar's Dice (6), both PCFR$^+$ and DCFR outperform our RTPCFR$^+$, and in Goofspiel (6) PCFR$^+$ converges faster than RTPCFR$^+$. Nevertheless, the fine-tuned version of RTPCFR$^+$, i.e. RTPCFR$^+$ (fine-tuned), achieves faster convergence rate than the average-iterate PCFR$^+$. **(3)** RTPCFR$^+$ (fine-tuned) can yield the best last-iterate convergence performance than the average-iterate convergence of all traditional CFR algorithms in all EFG settings, except that in Liar's Dice (6) DCFR's average-iterate strategy attains the fastest convergence rate. This implies that, even under the reward-transformation framework, PCFR$^+$ may not be able to perform best at large-size EFGs.

**Hyperparameter Sensitivity Analysis for RTPCFR$^+$ and RTPDCFR$^+$.** We conduct more experiments to investigate the convergence performance of RTPCFR$^+$ and RTPDCFR$^+$ under various hyperparameter settings. We set $\mu \in \{5e-1, 5e-2, 5e-3, 5e-4\}$ and $T_u \in \{10, 50, 100, 200, 500\}$ for both RTPCFR$^+$ and RTPDCFR$^+$. The experimental results for RTPCFR$^+$ are shown in Figures 5-8, and the experimental results for RTPDCFR$^+$ (with $\alpha = 2.8$) are shown in Figures 9-12. We can observe that: **(1)** Under the same $T_u$, with the decrease of $\mu$, RTPCFR$^+$ and RTPCFR$^+$ always converge to the lower level of exploitability. **(2)** Under the same regularization coefficient $\mu$, with the increase of $T_u$, both RTPCFR$^+$ and RTPDCFR$^+$ always converge slower to the lowest level of exploitability (especially in Kuhn Poker, Liar's Dice (4), Goofspiel (4) these three EFG benchmarks). **(3)** In the medium-size EFG benchmarks (like Goofspiel (5), Goofspiel (6) and Liar's Dice (6)), when $\mu$ is small enough (e.g. $\mu \le 5 \times 10^{-4}$), then for any choice of $T_u$, the performance of RTPCFR$^+$/RTPDCFR$^+$ remains truly similar. **(4)** When the initialized value of $\mu$ is not small enough (e.g. $\mu \ge 5 \times 10^{-2}$), then in EFG setting, RTPCFR$^+$ (e.g. Liar's Dice (2) in Figure 5) and RTPDCFR$^+$ (e.g. Liar's Dice (2) in Figure 9) seems unable to converge whatever value $T_u$ takes. This is also consistent with our theoretical analysis, since when $\mu$ is large, simultaneously decreasing $\mu$ and $\gamma$ possibly could not guarantee that $\mu$ decreases faster than $\gamma$ (i.e. $\mu \le \gamma$).

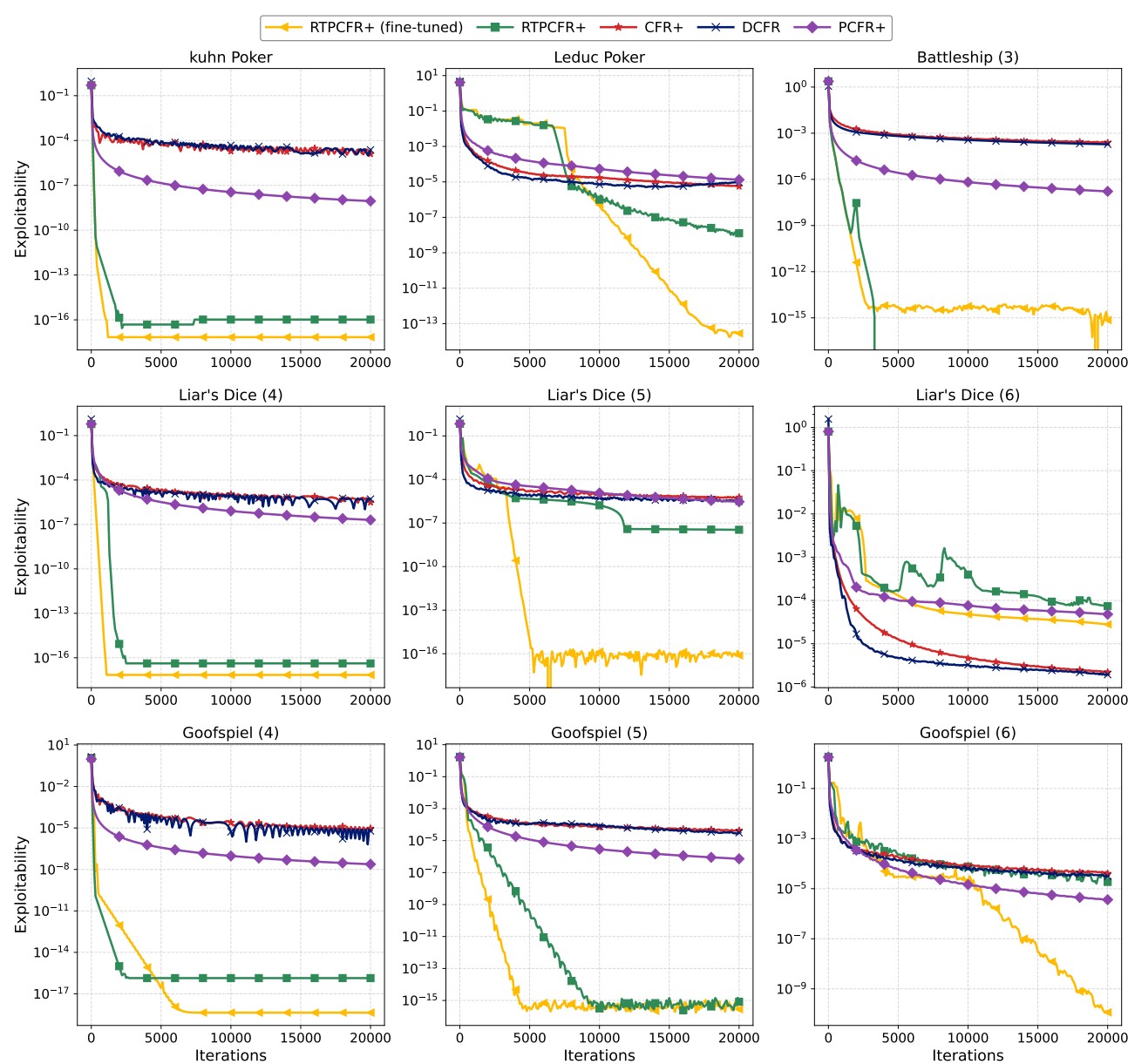

*Figure 4.* The comparison between last-iterate convergence of RTPCFR+ and average-iterate convergence of CFR variants in EFGs.

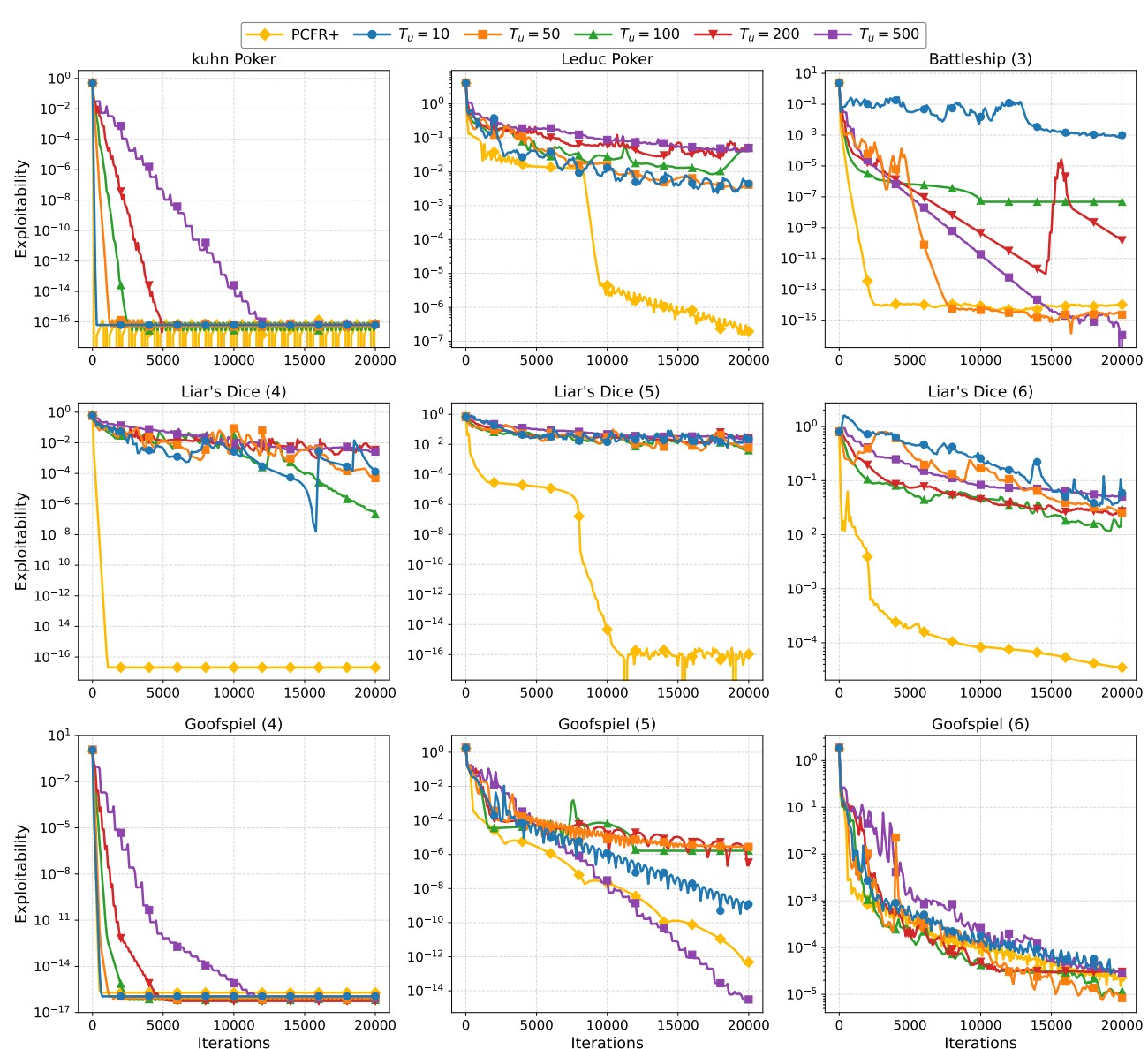

*Figure 5.* Empirical last-iterate convergence of RTPCFR$^+$ with $\mu = 5 \times 10^{-2}$.

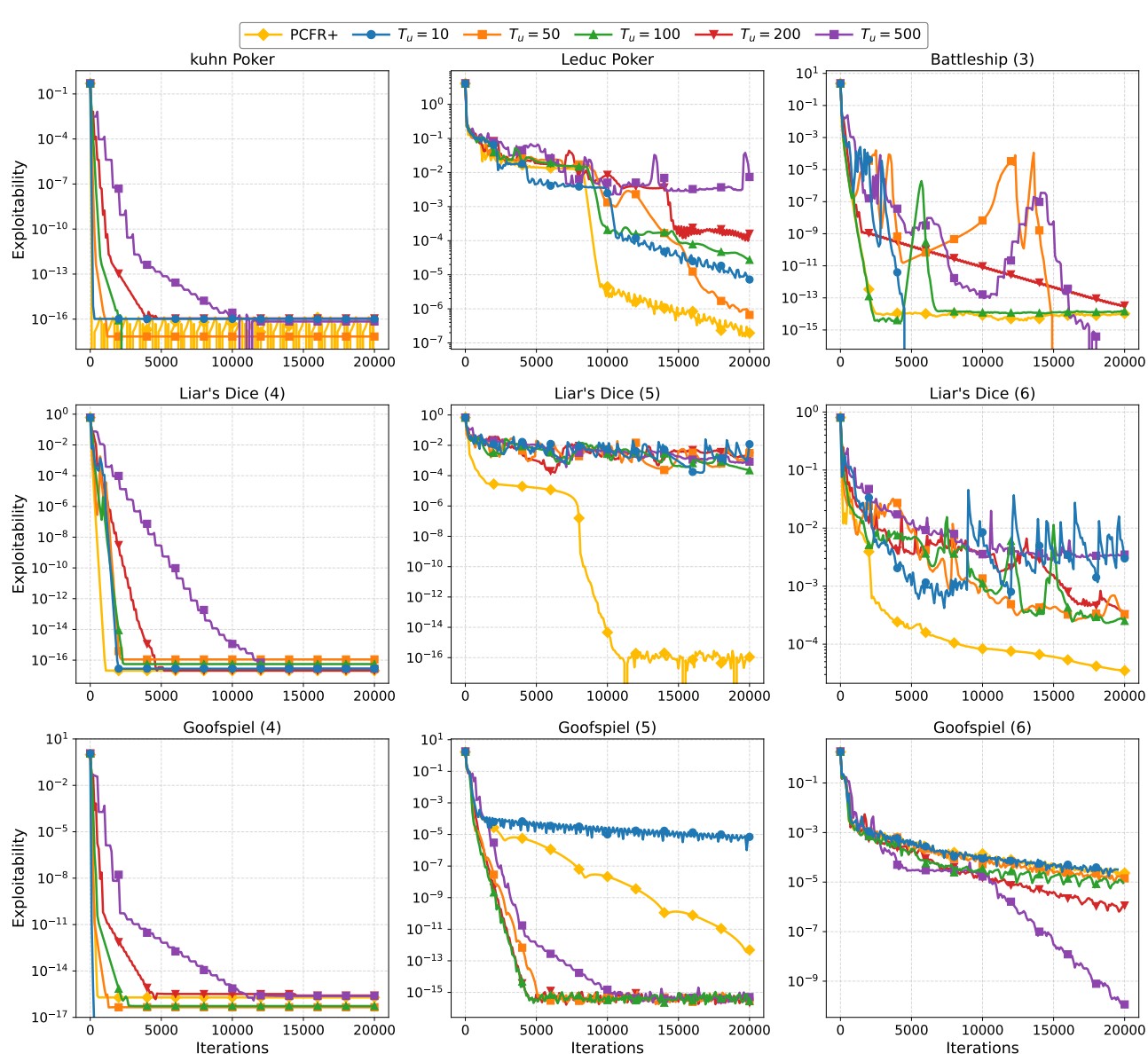

*Figure 6.* Empirical last-iterate convergence of RTPCFR$^+$ with $\mu = 5 \times 10^{-3}$.

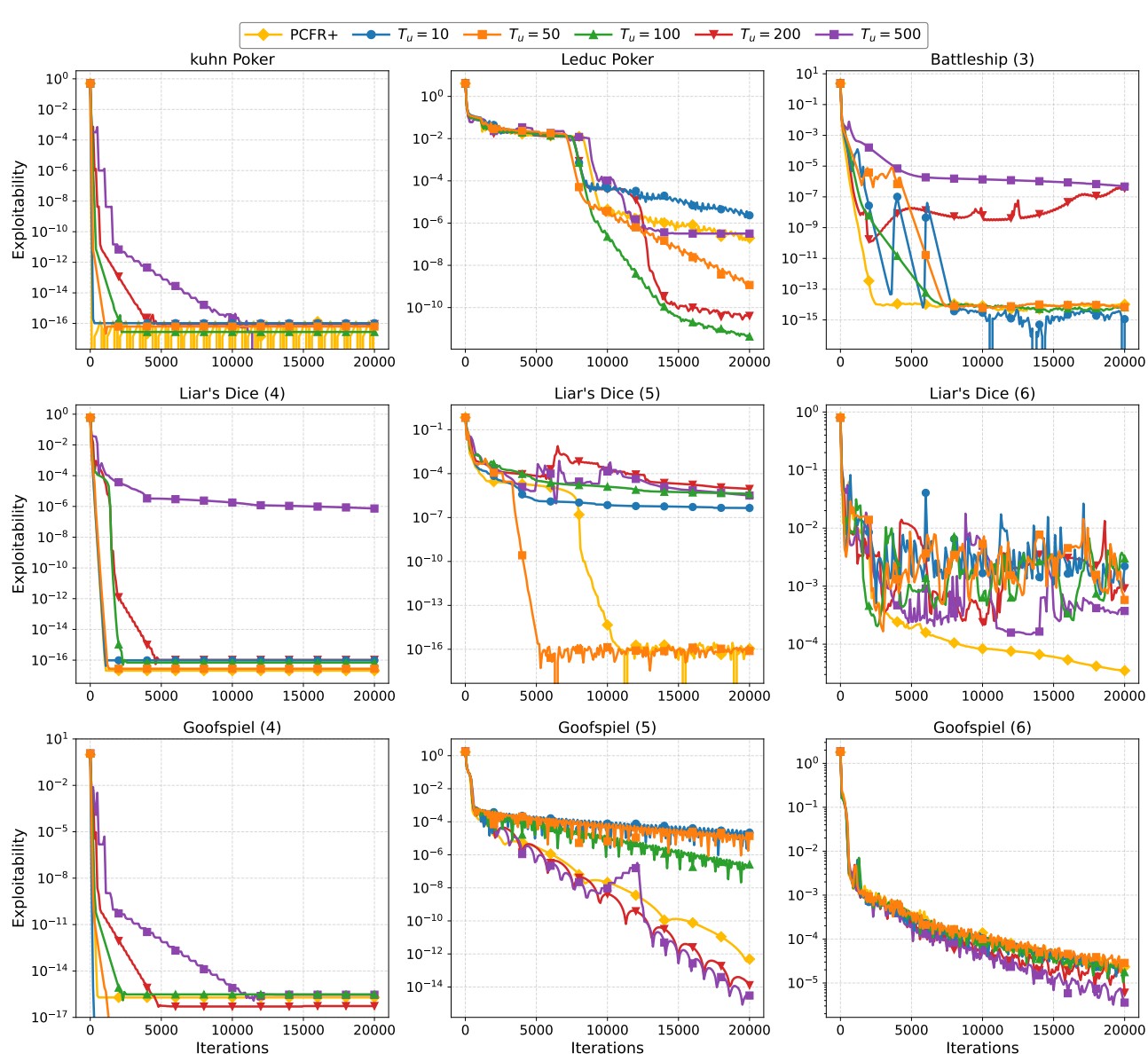

*Figure 7.* Empirical last-iterate convergence of RTPCFR$^+$ with $\mu = 5 \times 10^{-4}$.

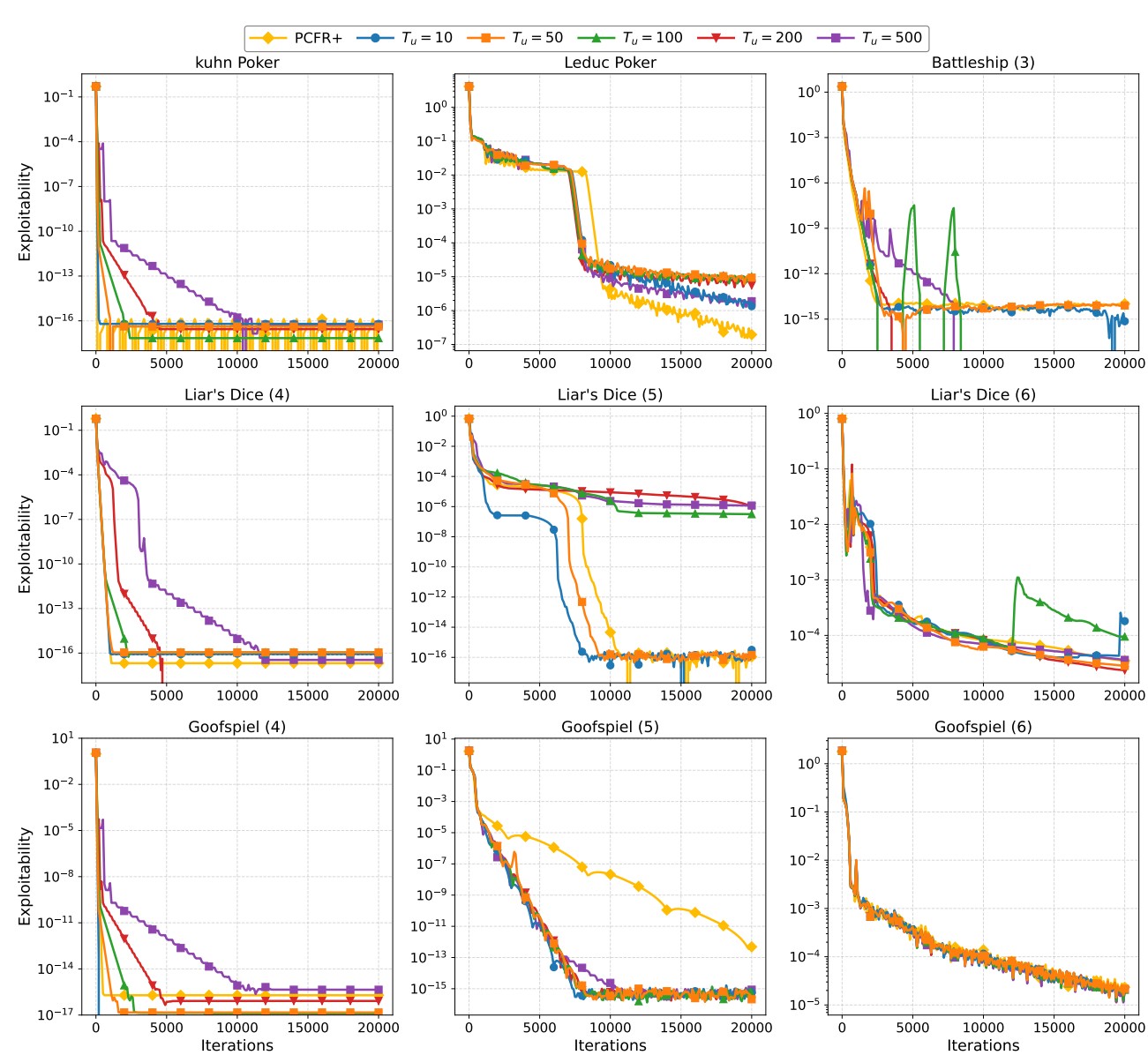

*Figure 8.* Empirical last-iterate convergence of RTPCFR$^+$ with $\mu = 5 \times 10^{-5}$.

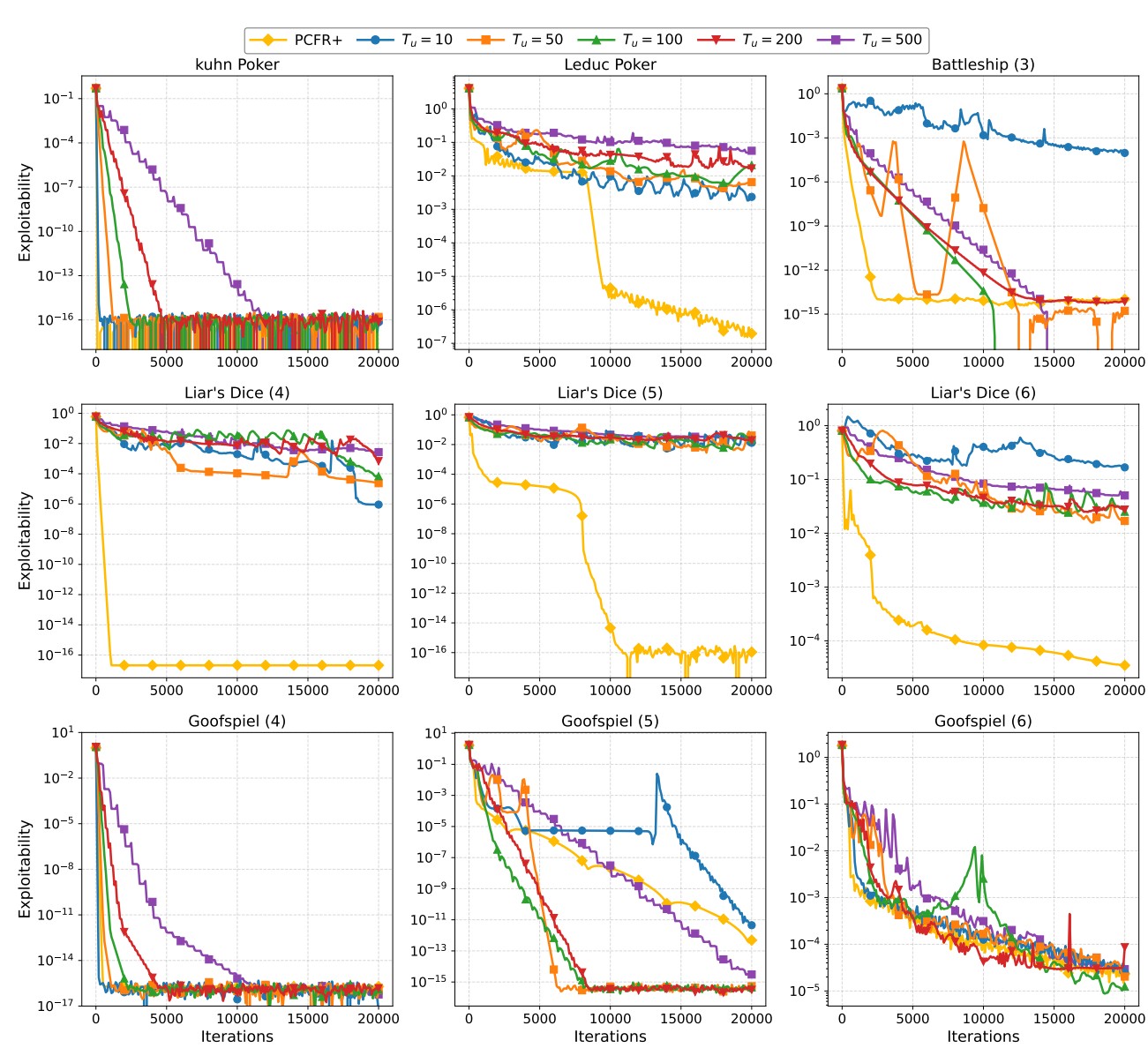

*Figure 9.* Empirical last-iterate convergence of RTPDCFR$^+$ with $\mu = 5 \times 10^{-2}$.

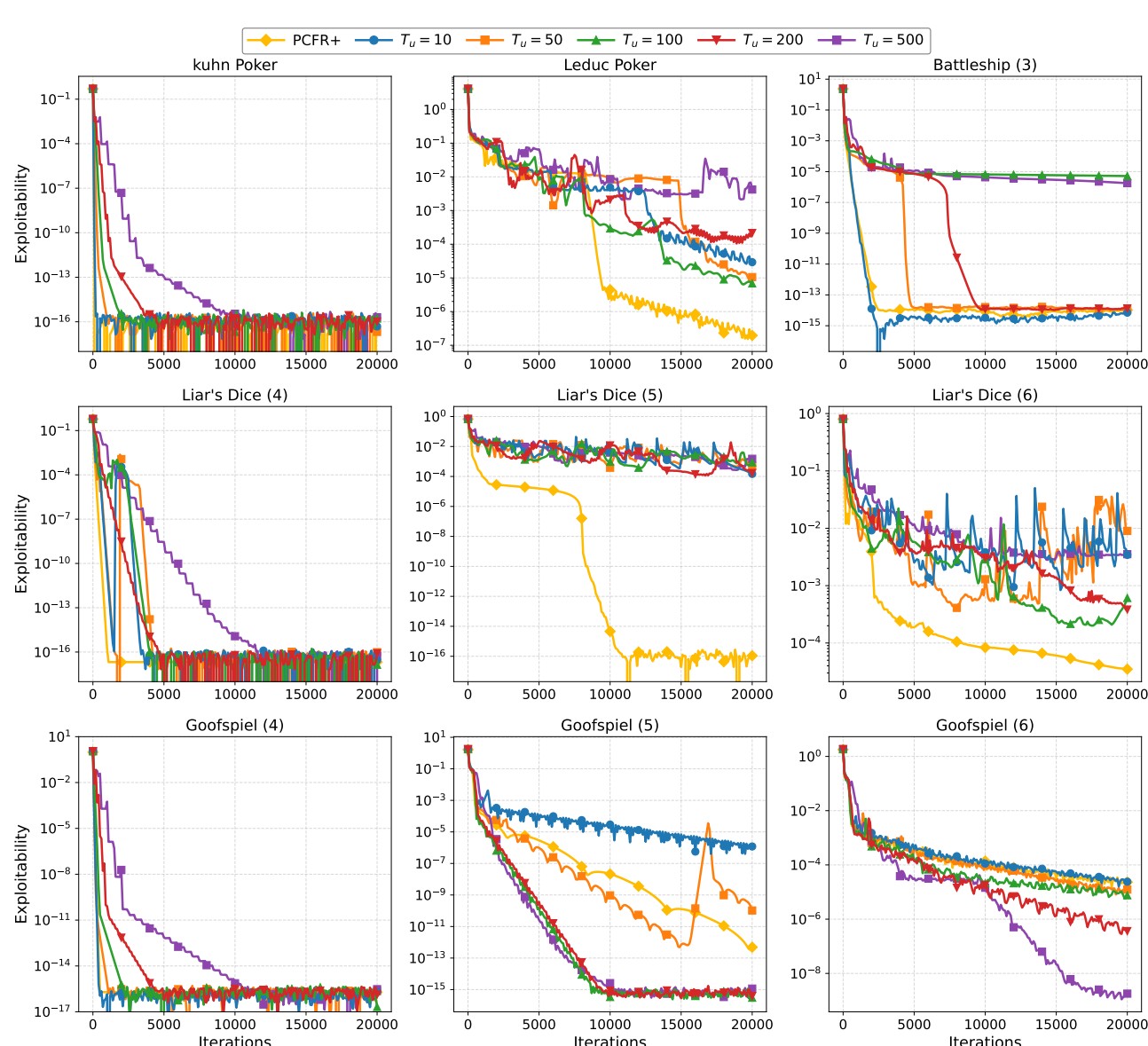

*Figure 10.* Empirical last-iterate convergence of RTPDCFR$^+$ with $\mu = 5 \times 10^{-3}$.

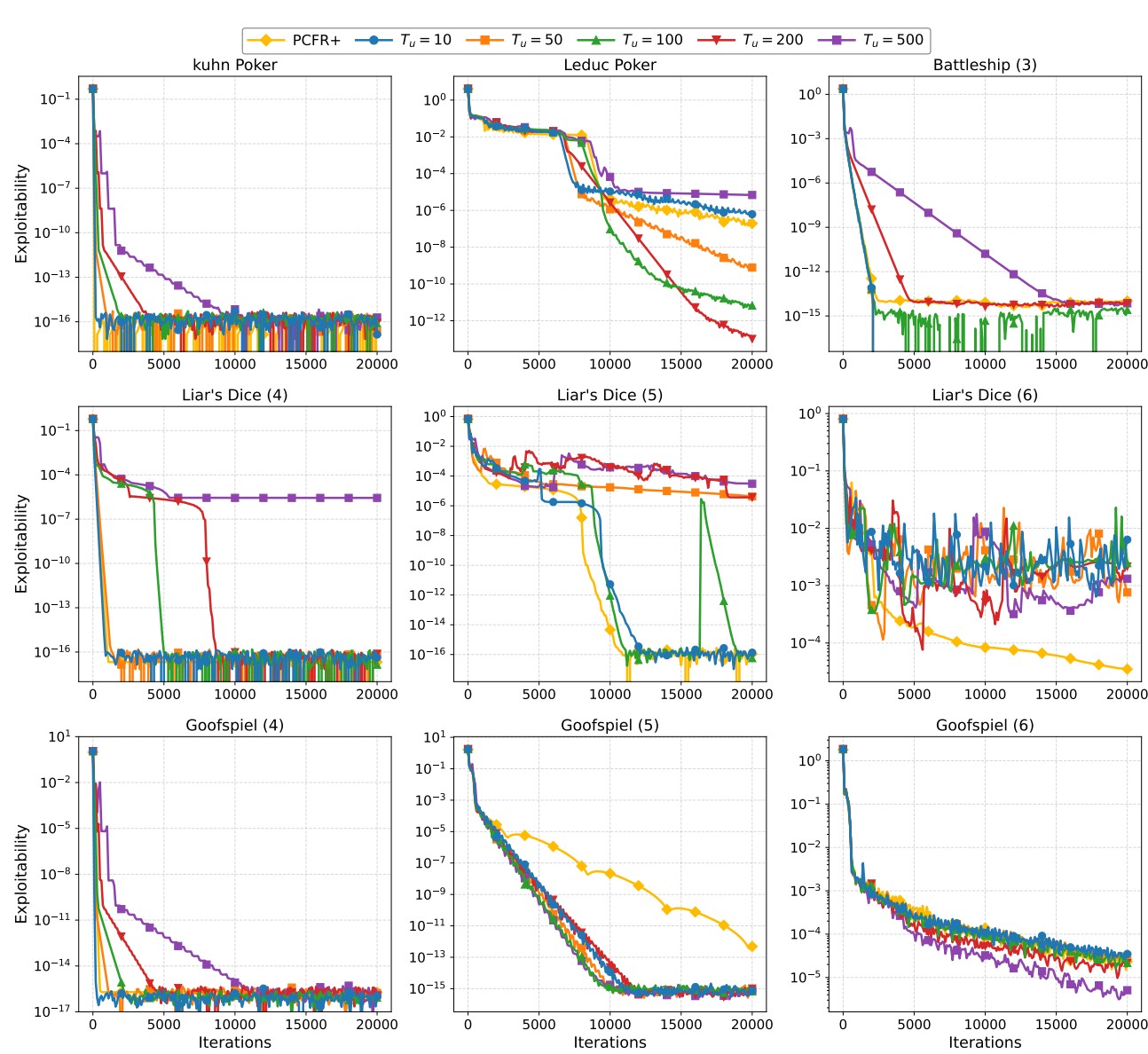

*Figure 11.* Empirical last-iterate convergence of RTPDCFR$^+$ with $\mu = 5 \times 10^{-4}$.

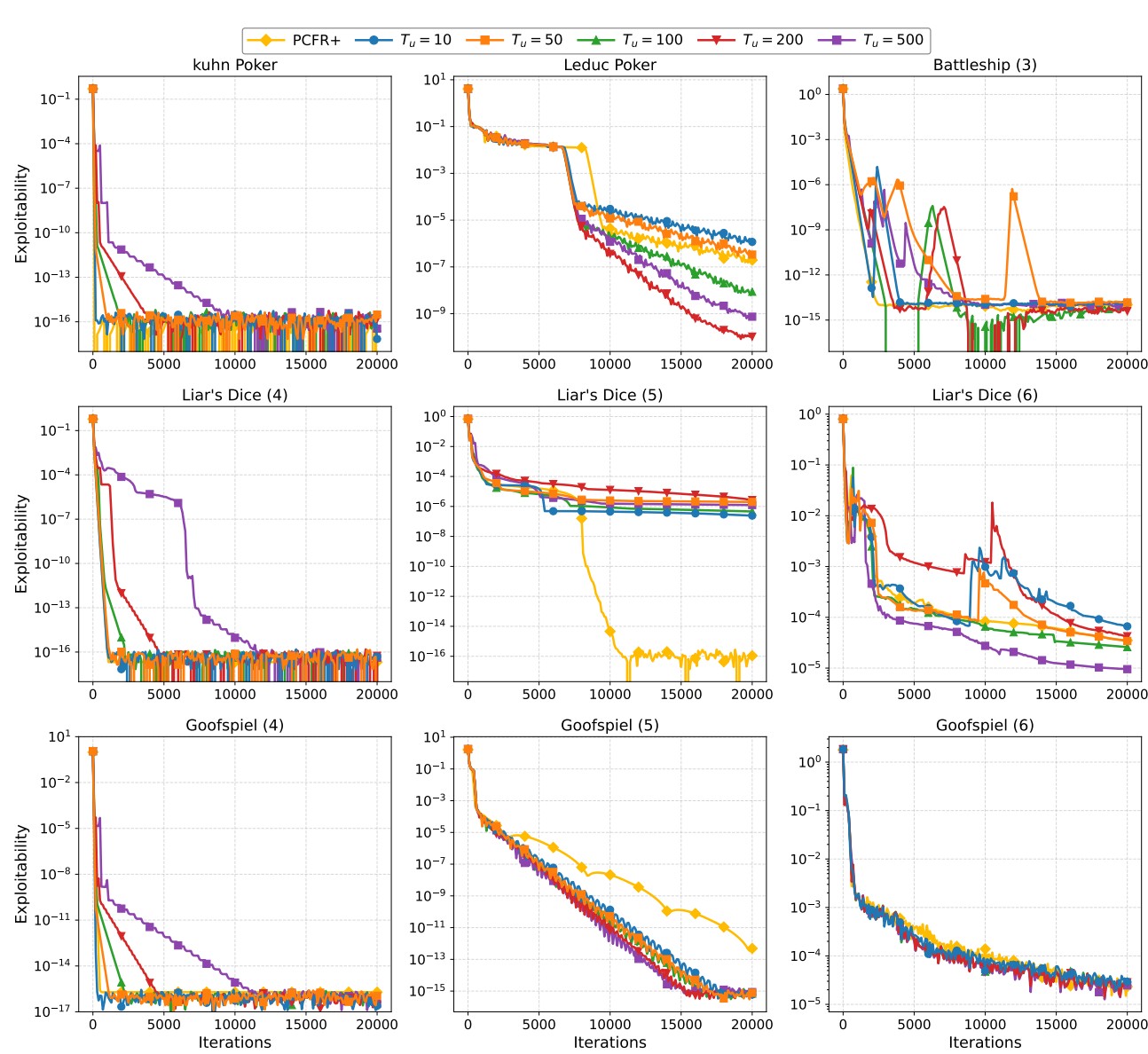

*Figure 12.* Empirical last-iterate convergence of RTPDCFR$^+$ with $\mu = 5 \times 10^{-5}$.

