# OpenReview forum: "Refined Last-Iterate Convergence Analysis with Optimism in Solving Extensive-Form Games"
_ICML.cc/2026/Conference — Submitted to ICML 2026_

### Official Review · Reviewer_ZVd6 · 2026-02-18

**Soundness:** 3
**Presentation:** 3
**Significance:** 3
**Originality:** 3
**Overall Recommendation:** 4
**Confidence:** 3

**Summary:**

The paper introduces two new algorithms (RTPCFR+ and RTPDCFR+) for solving extensive-form games (EFGs) under the reward transformation (RT) framework (from literature). RTPCFR+ is an optimistic/predictive variant of CFR+ casted in the RT setting, which solves a sequence of perturbed & regualarized EFGs and leverages the fact that the equilibria of this perturbed games converge to the set of NEs of the original EFG [Bernasconi, 2024]. RTPDCFR+ additionally incorporates regret discouting (in the spirit of [Brown & Sandholm, 2019]), downweighting the regret contributions from ealier time steps.

The authors establish asymptotic last-iterate convergence guarantees for their algorithms as well as average-iterate convergence results. The results come with the additonal feature of step-size invariance (i.e. hold for any constant step size and any initialization).

**Compliance With Llm Reviewing Policy:**

Affirmed.

**Key Questions For Authors:**

-	Step-size invariance vs. optimality (Thm. 5.2). Theorem 5.2 yields a suboptimal convergence rate under constant step sizes. Can the authors clarify whether this reflects an inherent trade-off—i.e., does step-size invariance (for all constant η) fundamentally limit what guarantees can be proven or whether it is primarily an artifact of the analysis technique? Concretely: is there evidence (lower bound or impossibility result) that achieving the faster convergence rate requires either step-size schedules or problem-dependent tuning?

-	Use of E.3 in average-iterate proof.
 The last-iterate proof on the perturbed game invokes E.3. Can the authors explain why the same line of reasoning as for gamma, \mu > 0 cannot (or can) be applied to the average-iterate convergence proof when \gamma=\mu=0? Is the bridge here essentially coming from a result like Bernasconi (2024), or is there a more direct argument specific to this framework?

-	Is the predictive update essential for step-size invariance?
 Is the predictive/optimistic update structurally necessary to obtain step-size invariant guarantees, or could similar invariance be proven for the non-predictive RT algorithms with an adapted analysis?

-	Conjecture on page 6:  The paper conjectures that step-size invariance is the greatest to achieve fast average-iterate convergence. Can the authors elaborate on that and provide more intuition?

-	Is step-size invariance a better predictor of empirical robustness across a range of untuned learning rates compared to algorithms that do not posess the step-size invariance property

**Limitations:**

-	Make explicit why Eq. (1269) is in O(1)
-	In the proof of Thm. 6.1, briefly restate the definition of zeta_t to simplify the proof reading
-	Shift at least one key discounted-vs-undiscounted comparison from the appendix to the main text

**Strengths And Weaknesses:**

Strengths:

-	Theoretical support. All proposed algorithms are accompanied by formal statements and proofs

-	Clarity and structure. The paper is well organized and does a good job positioning itself relative to prior RT and CFR-style frameworks, including careful discussion of differences to non-predictive variants.

 -      Step-size invariance is conceptually valuable. Even when suboptimal in rate (e.g., Theorem 5.2 gives O(1/\sqrt{T}) versus the optimal rates possible O(1) achievabe under adaptive learning rates e.g. [Liu et al., 2023]), these results help explain why the methods can work robustly in practice without delicate hyperparameter choices.

-	Novel techniques: Theorem 5.1 (RTPCFR+ last-iterate guarantee) uses a predictive-update-driven argument that differs from the non-predictive RT analogue (e.g., RTCFR+). They show that the computed output strategies obtainable via optimistic OMD is asymptotically equivalent independent of the learning rate and initialization.  (Other results build on that)

Weaknesses

-	Scope. As written, the main development relies on formulations like Eq. (6) (bilinear saddle point), which typically corresponds to two-player zero-sum EFGs (or closely related settings). If this is the intended scope, the paper could be simplified by stating and framing the main results explicitly for the two-player zero-sum case. Relatedly, some phrasing in the Introduction/Preliminaries may mislead unfamiliar readers. For instance, in the “Online Learning over Sequence-form strategies” paragraph, convergence of the average strategy to Nash equilibria is stated in a way that is correct for zero-sum but not for general-sum games unless moving to correlated/CCE-type notions

-	Motivation of Discounting. The paper introduces RTPDCFR+ but does not clearly explain what the discount factor buys over the undiscounted version—either theoretically (comparision guarantees) or empirically. From the plots (e.g., around Fig. 28), it is difficult to see a consistent advantage of the discounted variant.

-	Minor typos. especially the typo“inforsets” appears frequently

---

> ### Author Rebuttal · Authors · 2026-03-28
>
> **Q1. Scope. If Eq. (6) (bilinear saddle point) is the intended scope, the paper could be simplified by stating and framing the main results explicitly for the two-player zero-sum case.**\
> A1: Thanks for this suggestion, we will simplify the notations and frame the main results explicitly for the two-player zero-sum EFG in the revised version.
>
> **Q2. Motivation of Discounting. The paper introduces RTPDCFR+ but does not clearly explain what the discount factor buys over the undiscounted version.**\
> A2: Thanks for pointing this out. The reason for introducing discounting technique into RTPCFR+ is that, the state-of-the-art PCFR+ algorithm fails to outperform DCFR in large-scale EFGs (e.g., Subgame 3 in Heads-up Limit Texas Hold'em) in terms of average-iterate convergence. Given that discounting techniques have exhibited superior performance particularly in such large-scale settings, we investigate whether RTPDCFR+ (i.e., RTPCFR+ with discounting) can achieve theoretical last-iterate convergence and empirically superior performance.
>
> **Q3. Concretely: is there evidence (lower bound or impossibility result) that achieving the faster convergence rate requires either step-size schedules or problem-dependent tuning?**\
> A3: Yes, as discussed in Remark 5.5, under the counterfactual regret minimization framework, obtaining theoretically faster average-iterate convergence rate (e.g. $O(1/T)$ in [2, Sect J] and $O(1/T^{\frac{3}{4}})$ in [3, Thm 5.6]) for EFG always requires a small step size (like $\eta \leq 1/P$ with $P$ as the number of total infosets in [2]) or adaptive step sizes ([3, Eq.(5.2)]).
>
> **Q4. Can the authors explain why the same line of reasoning as for $\gamma, \mu > 0$ cannot (or can) be applied to the average-iterate convergence proof when $\gamma=\mu=0$?**\
> A4: Thanks for pointing this out. Our average-iterate convergence holds for any $\gamma, \mu \geq 0$, because in the proof for the average regret bound of PCFR+ (see Appendix F), we do not use the smoothness of the counterfactual value (i.e. Lemma D.4) that require the perturbation factor $\gamma >0$, but only use the optimization analysis of optimistic OMD and the step-size invariance of PCFR+ algorithm in Lemma G.3. Therefore, the average regret bound for PCFR+ holds for any $\gamma, \mu \geq 0$.
>
> **Q5. Is the predictive/optimistic update structurally necessary to obtain step-size invariant guarantees?**\
> A5: No. Actually the non-predictive RTCFR+ algorithm [1] also has the last-iterate convergence guarantee that is invariant to step sizes.
>
> **Q6. Is step-size invariance a better predictor of empirical robustness across a range of untuned learning rates compared to algorithms that do not possess the step-size invariance property**\
> A6: Thanks for this question. The empirical importance of step-size invariance seemingly depends on the scale of EFG, for two main reasons:\
> **(1)** In small-to-medium-size EFG instances (e.g. Kuhn, Leduc), the step-size-invariant algorithm like PCFR+ really obtains stronger empirical performance than those need to tune step size like OMD algorithm over the whole treeplex [4].\
> **(2)** In large-sized EFG instances (e.g. Subgame 3 in Heads-up Limit Texas Hold'em), the state-of-the-art algorithm DCFR actually does not possess the step-size invariance property.
>
> **References**\
> [1] Efficient Last-Iterate Convergence in Solving Extensive-Form Games. NeurIPS 2025.\
> [2] Regret matching+: (in)stability and fast convergence in games. NeurIPS 2023.\
> [3] The power of regularization in solving extensive-form games. ICLR 2023\
> [4] Stable-predictive optimistic counterfactual regret minimization. ICML 2019

---

> > ### Author Rebuttal · Reviewer_ZVd6 · 2026-04-03
> >
> > Thank you for the rebuttal. I appreciate the clarifications and the willingness to revise the presentation. In particular, making the scope explicit as the two-player zero-sum case would improve the paper substantially, and the clarification that predictive updates are not structurally necessary for step-size-invariant guarantees is helpful.
> >
> > That said, my main concerns are only partially resolved. Most importantly, the motivation for discounting remains unelear. The rebuttal explains why the authors chose to study a discounted variant, but it still does not make clear what discounting concretely buys in this paper.
> >
> > The discussion of the rate question is relevant, but it does not fully address whether the slower guarantee under constant step sizes reflects a limitation or mainly an artifact of the analysis. Similarly, the response on the average-iterate proof clarifies that the argument does not rely on the same smoothness ingredient, but it does not fully explain the contrast with the perturbed-game last-iterate proof or whether there is a more direct bridge.
> >
> > Overall, the rebuttal improves my understanding of the paper, but it does not fully resolve the remaining concerns about the role of discounting. I therefore keep my original evaluation.

---

### Official Review · Reviewer_qE6i · 2026-03-01

**Soundness:** 3
**Presentation:** 1
**Significance:** 2
**Originality:** 2
**Overall Recommendation:** 3
**Confidence:** 3

**Summary:**

This paper studies both last and average iterate convergence of CFR variants in EFGs. Under the reward transformation frameworks, the authors generalize the last-iterate convergence of CFR+ to predictive CFR+, for any positive stepsize and any initialization. For average-iterates, the paper shows that RTPCFR+ obtains regret bounded by $O(\sqrt{T})$, with the tradeoff of avoiding careful stepsize scheduling which leads to lower regret in other approaches in the literature. Next, the paper studies a further discounted version of the algorithm, showing last iterate convergence for any positive stepsizes and initialization. Average-iterate convergence to equilibria using this algorithm is also shown for any amount of regularization and perturbation, under appropriate discounting. Finally, experiments are done to show that the algorithms studied are generally comparable or significantly improve upon the performance of standard baseline algorithms in the literature.

**Compliance With Llm Reviewing Policy:**

Affirmed.

**Final Justification:**

My concerns were not addressed adequately by the rebuttal, and so I maintain my score.

**Key Questions For Authors:**

- The last-iterate convergence results in this paper and in prior work are asymptotic in nature. What are the key challenges towards adapting the convergence proof towards obtaining explicit rates for suitable regimes of parameters? For instance, in normal-form settings one can achieve an explicit logarithmic convergence rate in terms of last-iterate [1]. Would a similar technique apply to two-player zero-sum EFGs with perfect recall?

**Limitations:**

yes

**Strengths And Weaknesses:**

### Strengths
- The paper gives some interesting technical results that could be of interest in future. For instance, the strong notion of `invariant' last-iterate convergence seems to be a useful property which avoids potentially costly or restrictive stepsize regimes for convergence in EFGs.
- The analysis of certain results, mainly Theorems 5.1 and 6.1, generalize techniques from prior work. The tradeoff between carefully designed stepsizes which achieve fast convergence and broadly applicable stepsizes that have slower convergence is also a nice insight from the average-iterate convergence results.


### Weaknesses
- While the technical results seem sound, and the analysis in some cases is novel, the paper suffers from a lack of focus and clarity. The dense notation and lack of discussion of proof techniques makes it difficult to understand which parts of the analysis are taken directly from prior work, and which are novel.
- The average-iterate results seem to add little to the overall contributions of the paper, especially theorem 6.2. If there is a significant contribution here, it is not communicated clearly to the reader. Rather, I feel that a stronger structure for the paper would be to focus on the invariant stepsize regime, and motivate the techniques and algorithmic design that allows for this invariance to hold. As it stands, the paper feels like a collection of results that fill in the gaps in prior work but do not have a clear overarching story or conceptual innovation.
- The experimental section further exacerbates my criticisms of the paper: to me, it seems that the invariance of stepsizes is one of the key technical contributions of the work, since the non-fine-tuned versions of RTPCFR+ and RTPDCFR+ perform comparably or even better than baseline algorithms which are ostensibly fine-tuned. However, the experiments are designed in such a way that interpreting the results is difficult, as fine-tuned variants of RTPCFR+ and RTPDCFR+ are also presented and compared.

Overall, I am curious to understand what the authors see as the main scientific contribution of this work. As it stands, I think the paper contains some interesting ideas and technical results that could be useful to the community, but is written in an unclear manner. Indeed, it seems to be a work that could be significantly improved with a streamlining of the discussions around the main results, and clearer exposition of the novel techniques used in the main text. For this paper to be accepted, I would expect a fairly significant rewrite of the main text, and for that reason I err on the side of rejection, at least for this cycle.

---

> ### Author Rebuttal · Authors · 2026-03-28
>
> **Q1. The dense notation and lack of discussion of proof techniques makes it difficult to understand which parts of the analysis are taken directly from prior work, and which are novel.**\
> A1: Sorry for the confusion. When compared with the last iterate convergence of CFR+ [1], our work has three main novelties:\
> **(1)** For the **last-iterate convergence of PCFR+**: the update rule of CFR+ includes only one OMD step, but the update rule of PCFR+ includes two OMD steps. We choose to apply the first-order optimality condition to the second OMD update step, and upper bound the extra term with the first-order optimality condition of the first OMD update step. Due to the generalization to the two-step OMD, our proofs include the last-iterate convergence of CFR+ as a special case and can be further extended to obtain the last-iterate convergence of other optimistic CFR variants such as PDCFR+ (Remark D.9).\
> **(2)** For the **last-iterate convergence of PDCFR+**: the main challenge lies in demonstrating the step-size invariance. To solve it, for any non-increasing step sizes $\eta^t$, we construct a novel sequence of step sizes ${\eta^{t+1}}'=\frac{\eta^{t'}}{\eta^{t}}\eta^{t+1}$, prove its non-increasing property and the existence of the limit point of $\eta^{t+1'}$ by applying monotone convergence theorem (Lemma G.2), and prove the step-size invariance by induction in Lemma G.3.\
> **(3)** For the **average-iterate convergence of PCFR+**: in contrast to the complicated Blackwell approachability analysis in [2, Prop 2], we utilize our last-iterate convergence analysis for PCFR+ and Lemma D.6 to obtain an improved regret bound for PCFR+ algorithm (Remark 5.4).
>
> **Q2. I feel that a stronger structure for the paper would be to focus on the invariant step size regime.**\
> A2: Many thanks for this suggestion. In the revised version, we will focus on the invariant step size regime and add more experiments for the invariance of step size in the main text.\
>
> **Q3. The experiments are designed in such a way that interpreting the results (i.e the invariance of step sizes) is difficult.**\
> A3: Thanks for pointing this out. We evaluate the invariance of PCFR+/PDCFR+ to step sizes across 9 EFG instances over 20,000 iterations, and find that such property holds in practice (but may lead to slow convergence with certain step sizes). Due to the limited space, we only report the last-iterate exploitability of PCFR+ in 5 EFGs.
> |$\eta$| Kuhn | Leduc | Battleship | Liar’s Dice 4|Goofspiel 4 |
> | :-----:| :-----:| :----: | :----: | :-----:| :-----:|
> |0.1| 2.08e-17 | 6.37e-07 | 8.37e-06 | 2.78e-17 | 1.25e-16 |
> |1| 1.04e-16 | 9.87e-10 | 8.99e-15 | 0 | 0 |
> |5| 0 | 0.025 | 0 | 5.55e-17 | 1.64e-15 |
> |10| 6.94e-17 | 0.031 | 0 | 2.15e-16 | 3.12e-16|
>
> **Q4. I am curious to understand what the authors see as the main scientific contribution of this work.**\
> A4: Thanks. Our main scientific contributions are summarized in two aspects:\
> **(1)** Theoretically, for the first time, we provide the (asymptotic) last-iterate convergence of optimistic CFR methods (PCFR+ and PDCFR+) in solving perturbed regularized EFG. By decreasing the perturbation/regularization factors, RTPCFR+/RTPDCFR+ converge to the set of NEs of original EFG. We also provide improved regret bounds for PCFR+/PDCFR+ via a concise analysis \
> **(2)** In experiments, we validate the good empirical performance of the last-iterate strategy of RTPCFR+/RTPDCFR+. RTPCFR+ achieves faster convergence than other baselines (including their last-iterate strategies in Fig 1 and their averaged strategy in Fig 4) in most medium-sized EFGs, and RTPDCFR+ outperforms existing methods (e.g. DCFR) in large-sized EFG in Fig 3.\
> Therefore, this work demonstrates the last-iterate convergence of RT-based optimistic CFRs both theoretically and empirically, shedding light on running CFRs without averaging strategies.
>
> **Q5. What are the key challenges towards adapting the convergence proof towards obtaining explicit rates?**\
> A5: Thanks. Our explanations are two-fold:\
> **(1)** If we run online algorithms (like OMD) with dilated regularization over the whole treeplex to update the sequence-form strategy, then we can employ techniques from normal-form game to achieve a linear convergence rate in the last iterate (like [3, Thm 5.1]). \
> **(2)** However, for CFR variants that update the behavioral strategy on local infosets, achieving its explicit convergence rate is more difficult because the individual iterates of CFR variants are obtained from independent regret-minimizers and may reach an NE of the game asynchronously. Thus, only asymptotic last-iterate convergence for CFR-type algorithms has been established (see [1, Thm 4.1] and [3, Thm 5.3]).
>
> **References**\
> [1] Efficient Last-Iterate Convergence in Solving Extensive-Form Games. NeurIPS 2025\
> [2] Faster Game Solving via Predictive Blackwell Approachability. AAAI 2021\
> [3] The power of regularization in solving extensive-form games. ICLR 2023

---

> > ### Author Rebuttal · Reviewer_qE6i · 2026-04-04
> >
> > Thank you to the authors for the comprehensive rebuttal. After reading the rebuttal and the other reviews, I am still of the opinion that the paper would need a significant rewrite to include the proposed changes: i) including more discussion on which techniques are novel, ii) increased focus on the stepsize regime considered, and iii) restructuring the presentation and overall writing. Due to the significance of the revisions required, I still err on the side of rejection for this cycle.

---

### Official Review · Reviewer_jMLp · 2026-03-14

**Soundness:** 3
**Presentation:** 3
**Significance:** 2
**Originality:** 2
**Overall Recommendation:** 3
**Confidence:** 3

**Summary:**

This paper studies the last-iterate convergence of variants of Counterfactual Regret Minimization (CFR) algorithms to the Nash Equilibrium (NE) of extensive-form games (EFGs). They work in the perturbed regularized two-player zero-sum extensive-form game setting: (1) perturbation means the local strategy simplex is clipped to make sure every action receives probability at least $\gamma$; (2) each player's utility is also regularized by a strongly convex function controlled by parameter $\mu$. Such a perturbed regularized game admits a unique Nash equilibrium, which converges to the original game's Nash equilibria when $\gamma, \mu \to 0$. Previous works have established asymptotic last-iterate convergence of CFR composed with OMD methods, and the regret matching+ (RM+) method. This paper considers the PCFR+ algorithm (CFR composed with the predictive RM+ method) and show asymptotic last-iterate convergence in the perturbed regularized game. They also develop a slightly sharper bound for the $O(1/\sqrt{T})$ average-iterate convergence rate of PCFR+ in the regularized game and th original game. They then extend these two results to the variant of PCFR+ that incorporates discouting regret vectors. They show experimental results on 9 extensive-form games that these two algorithms are faster in last iterate than other popular methods.

**Compliance With Llm Reviewing Policy:**

Affirmed.

**Final Justification:**

After the rebuttal, I have concerns about the technical contribution that remains unsolved.  I agree with other reviewers and think this paper needs a major revision before acceptance.

**Key Questions For Authors:**

See weaknesses. Especially, please explain the inconsistency in Theorem 5.1/6.1.

**Limitations:**

yes

**Strengths And Weaknesses:**

Strengths:
1. Understanding the last-iterate/average-iterate convergence of the CFR+ family algorithm is an important question. This paper provides new asymptotic last-iterate convergence results for PCFR+ in the perturbed regularized game and a sharper average-iterate convergence rate in the original game. The analysis also extends to recent variants that incorporate discounted regret vectors. These two algorithms are practical and exhibit good numerical performance.

Weaknesses: There are some issues with the technical contribution and the presentation.
1. The technical contribution seems incremental given [1]. While [1] analyzes only the CFR+ algorithm, extending to PCFR+ seems straightforward. The challenges described in Remark D.9 do not seem particularly critical and can be addressed within the standard analysis framework for optimistic methods.
2. The presentation of the results is inconsistent and confusing. In the abstract, the authors present results for PCFR+, but all the theorems (Theorem 5.1, 5.3, 6.1, 6.2) are for the RTPCFR+/RTPDCFR+ algorithm (only presented in Appendix B). In my understanding, Theorems 5.1 and 6.1 on the last-iterate convergence only apply to PCFR+ and PDCFR+ (not the RT version) in the perturbed regularized game with fixed $\gamma, \mu > 0$. The RT versions of these two algorithms gradually decrease the values of $\gamma$ and $\mu$ (line 12) and restart to ensure last-iterate convergence in the original game.
3. However, this introduces another issue. For fixed $\ gamma$ and $ \mu$, the game is strongly monotone, and I think one can prove a linear convergence rate in the last iterate. But Theorems 5.1 and 6.1 only provide asymptotic convergence. Proving a linear convergence is important, as otherwise $T_u$ needs to be very large to ensure the convergence of the RTPCFR+/RTPDCFR+ algorithm in the original game. In fact, such a discussion is also missing.
4. It is unclear how a Nash equilibrium in the perturbed regularized game performs in the original game. Such a discussion is important.
5. The experimental results are incomplete. To ensure fair comparison, please consider outputting the linear averaging iterates of PCFR+/CFR+ as it often performs better than the uniform averaging iterates.

[1] Efficient Last-Iterate Convergence in Solving Extensive-Form Games, Linjian Meng, Tianpei Yang, Youzhi Zhang, Zhenxing Ge, Shangdong Yang, Tianyu Ding, Wenbin Li, Bo An, Yang Gao. NeurIPS 2025.

---

> ### Author Rebuttal · Authors · 2026-03-28
>
> **Q1. The technical contribution seems incremental given [1]. While [1] analyzes only the CFR+ algorithm, extending to PCFR+ seems straightforward.**\
> A1: Thanks, we agree that our work borrows the idea of the last-iterate convergence of CFR+ in [1], and extend their techniques to the last-iterate convergence of PCFR+/PDCFR+. Strictly speaking, our proofs are truly not that technical, but still have three novelties:\
> **(1)** For the **last-iterate convergence of PCFR+**: The update rule of CFR+ in solving Eq.(6) includes only one OMD step, but the update rule of PCFR+ includes two OMD steps. We then choose to apply the first-order optimality condition to the second OMD update step, and upper bound the extra term with the first-order optimality condition of the first OMD update step. Due to the generalization to the two-step OMD, our proofs include the last-iterate convergence of CFR+ as a special case and can be further extended to obtain the last-iterate convergence of other optimistic CFR variants such as PDCFR+.\
> **(2)** For the **last-iterate convergence of PDCFR+**: the main challenge lies in demonstrating the step-size invariance. To solve it, for any non-increasing step sizes $\eta^t$, we construct a novel sequence of step sizes ${\eta^{t+1}}'=\frac{\eta^{t'}}{\eta^{t}}\eta^{t+1}$, prove the non-increasing property as well as the existence of the limit point of $\eta^{t+1'}$ by applying monotone convergence theorem (Lemma G.2), and prove $\hat{{\theta}} _{I}^{t'}=\frac{\eta^{t'}}{\eta^{t}}\hat{{\theta}} _{I}^{t}$ by induction in Lemma G.3, finally leading to the invariant last-iterate convergence for PDCFR+.\
> **(3)** For the **average-iterate convergence of PCFR+**: in contrast to the complicated Blackwell approachability analysis in [1, Prop 2], we utilize our concise last-iterate convergence analysis for PCFR+ and Lemma D.6 (the relation between counterfactual regret and individual regret) to obtain an improved regret bound for PCFR+ algorithm (Remark 5.4).
>
> **Q2. The presentation of the results is inconsistent and confusing. In the abstract, the authors present results for PCFR+, but all the theorems (Theorem 5.1, 5.3, 6.1, 6.2) are for the RTPCFR+/RTPDCFR+ algorithm.**\
> A2: Sorry for the confusion. We will revise the statements of theorems in the new version and our explanations are two-fold: \
> **(1)** It is true that all our theorems (especially the proofs) only provide last-/average-iterate convergence for PCFR+/PDCFR+ in solving perturbed regularized EFGs. \
> **(2)** Then, as shown in [3, Thm 4.3], gradually decreasing the values of $\gamma$ and $\mu$ guarantees the convergence of NEs of perturbed regularized EFGs to the set of NEs of original EFG, which corresponds to the last-iterate convergence of RTPCFR+/RTPDCFR+ algorithm.
>
> **Q3. For fixed $\gamma, \mu$, the game is strongly monotone, and I think one can prove a linear convergence rate in the last iterate. But Theorems 5.1 and 6.1 only provide asymptotic convergence.**\
> A3: Thanks for pointing this out. There are two reasons why we can only obtain asymptotic last-iterate convergence:\
> **(1)** It is true that for fixed $\gamma$ and $\mu$, the algorithms (like OMD) that directly update the sequence-form strategy over the whole treeplex can achieve a linear convergence rate in the last iterate (such as [2, Thm 4.1] and [3, Thm 5.1]). However, in practice, such algorithms can not outperform CFR variants that update the behavioral strategy over the infosets (see [1, Sect 5], [2, Appendix B] and our experiments). \
> **(2)** For CFR variants in solving perturbed regularized EFGs, achieving its explicit convergence rate is more difficult because unlike OMD, the individual iterates of CFR variants are obtained from independent regret-minimizers and may reach an NE of the game asynchronously. For CFR-type algorithms in this setting, only asymptotic last-iterate convergence has been established (see [1, Thm 4.1] and [2, Thm 5.3]).
>
> **Q4. It is unclear how a Nash equilibrium in the perturbed regularized game performs in the original game.**\
> A4: Thanks. According to [3, Thm 4.3], by continuously decreasing the value of $\gamma, \mu$ (with $\mu$ decreasing faster than $\gamma$) and updating ${r}$ to $\hat{{x}}^{*,\gamma}$, the sequence of NEs of perturbed regularized EFGs converges to the set of NEs of original EFG.
>
> **Q5. Please consider outputting the linear averaging iterates of PCFR+/CFR+ as it often performs better than the uniform averaging iterates.**\
> A5: Thanks for this suggestion. We examined the code for the experiments once again and confirmed that, in the experiments of Fig.4, we use linear averaging iterates of PCFR+/CFR+ instead of uniform averaging iterates.
>
> **References**\
> [1] Efficient Last-Iterate Convergence in Solving Extensive-Form Games. NeurIPS 2025.\
> [2] The power of regularization in solving extensive-form games. ICLR 2023.\
> [3] Learning extensive-form perfect equilibria in two-player zero-sum sequential games. AISTAT 2024.

---

> > ### Author Rebuttal · Reviewer_jMLp · 2026-04-03
> >
> > I thank the authors for the rebuttal.
> >
> > I have a follow-up question on the relationship between NE in the perturbed regularized game and NE in the original game. I understand that by decreasing the values of $\gamma$ and $\mu$, the perturbed regularized NE would converge to an NE in the original game. But I think it would be very helpful to provide an upper bound showing that a $(\gamma,\mu)$-perturbed regularized NE is an $\epsilon(\gamma,\mu)$-approximate NE in the original game. Such an upper bound would help practitioners to set appropriate $(\gamma,\mu)$ when they have a target accuracy.
> >
> > Thank you!

---

### Official Review · Reviewer_rMj7 · 2026-03-14

**Soundness:** 3
**Presentation:** 2
**Significance:** 2
**Originality:** 2
**Overall Recommendation:** 3
**Confidence:** 4

**Summary:**

This paper establishes the asymptotic last-iterate convergence of predictive CFR+ (PCFR+) and predictive discounted CFR+ (PDCFR+) for two-player zero-sum imperfect-information extensive-form games (IIEFGs). Experiments on several game environments validate the effectiveness of PCFR+ and PDCFR+.

**Compliance With Llm Reviewing Policy:**

Affirmed.

**Final Justification:**

My main concern regarding the technical novelty remains unresolved.

**Key Questions For Authors:**

Please see my questions above.

**Limitations:**

Please see my questions above.

**Strengths And Weaknesses:**

**Strengths**
1. **Motivation:** This paper studies an important topic of learning IIEFGs with last-iterate convergence.

**Weaknesses**
My main concerns are as follows:

1. **Presentation:** Many parts of this work are clear. Yet, I still find some key parts of this work a bit confusing or hard to follow. For instance:

   a) What are the differences between **PCFR+**, **RTCFR+**, and **RTPCFR+**? In the abstract, the authors say that “we provide for **PCFR+** the last-iterate convergence.” On Lines 96–99, the authors say that “We develop … of **PCFR+** in solving … for **RTCFR+** as ...” while very shortly afterward, on Line 104, the authors say “for the proposed **RTPCFR+** algorithm.” The focus of the audience is forced to change from **PCFR+** and **RTCFR+** to **RTPCFR+** without any explanations.

   b) To me, the last two paragraphs of Sec. 4 discuss the same thing: perturbing the original game by adding a small amount of regularization. From this perspective, it makes me a bit confused why these two paragraphs have two different subtitles. It would probably be better to merge these two paragraphs.

2. **Originality, Significance, and Novelty:** My other main concern is that some contributions of this work are not very clear to me. PCFR+ has already been investigated both theoretically and empirically in previous works (e.g., [1]). Also, the discounting technique in CFR+ has already been proposed in, *e.g.*, [2]. Reward Transformation (RT) has also appeared in many recent works. In this sense, the technical value of the proposed RTPCFR+/RTPDCFR+ algorithms seems a bit limited. Moreover, it is also not very clear to me whether there are additional technical difficulties for obtaining the theoretical results in this work, and whether new techniques are proposed to address these difficulties, given the previous analysis techniques in works studying last-iterate convergence for CFR+.

----
[1] Farina et al. *Faster Game Solving via Predictive Blackwell Approachability: Connecting Regret Matching and Mirror Descent.* AAAI, 2021.

[2] Brown et al. *Solving Imperfect-Information Games via Discounted Regret Minimization.* AAAI, 2019.

---

> ### Author Rebuttal · Authors · 2026-03-28
>
> **Q1. From Lines 96–99 to Line 104, the focus of the audience is forced to change from PCFR+ and RTCFR+ to RTPCFR+ without any explanations.**\
> A1: Sorry for the confusion. Our explanations are as follow and will be added in the revised version:\
> **(1)** For any fixed $\gamma$ and $\mu$, we use PCFR+ algorithm to solve the perturbed regularized EFG in Eq.(6), and our Thms 5.1/5.3 provide last-iterate convergence and average-iterate convergence (of order $O(1/\sqrt{T_\mu})$) respectively for PCFR+ to the NE of the perturbed regularized problem in Eq.(6).\
> **(2)** After $T_\mu$ iterations, we decrease $\gamma$ and $\mu$, and still run PCFR+ to solve Eq.(6) with the decreased $\gamma, \mu$. Such process leads to our RTPCFR+ algorithm, which is guaranteed to converge to the set of NEs (e.g. extensive-form perfect equilibria) of original EFG by continuously decreasing $\gamma$ and $\mu$,.
>
> **Q2. It would probably be better to merge the last two paragraphs in Sect 4 that discuss the same thing.**\
> A2: Thanks for this suggestion. The last two paragraphs truly discuss the same thing of RT framework, where the first one discusses its high-level benefit and the second one discusses it detailed formula. We will merge them in the revised version.
>
> **Q3. PCFR+, discounting technique, and Reward Transformation (RT) have been investigated in previous works. In this sense, the technical value of the proposed RTPCFR+/RTPDCFR+ algorithms seems a bit limited.**\
> A3: Thanks. It is true that PCFR+, discounting technique and RT have been proposed in previous works, and so our technical value does not lie in the design of RTPCFR+/RTPDCFR+ algorithms, but in the provable last-iterate convergence and the state-of-the-art empirical performance in all EFG instances. Concrete explanations are two-fold:\
> **(1)** For the first time, we provide the (asymptotic) last-iterate convergence of optimistic CFR methods (PCFR+ and PDCFR+) in solving perturbed regularized EFG. By continuously decreasing the perturbation and regularization factors, the corresponding RTPCFR+/RTPDCFR+ are guaranteed to converge to the set of NEs of original EFG.\
> **(2)** The second main contribution lies in the validation of good empirical performance of the last-iterate strategy of RTPCFR+/RTPDCFR+. Our fine-tuned RTPCFR+ algorithm achieves faster convergence than other baselines (including their last-iterate strategies in Fig 1 and their averaged strategies in Fig 4) in most small-to-medium-sized EFGs, and RTPDCFR+ outperforms existing methods (e.g. DCFR) in large-sized EFG in Fig 3.\
> Thus, this work demonstrates the last-iterate convergence of RT-based optimistic CFRs both theoretically and empirically, shedding light on running CFRs without averaging their strategies.
>
> **Q4. It is also not clear to me whether there are additional technical difficulties for obtaining the theoretical results, and whether new techniques are proposed to address these difficulties.**\
> A4: Fairly speaking, we borrow the idea of the last-iterate convergence of CFR+ in [3], and extend their techniques to the last-iterate convergence of PCFR+/PDCFR+ (Remarks D.9 and E.2). There are three main technical novelties:\
> **(1)** For the **last-iterate convergence of PCFR+**: The update rule of CFR+ in solving Eq.(6) includes only one OMD step, but the update rule of PCFR+ includes two OMD steps. We choose to apply the first-order optimality condition to the second OMD update step, and upper bound the extra term with the first-order optimality condition of the first OMD update step. Due to the generalization to the two-step OMD, our proofs include the last-iterate convergence of CFR+ as a special case and can be further extended to obtain the last-iterate convergence of other optimistic CFR variants such as PDCFR+.\
> **(2)** For the **last-iterate convergence of PDCFR+**: the main challenge lies in demonstrating the step-size invariance. To solve it, for any non-increasing step sizes $\eta^t$, we construct a novel sequence of step sizes $\eta^{t+1’}=\frac{\eta^{t'}}{\eta^{t}}\eta^{t+1}$, and prove the non-increasing property as well as the existence of the limit point of $\eta^{t+1’}$ by applying monotone convergence theorem (Lemma G.2). The above analysis is critical to prove $\hat{{\theta}} _{I}^{t'}=\frac{\eta^{t'}}{\eta^{t}}\hat{{\theta}} _{I}^{t}$ by induction in Lemma G.3, finally leading to the invariant last-iterate convergence for PDCFR+.\
> **(3)** For the **average-iterate convergence of PCFR+**: in contrast to the complicated Blackwell approachability analysis in [1, Prop 2], we utilize our concise last-iterate convergence analysis for PCFR+ and Lemma D.6 (the equivalence between counterfactual regret and individual regret) to obtain an improved regret bound for PCFR+ algorithm (Remark 5.4).
>
> **References**\
> [1] Faster Game Solving via Predictive Blackwell Approachability. AAAI, 2021.\
> [3] Efficient Last-Iterate Convergence in Solving Extensive-Form Games. NeurIPS 2025.

---

> > ### Author Rebuttal · Reviewer_rMj7 · 2026-04-03
> >
> > I thank the authors for their explanations, which resolve my concerns about the presentation. While I understand that there are indeed some new modifications needed in the problem studied in this work, from my point of view, these modifications are relatively standard, given the fact that adaptively decreasing the strength of the regularization and perturbation to obtain last-iterate convergence in EFGs has already appeared in various prior studies (say, [1]). From this perspective, this work seems more like using existing analysis techniques for obtaining last-iterate convergence to analyze existing algorithms (PCFR+, RTPCFR+, and RTPDCFR+). Therefore, at this point, I’m inclined to maintain my current score for this work.
> >
> > [1] Cai et al. Uncoupled and Convergent Learning in Two-Player Zero-Sum Markov Games with Bandit Feedback. NeurIPS, 2023.

---

### Decision · Program_Chairs · 2026-04-30

**Decision:**

Reject

**Comment:**

This paper mainly studies the asymptotic last-iterate convergence of several variants of CFR algorithms to the Nash equilibrium of perturbed extensive-form games.
While the reviewers recognized the importance of the problem, they had concerns about the paper's clarity and most could not fully appreciate its contributions to the community. The reviewers were unanimous in their opinion that the paper would benefit from revision and another round of review.